# A heterozygous *CEBPA* mutation disrupting the bZIP domain in a *RUNX1* and *SRSF2* mutational background causes MDS disease progression

Ruba Almaghrabi[1,2,26], Alyahyawi Yara[1,3,26], Peter Keane [4], Syed A. Mian [5], Khadidja Habel [5], Amelia Atkinson[1], Carl Ward [1], Rachel Bayley[1], Claudia Sargas[6], Pablo Menendez[7,8,9,10,11,12], George J. Murphy [13,14], Turki Sobahy[15], Mohammed A. Baghdadi[15], Arwa F. Flemban [16], Saeed M. Kabrah [17], Raul Torres-Ruiz[9,18,19,20,21], Eirini P. Papapetrou [22,23], Ildem Akerman [24], Manoj Raghavan [1], Eva Barragan [6,8], Dominique Bonnet [5], Constanze Bonifer [1,25] & Paloma Garcia [1] ✉

Myelodysplastic syndrome disease (MDS) is caused by the successive acquisition of mutations and thus displays a variable risk for progression to AML. Mutations in *CEBPA* are commonly associated with a high risk of disease progression, but whether they are causative for AML development is unclear. To analyse the molecular basis of disease progression we generated MDS patient-derived induced pluripotent stem cells from a low risk male patient harbouring *RUNX1*/*SRSF2* mutations. This experimental model faithfully recapitulates the patient disease phenotypes upon hematopoietic differentiation. Introduction of a frameshift mutation affecting the C/EBPα bZIP domain in cells from low-risk stages mimicks disease progression by reducing clonogenicity of myeloid cells, blocking granulopoiesis and increasing erythroid progenitor self-renewal capacity. The acquisition of this mutation reshapes the chromatin landscape at distal cis-regulatory regions and promotes changes in cellular composition as observed by single cell RNAseq. Mutant C/EBPα is therefore causative for MDS disease progression. Our work identifies mutant *CEBPA* as causative for MDS disease progression, providing a new isogenic MDS experimental model for drug screening to improve diagnostic and therapeutic strategies.

During ageing hematopoietic stem/progenitor cells (HSPCs) accumulate somatic mutations leading to clonal haematopoiesis of indeterminate potential (CHIP) and the development of blood disorders such as myelodysplastic syndromes (MDS). MDS are a heterogeneous group of clonal haematological diseases characterized by cytopenia and impaired haematopoiesis, with 30% of patients eventually progressing to acute myeloid leukaemia (AML)[1,2]. Next generation sequencing (NGS) studies have shed some light on the clonality of the disease, (i) showing that MDS founding clones contain mutations which persist when patients progress to AML[3,4], and (ii) that commonly mutated genes such as *DNMT3A, TET2, ASXL1, TP53* and *SF3B1*, are important for initiating the disease[5,6]. Patients with high-risk MDS are often treated with DNA demethylating agents such as azacytidine, but this treatment proves ineffective in more than half of the patients

(reviewed in ref. [7]). The reasons behind treatment failure and AML progression have not been thoroughly investigated due to insufficient cell numbers that can be obtained from MDS patients and the lack of animal model systems that can recapitulate the spectrum of genetic mutations and chromosome alterations. The reprogramming of MDS cells into induced-pluripotent stem cells (iPSCs) has proven to be of great utility in identifying key disease associated genes such as those located in del(7q)[8]; determining the order of mutations contributing to the clonal evolution of a MDS patient harbouring t(4;12), SF3B1, EZH2 and del5q mutations[9], and in constructing a phenotypic roadmap of the clonal evolution of MDS to AML with NRAS as a driver of disease progression[10].

However, to understand the precise role of other common mutations associated to disease progression leading to clonal heterogeneity, disease progression and high variability in disease phenotype requires further investigation. It is known that mutation in transcription factors regulating myeloid differentiation, in particular RUNX1 and CCAAT/ enhancer binding protein alpha (C/EBPα) are common in the progression from MDS to secondary AML[11–13]. Of particular interest is the acquisition of mutations in the bZIP domain of CEBPA, which is required for DNA binding and dimerization. The majority of bZIP mutations are in frame insertions/deletions whereas frameshift insertions/deletions or nonsense mutations causing a premature translational termination are less common[13]. In fact, the current World Health Organization (WHO) AML classification considers C-terminal $CEBPA^{bZIP}$ mutations as a distinct entity with favourable prognosis, better overall survival and lower risk of relapse[14,15]. Most studies are based on mutations acquired in the N-terminal transactivation domain and bZIP C-terminal domain of the protein. However, a third of the CEBPA single mutations occurred between these two regions[16]. How these mutations affect the bZIP domain in the context of MDS have not been explored and despite being associated to disease progression it is unknown to what extent they are causative for a high risk of MDS disease progression.

Here, we report the generation of MDS iPSC cells derived from a patient diagnosed with low risk MDS which are characterized by normal karyotype and carry SRSF2 and RUNX1 mutations. The patient then progressed to high risk MDS acquiring heterozygous disruption of the C/EBPα bZIP domain. We recapitulated the high-risk phenotype by introducing a similar heterozygous C/EBPα bZIP domain disruption through CRISPR-Cas9 mediated genome editing of low-risk iPSC cells, by introducing a frameshift mutation in the mid region of the protein ($CEBPA^{mut}$). The haematological phenotype of these isogenic lines revealed a marked diserythropoiesis in the low-risk lines, which was accentuated during disease progression, mimicking the clinical phenotype observed in the patient. Disruption of the C/EBPα bZIP domain in $CEBPA^{mut}$ HPCs led to a block in granulocytic differentiation, skewed differentiation towards the erythroid lineage and acquisition of self-renewal properties of aberrant erythroblasts. These changes were accompanied by alterations in the chromatin of genes important for myeloid differentiation and AML such as MAF, CEBPE, CELSR3 and RUNX1. Moreover, disruption of the C/EBPα bZIP domain led to changes in cellular composition. Finally, our work revealed that the erythroid differentiation and self-renewal capacity observed in the high-risk iPSC lines was abolished when SRSF2 and RUNX1 mutations were rescued by CRISPR. Our work therefore defines a causative role of bZIP domain CEBPA mutations as a driver of disease progression in the context of SRSF2 and RUNX1 mutations.

## Results

### Generation of iPSCs from a low-risk MDS patient harbouring RUNX1 (Gly217fs) and SRSF2 (Pro95His) mutations

To delineate a molecular and phenotypic roadmap to MDS progression, we selected a patient (MDS27) who progressed from low-risk to high-risk MDS and finally to AML, and for which longitudinal samples were available. This patient was diagnosed with a multilineage dysplasia and refractory anaemia and presented with a normal karyotype (low-risk MDS). Two years later, after an initial response to daunorubicin and cytarabine (DA), the number of blast cells increased to 9–17%; the patient had progressed to high-risk MDS and was treated with 5-Azacitidine demethylating chemotherapy agents (5-AzaC) but failed to respond (Supplementary Fig. 1A–D). Mutational screening from the sample at diagnosis revealed the presence of mutations in RUNX1 (Gly217fs) and SRSF2 (Pro95His), whilst after disease progression the patient acquired an additional mutation in the C/EBPα mid region domain (Gly257fs) changing the open reading frame. This mutation altered the sequence of the DNA binding domain and leading to a truncated protein with no dimerization domain (Fig. 1A and Supplementary Data Table 1).

Somatic reprogramming to hiPSC from this patient was performed on peripheral blood mononuclear cells (PBMCs) at the time of diagnosis using Sendai virus (SeV), a non-integrative system[17]. 24 individual iPSC colonies were isolated, expanded and established as MDS27 low-risk iPSC lines. Genotyping screening confirmed that all the hiPSC colonies examined (15 colonies in total) were positive for the same RUNX1 and SRSF2 mutations present in the original PBMCs (Supplementary Data Table 1). Mutational screening confirmed that no isogenic healthy control clones from MDS27 were generated nor colonies with different combinations of these mutations, despite performing somatic reprogramming on three different occasions and with two different non-integrative methods (episomal and SeV)[18]. These findings suggest that cells harbouring these two mutations seem to be more amenable to reprogramming.

Given that no isogenic healthy iPSCs were obtained, a published non-isogenic iPSC line generated from healthy PBMCs was used as a control, namely BU3.10 (CREM003i-BU3C2; WiCell). Three different hiPSC clones generated from low-risk MDS27 by SeV (C8, C11, C22) were used for further characterization. All three clones maintained a normal karyotype (Supplementary Fig. 1E), were positive for the expression of pluripotent markers TRA1-81, NANOG and SOX2[19,20] (Supplementary Fig. 1F) and showed positive alkaline phosphatase staining (AP)[21,22] (Supplementary Fig. 1G). Furthermore, all clones were capable of differentiation to the three germ layers as shown by the expression of OTX2, SOX17 and Brachyury when differentiated towards ectoderm, endoderm, and mesoderm, respectively[23–26] (Supplementary Fig. 1H). These clones exhibit all aspects of characteristics of pluripotent stem cells: they are morphologically undifferentiated, positive for AP, express the pluripotent markers and can differentiate into the three germ lineages.

### Low-risk MDS27-iPSC harbouring RUNX1(Gly217fs) and SRSF2(Pro95His) mutations retain myeloid and erythroid progenitor potential

To assess the differentiation potential of the MDS27-iPSC clones towards hematopoietic progenitor cells (HPCs), we used the STEMdiff Hematopoietic protocol from STEM Cell Technology (Supplementary Fig. 2A). The course of differentiation was monitored using CD34 (an HSC and early progenitor marker), CD43 (an early hematopoietic marker which persists in differentiating precursor cells)[27], and CD45 (the key marker of human hematopoietic cells)[28]. Expression of these markers was assessed by flow cytometric analysis on progressive days (10, 12, and 14) to determine the dynamics of hematopoietic differentiation. Three different iPSC clones derived from the MDS27 patient sample (C8, C11, and C22) were investigated to control for possible variations due to the reprogramming process. Throughout the hematopoietic differentiation, the hiPSC control and MDS27 hiPSC clones behaved very similarly (Supplementary Fig. 2B). Typically, by day 14, similar HPC numbers could be obtained from all the clones examined; around 90% of cells were CD43⁺ and 25–35% were CD45⁺/ CD34⁺ (Fig. 1B). In summary, this analysis demonstrates that we successfully differentiated MDS27 hiPSC clones towards early and late hematopoietic cells at similar rates as hiPSC control and thus, that

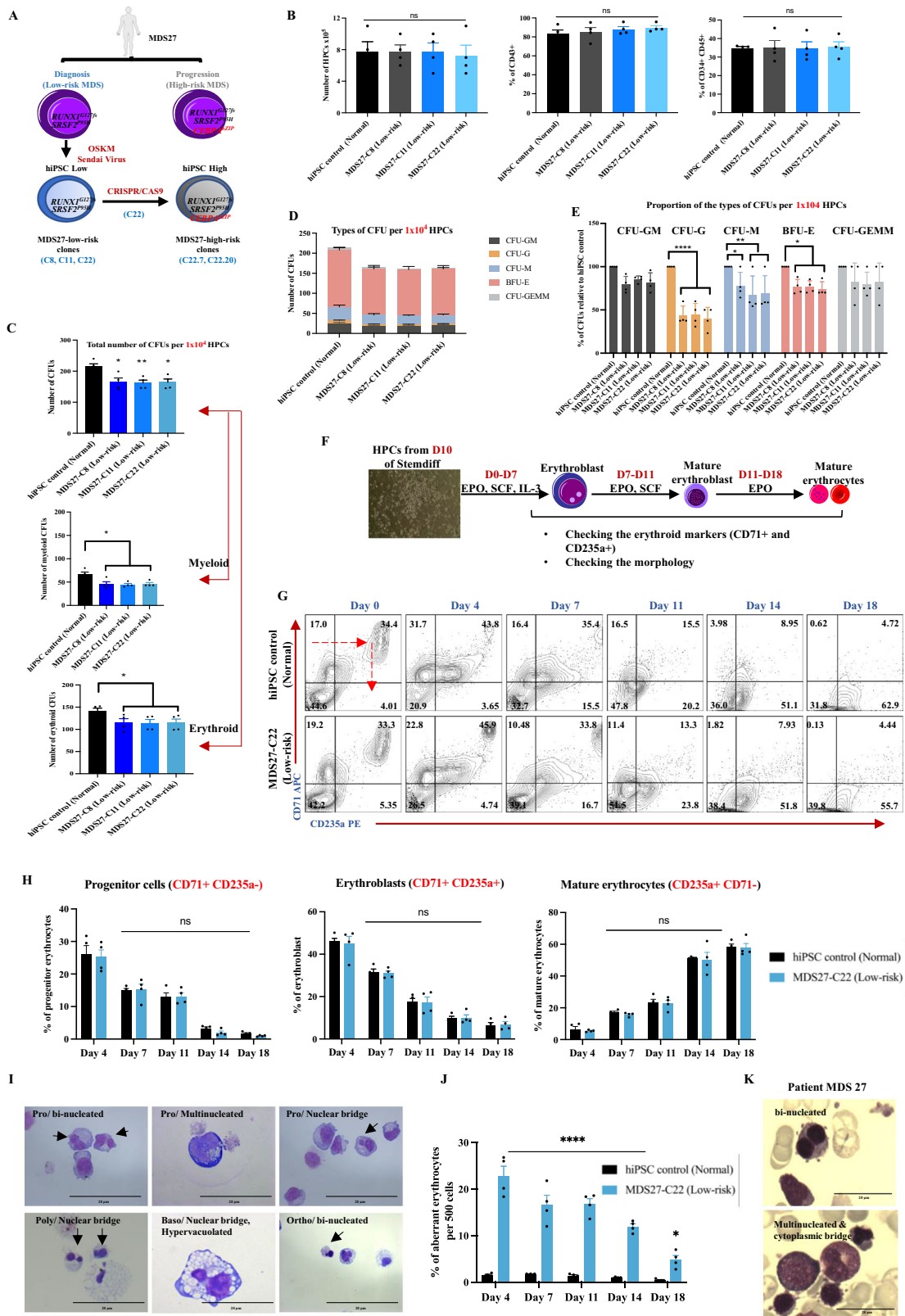

*RUNX1* (Gly217fs) and *SRSF2* (Pro95His) mutations do not affect the generation of HPCs from hiPSC-MDS27 clones.

We next proceeded to assess the myeloid maturation potential of iPSC generated HPCs. HPCs from control and MDS27 hiPSC clones were obtained at day 12 and plated in methylcellulose medium enriched with recombinant cytokines that induce the differentiation into colony forming units of mixed lineage (CFU-GEMM), bursting forming

units of committed erythroid (BFU-E), and colony forming units of myeloid lineage progenitors (CFU-G, CFU-M, CFU-GM) (Supplementary Fig. 2C). On day 14, the number of colonies were counted and scored based on their phenotypic characteristics. All clones derived from the MDS27 patient sample displayed a similar behaviour (Fig. 1C−E). Consistently throughout all independent experiments, we observed a lower number of colonies with HPCs derived from all three

**Fig. 1 | Low-risk MDS27-hiPSC lines are able to differentiation into hematopoietic progenitors in semi-solid medium and into erythroid cells in the liquid culture. A** Genotyping screening for MDS27 patient sample taken at the time of diagnoses and after disease progression as well as for iPSC clones generated from the diagnosis MDS27 patient sample. Artwork generated with powerpoint Bundle-Biology. **B** Bar graph showing the total number of HPCs, percentage of early hematopoietic population (CD43[+]) and late hematopoietic population (CD34[+] CD45[+]) on day 14 of differentiation. Statistical results are presented as mean ± SEM. ns = no significant, one-way ANOVA with Dunnett's multiple comparisons. $N = 4$ independent experiments. Source data are provided as a Source Data file. **C** Total number of CFUs from $10^4$ iPSC-HPC cells grown for 14 days in semi solid medium. Statistical results are presented as mean ± SEM. * $p < 0.05$, ** $p < 0.001$. Two-tailed unpaired $t$-test. $N = 4$ independent experiments. Exact $p$-values can be found in the Supplementary Data Table 5. **D** Number of each type of CFUs after 14 days in semisolid medium. Mean and SEM of different lines are shown. $N = 4$ independent experiments. Source data are provided as a Source Data file. **E** Relative percentage of each type of CFUs for $1 \times 10^4$ of HPCs after 14 days in semisolid media. Statistical results are presented as mean ± SEM and **** $p < 0.0001$, ** $p < 0.001$, * $p < 0.05$, ns: no significant. Two-way ANOVA with Dunnett's multiple comparisons. $N = 4$ independent experiments. Exact $p$-values can be found in the Supplementary Data

Table 5. Source data are provided as a Source Data file. **F** Schematic representation of the experimental set up for the erythroid differentiation of HSPCs. Artwork generated with powerpoint Bundle-Biology. **G** Flow cytometry contour plots for CD71 and CD235a markers on Day 0, 4, 7, 11, 14 and 18 during erythroid differentiation. $N = 4$ independent experiments. Source data are provided as a Source Data file. **H** Percentage of erythrocyte (CD71[+]), erythroblasts (CD71[+] CD235a[+]) and mature erythrocytes (CD235a+) during erythroid differentiation. Mean and SEM are shown (ns, no significant), two-way ANOVA with Dunnett's multiple comparisons $N = 4$ independent experiments. Source data are provided as a Source Data file. **I** Diff-quick stained cytospins showing common aberrant morphology (black arrow) observe in MDS27-C22. The pictures were taken with a Leica DM6000 at ×100 magnification. $N = 3$ independent experiments. **J** Percentage of erythroid cells with aberrant morphology. Statistical results are presented as mean ± SEM. *** $p < 0.0001$ and * $p < 0.05$. Two-way ANOVA with Dunnett's multiple comparisons. $N = 4$ independent experiments. Exact $p$-values can be found in the Supplementary Data Table 5. Source data are provided as a Source Data file. **K** May–Grunwald Giemsa staining stained bone marrow smears from MDS27 patient at the time of diagnosis showing aberrant erythroid cells. The pictures were taken with a Leica DM6000 at ×100 magnification. Taken from Supplementary Fig. 1A.

MDS27-hiPSC clones compared to the healthy hiPSC control. The reduction in the number of myeloid CFUs indicated an impaired hematopoietic colony-forming capacity of these clones (Fig. 1C). All clones derived from the MDS27 patient sample were able to generate all types of CFUs (Fig. 1D), but the proportion of CFU-M and CFU-G, and in lesser extend BFU-E, was lower in comparison to the control line (Fig. 1E). Despite the decrease in hematopoietic colony number in MDS27-hiPSC clones, no difference in the morphology and size of the colonies formed was observed (Supplementary Fig. 2D).

### Low risk MDS27-hiPSC harbouring RUNX1(Gly217fs) and SRSF2(Pro95His) mutations display aberrant erythroid differentiation

Given that anaemia is one of the most common features of MDS patients we sought to determine whether erythropoiesis was impacted in patient-derived low-risk clones and we evaluated the erythroid differentiation potential of generated HPCs in liquid culture. As all the clones generated from low-risk MDS behaved similarly in terms of clonogenic capacity and differentiation potential, one clone was selected for further study (clone 22). Day 10 HPCs obtained from control and MDS27-C22 (low risk) hiPSC were cultured in an established 3-phase erythropoiesis liquid culture over 18 days (Fig. 1F). The culture was monitored every 3–4 days, and the distribution over different erythroid maturation stages was assessed by measuring the expression of CD71 (progenitor marker) and CD235a (Glycophorin A). After culturing HPCs in erythroid media for 7 days, roughly 15% of cells were CD71[+] (erythroid progenitors) and 40% of cells co-expressed CD71[+] and CD235a[+] (erythroblasts), while the percentage of CD71[−] cells expressing the mature marker CD235a[+] was very low at this early stage of the differentiation. This cellular distribution was very similar for both hiPSC control and MDS27-C22 hiPSC (Fig. 1G, H and Supplementary Fig. 2E). As differentiation continued, the percentage of erythroblasts decreased with a concomitant increase in the percentage of the mature cells. On the final day of erythroid differentiation (day 18), the erythroid lineage cells became more mature, with over 55% of the cells expressing CD235a[+] and only 5% of cells being still at the erythroblast stage (CD71[+] CD235a[+]) (Fig. 1G, H and Supplementary Fig. 2F). Analysis of these surface markers by flow cytometry did not highlight any immunophenotypic differences in MDS27 clones compared to the control (Fig. 1H).

Morphological evaluation confirmed that both control and MDS27-C22 hiPSCs had given rise to a heterogeneous population of erythroid cells. All the stages of maturation appeared during the several days of the differentiation (Supplementary Fig. 2G). Nucleated erythroblasts were observed in the cultures from day 4 in both hiPSC control and MDS27-C22 hiPSC clones. Enucleated erythroid cells (mature cells) were

observed as early as day 7 (Supplementary Fig. 2G, yellow arrows). Moreover, the morphological analysis revealed that the erythroid differentiation of MDS27-C22 hiPSCs was dysplastic as cells presented aberrant morphology (binucleated, multinucleated and nuclear bridges) (Fig. 1I, J and Supplementary Fig. 2G), common dysplastic features found on MDS patients[29], including patient MDS27 (Fig. 1K and Supplementary Fig. 1A). The increase in cells with aberrant morphology was evident in the erythroid cultures from MDS27-C22 iPSCs compared to control hiPSC at all time points (Day 4, 7, 11, 14 and 18) (Fig. 1J). These data indicate that *SRSF2* and *RUNX1* mutations promote an aberrant maturation of erythroid cells in low-risk MDS, and importantly, that the disease phenotype observed in the patient could be reproduced in vitro.

### Generation of high-risk MDS27 iPSC by CRISPR-Cas9-mediated genome editing

Mutational screening of the MDS27 patient sample obtained two years after diagnosis revealed the acquisition of a heterozygous C/EBPα frameshift mutation in the mid region of the protein disrupting the open reading frame of the bZIP domain. Somatic reprogramming to iPSC was performed on PBMCs obtained at this stage of disease using SeV, but iPSC generated were not stable in culture and suffered spontaneous non-specific differentiation. Thus, to obtain cells before and after disease progression with the same genetic background (harbouring mutations in *RUNX1* and *SRSF2*), and to understand the contribution of mutations disrupting the C/EBPα bZIP domain to the disease progression, we applied CRISPR-Cas9 technology to introduce a *CEBPA* mid-region mutation disrupting the bZIP domain into the MDS27-C22 hiPSCs generated (hereafter, *CEBPA*[bZIP-fs]), mimicking the disruption of the bZIP domain observed in the MDS patient.

24 h post-nucleofection, cells were dissociated and the GFP[+] cells sorted by FACS (Supplementary Fig. 3A and B). 3 clones (C22.5, C22.7 and C22.20) out of 7 were found to contain a heterozygous mutation based on the T7EI assay (Supplementary Fig. 3C). Sanger sequencing confirmed that all 3 clones harboured the deletion in the target site in one allele of *CEBPA* (Supplementary Fig. 3D). The sequencing data showed that MDS27-C22.5 contained multiple integrations and deletions whilst MDS27-C22.7 and MDS27-C22.20 had both a 43 bp deletion in the targeted region of interest with a consequent premature termination of C/EBPα protein, altering the DNA binding domain of C/EBPα, and thus its functionality (Supplementary Fig. 3E). Characterization of the new clones generated (MDS27- C22 CRISPR control, MDS27-C22.7 (High risk 1) and MDS27-C22.20 (High risk 2) confirmed the absence of aneuploidy (Supplementary Fig. 3F) and positive expression of pluripotent protein markers (TRA1-81, SOX2 and NANOG) (Supplementary Fig. 3G).

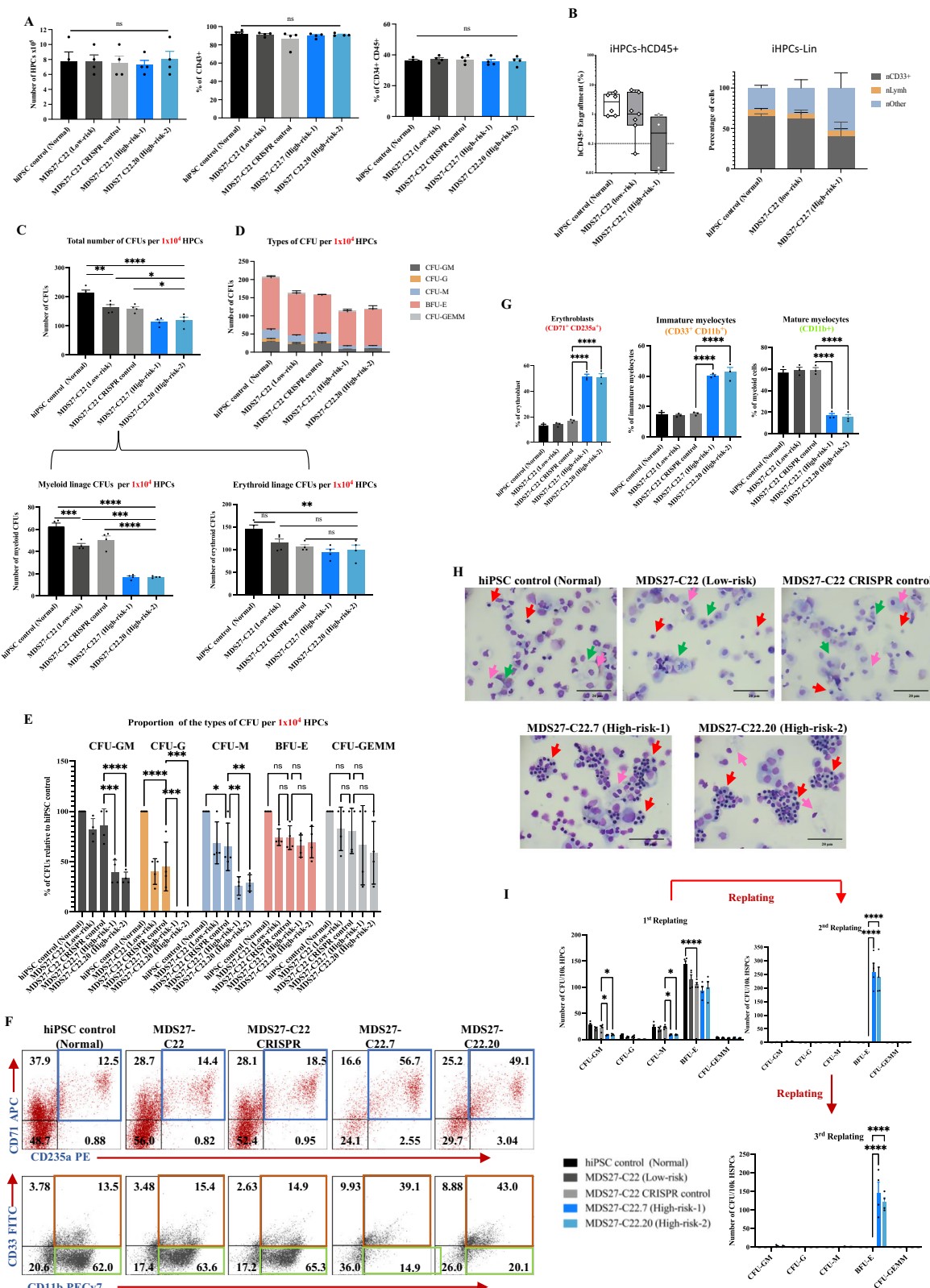

## Erythroid-biased differentiation and increased self-renewal capacity of high-risk MDS-iPSC containing C/EBPα mutations

We next sought to define whether the C/EBPα mutation would alter the hematopoietic differentiation phenotype observed in cells containing mutations in *SRSF2* and *RUNX1*, and thus serve as a model system to study disease progression for this type of MDS. The differentiation of MDS27-C22 (Low risk), MDS27-C22 CRISPR control, MDS27-C22.7

(High risk-1) and MDS27-C22.20 (High risk-2) iPSC to HPCs revealed no significant differences in the number of HPCs or expression of hematopoietic markers at day 14, indicating that inclusion of the *CEBPA*^bZIP-fs mutation did not affect the generation of HPCs (Fig. 2A). In agreement with previous studies, HPCs derived from iPSC did not show in vivo engraftment potential into NBSGW mice when cells were injected via the intravenous or intra-bone routes[30,31] (Supplementary Fig. 4A).

**Fig. 2 | Erythroid-biased differentiation and increased self-renewal capacity of high-risk MDS-iPSC containing a C/EBPα bZIP frameshift mutation. A** Bar graph showing the total number of HPCs, percentage of early hematopoietic population (CD43⁺) and late hematopoietic population (CD34⁺ CD45⁺) on day 14 of differentiation. Total number of HPCs for normal and C22 taken from Fig. 1C. Statistical results are presented as mean ± SEM. ns = no significant, One-way ANOVA with Dunnett's multiple comparisons. $N = 4$ independent experiments. Source data are provided as a Source Data file. **B** Percentage of hCD45⁺ cells (Left graph) engrafted in humanized niches implanted in NSG-SGM3 mice at 12 weeks. Each point represents an individual scaffold; each sample was transplanted into two mice, with technical replicates shown. Box plots display the full range of values (minimum to maximum), with the median indicated by a horizontal line. Lineage distribution (Right graph) within hCD45⁺ cells recovered from humanized niches in NSG-SGM3 mice. Bars represent mean values, and error bars indicate the standard error of the mean (SEM). Source data are provided as a Source data file. **C** Total number of CFUs from 10⁴ –iPSC-HPCsgrown for 14 days in semi-solid medium. Statistical results are presented as mean ± SEM. ****$p < 0$, ***$p < 0.0001$, **$p < 0.001$, *$p < 0.05$ and (ns, no significant). One-way ANOVA with Dunnett's multiple comparisons. $N = 4$ independent experiments. Exact p-values can be found in the Supplementary Data Table 5. Source data are provided as a Source Data file. **D** Number of each type of CFUs after 14 days in semisolid medium. Mean and SEM of different lines are shown. $N = 4$ independent experiments. Source data are provided as a Source Data file.

**E** Relative percentage of each type of CFUs for $1 \times 10^4$ of HPCs after 14 days in semisolid media. Statistical results are presented as mean ± SEM and ****$p < 0.0001$ and *$p < 0.05$, Two-way ANOVA with Dunnett's multiple comparisons. $N = 4$ independent experiments. Exact p-values can be found in the Supplementary Data Table 5. Source data are provided as a Source Data file. **F** Representative flow cytometry panels showing the percentage of erythroblasts cells (blue square) (CD71+ CD235a+), immature myeloid cells, orange square (CD33⁺ CD11b⁺), and mature myeloid cells, green rectangle (CD11b+) obtained from colony assays. $N = 3$ independent experiments. Source data are provided as a Source Data file. **G** Fraction of erythroblasts (CD71⁺ CD235a⁺), immature myeloid (CD33⁺ CD11b⁺) and mature myeloid cells (CD11b+). Mean and SEM are shown. **** $p < 0.0001$, One-way ANOVA with Dunnett's multiple comparisons. $N = 3$ independent experiments. Source data are provided as a Source Data file. **H** Diff-quick stained cytospins from colony assay showing erythroid cells (Red arrows), granulocytes (Green arrows) and monocytes (Pink arrows). The pictures were taken with a Leica DM6000 at ×40, 20 μm scale bar. $N = 3$ independent experiments. **I** Number of CFUs obtained from control, WT clones and mutant clones after second and third re-plating, each maintained for 14 days. Mean and SEM are shown. ****$p < 0.0001$, *$p < 0.05$, Two-way ANOVA with Dunnett's multiple comparisons. $N = 4$ independent experiments. Exact $p$-values can be found in the Supplementary Data Table 5. Source data are provided as a Source Data file.

However, using a humanized NSG-SGM3 mice transplant model[32], the HPC cells derived from healthy, low-risk and high-risk representative clones showed engraftment 12 weeks after transplantation (Fig. 2B and Supplementary Fig. 4B, C). The majority of the engrafted cells belonged to myeloid lineage, consistent with findings from previous studies (Fig. 2B). The differentiation potential and clonogenic capacity of HPCs derived from hiPSC control, MDS27-C22 (Low risk), MDS27-C22 CRISPR control, MDS27-C22.7 (High risk-1) and MDS27-C22.20 (High risk-2) hiPSCs was assessed by a colony assay in methylcellulose medium (Supplementary Fig. 4D, E). HPCs from both controls and low-risk iPSCs, gave rise to both myeloid and erythroid CFUs whilst the HPCs derived from high-risk iPSCs (containing the *CEBPA*^bZIP-fs mutation), exhibited a significant reduction in their clonogenic capacity affecting only the myeloid lineage (Fig. 2C and D and Supplementary Fig. 4E). Scoring the type of CFUs showed that *CEBPA*^bZIP-fs mutant clones were not able to differentiate to CFU-G but were mainly forming BFU-E (Fig. 2D and E).

To corroborate this finding, differentiating cells were collected to assess morphology or to be stained with antibodies against the myeloid (CD33, CD11b) and erythroid markers (CD71 and CD235a) to assess their expression by flow cytometry. This analysis revealed that *CEBPA* mutant clones gave rise to a lower percentage of myeloid cells composed mostly of immature cells CD33⁺CD11b⁺ (orange square) whilst the hiPSC control, MDS27-C22 and MDS27-C22 CRISPR control hiPSCs, were able to form mature myeloid CD11b⁺ cells (green rectangle) (Fig. 2F and G). The flow cytometry analysis confirmed that *CEBPA*^bZIP-fs mutant clones, MDS27-C22.7 and MDS27-C22.20, could generate significant erythroid colonies in methylcellulose culture with approximately half being erythroblast cells (CD71⁺CD235a⁺, blue square) (Fig. 2F and G). Erythroid-biased differentiation was also evident in the increase in erythroblast cells observed with diff-quick staining (Fig. 2H).

To assess whether the *CEBPA*^bZIP-fs mutation conferred an increase in self-renewal capacity, serial re-plating in methylcellulose was performed. Secondary re-plating resulted in the generation of BFU-E at a frequency of 264 ± 31 and 243 ± 36 colonies in MDS27-C22.7 and MDS27-C22.20 hiPSCs, respectively, whilst both controls and low-risk hiPSCs, had very limited re-plating capacity with a frequency of 2–3 colonies of CFU-GM and no BFUE colonies (Fig. 2I). In addition, a dramatic increase in the number of cells was observed in the *CEBPA*^bZIP-fs mutant lines from first re-plating to second re-plating. This result was in contrast with the other iPSC cell lines (hiPSC control, MDS27-C22 (Low risk) and MDS27-C22 CRISPR control) in which a decrease in the total number of cells was

found from the first to the second re-plating (Supplementary Fig. 4F). Cells from the *CEBPA*^bZIP-fs mutant clones (high risk) were able to proliferate in third-replating whereby the number of BFU-E had decreased to 149 ± 29 and 133 ± 18 colonies, respectively (Fig. 2I) and cells lost their proliferation capacity at the fourth re-plating.

To determine whether the *CEBPA*^bZIP-fs mutation alone was responsible for the gain in self-renewal capacity, a *CEBPA* mutation disrupting the bZIP domain was generated in BU3.10 wild-type iPSC cells (BU3.10-*C/EBPα*^mut C5 and C12) (Supplementary Fig. 5A, B). The BU3.10-*C/EBPα* mutant iPSC lines were able to differentiate to HPCs in equal numbers and proportions as their isogenic controls (Fig. 3A) but displayed a reduction in CFU formation when their differentiation potential was assessed by colony assays, specially CFU-G and BFU-E (Fig. 3B–D), which was also evident after the morphological examination (Fig. 3E). However, in contrast to MDS27-C22.7 and MDS27-C22.20 (high risk) iPSCs, BU3.10-*C/EBPα*^mut C5 and C12 did not form colonies after secondary and tertiary re-plating (Fig. 3D) and a decrease in total cell number were observed similar to isogenic controls (Fig. 3F). These data suggested that the disruption of the CEBPA bZIP domain affects myeloid differentiation but does not confer self-renewal capacity in a wild type environment. To determine whether the self-renewal capacity observed in our high-risk clones was indeed due to the acquisition of the *CEBPA* mutation we decided to rescue the *SRSF2* P95H and *RUNX1 Gly217fs* mutation in one of the MDS27 high-risk clones (MDS27-C22.7) by CRISPR (Supplementary Fig. 5C). Two clones (C5 and C8) showing a reverted genotype were taken for further analysis (MDS27-C22.7 S/*R*-reverted C5 and MDS27-C22.7 S/*R*-reverted C8). The number and percentage of HPCs derived from the two *SRSF2* and *RUNX1*-reverted iPSC clones were similar to those of the parental isogenic line (MDS27-C22.7 (high risk)) and MDS27-C22 (low risk) (Fig. 3G). The HPCs from the high-risk MDS27-C22.7- S/*R*-reverted iPSC produced a higher number of CFUs than isogenic controls (Fig. 3H–J), with a decrease in the percentage of erythroblast cells (Fig. 3L and M) and similar to high risk with a lack of mature myeloid cells. Moreover, the HPCs from the high-risk MDS27-C22.7- S/*R*-reverted iPSC had lost the capability of colony formation after replating (Fig. 3J and K).

Taken together, these results demonstrate that frameshift mutations leading to the disruption of the *CEBPA* bZIP domain affect myeloid differentiation as expected from previous studies[33], thus validating our model. Our results also show that introducing a *CEBPA* mutation disrupting the bZIP domain on the background of additional mutations (*RUNX1 and SRSF2*) but not by itself increased the self-renewal capacity of the committed progenitor cells and blocked

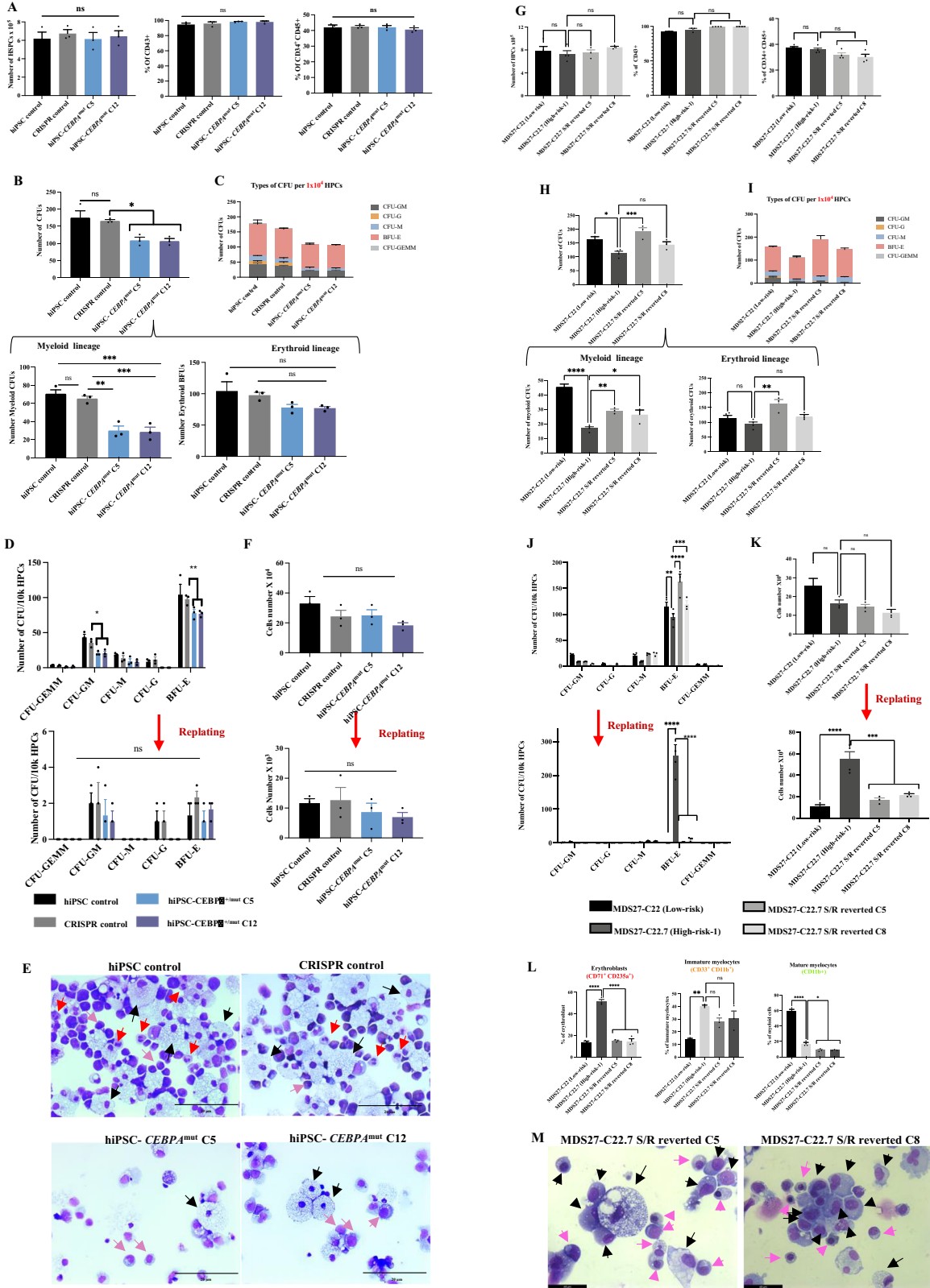

myeloid differentiation, which is the main characteristic of high-risk MDS and AML.

### Reduced myeloid differentiation capacity of high-risk MDS-iPSC containing C/EBPα bZIP-fs mutation

Colony assays revealed that the *C/EBPα* mutation disrupting the bZIP domain in combination with, *RUNX1* and *SRSF2* mutations affect the

myeloid differentiation capacity of HPCs (Fig. 2). Thus, to get a better understanding of how these mutations reduce myeloid differentiation potential, we performed myeloid lineage differentiation in liquid culture (Fig. 4A). The emergence of myeloid cells was assessed by flow cytometric analysis throughout the differentiation time course (day 0, 4 and 7) by looking at the percentage of pure granulocytic cells (CD11b⁺CD14⁻) and monocytic cells (CD11b⁻CD14⁺/

**Fig. 3 | C/EBPα bZIP frameshift mutation on its own does not confer self-renewal capacity to healthy HPCs or to diseased HPCs after correction of SRSF2 and RUNX1 mutations. A** Bar graph showing the number of HPCs and percentage of early hematopoietic population (CD43+) and late hematopoietic population (CD34+ CD45+) on day 14 of differentiation. Statistical results are presented as mean ± SEM. ns = no significant, One way ANOVA with Dunnett's multiple comparisons. N = 3 independent experiments. Source data are provided as a Source Data file. **B** Total number of CFUs from 10⁴ iPSC-HPC cells grown for 14 days in semisolid medium. Statistical results are presented as mean ± SEM. *p < 0.05; **p < 0.01, t-test. One-way ANOVA with Dunnett's multiple comparisons. N = 3 independent experiments. Exact p-values can be found in the Supplementary Data Table 5. Source data are provided as a Source Data file. **C** Number of each type of CFU after 14 days in semisolid medium. Mean and SEM of different lines are shown. N = 3 independent experiments. Source data are provided as a Source Data file. **D** Number of CFUs obtained from healthy iPSC, healthy iPSC CRISPR control, and healthy iPSC clones containing C/EBPα^bZIP-fs (C5 and C12) after first and second re-plating, each maintained for 14 days. Mean and SEM are shown. **p < 0.001, *p < 0.05 and (ns, no significant), Two-way ANOVA with Dunnett's multiple comparisons. N = 3 independent experiments. Exact p-values can be found in the Supplementary Data Table 5. Source data are provided as a Source Data file. **E** Diff-quick stained cytospins from colony assay showing erythroid cells (Pink arrows), granulocytes (Red arrows), and monocytes (black arrows). The pictures were taken with a Leica DM6000 at ×40, 20 μm scale bar. N = 3 independent experiments. **F** Bar graphs showing the number of cells obtained after the first and second replating. Statistical results are presented as mean ± SEM. ns, no significant, t test. One-way ANOVA with Dunnett's multiple comparisons. N = 3 independent experiments. Source data are provided as a Source Data file. **G** Bar graph showing the number of HPCs and percentage of early hematopoietic population (CD43+) and late hematopoietic population (CD34+ CD45+) on day 14 of differentiation derived from MDS27-C22.7 S/R reverted (C5 and C8). Statistical results are presented as mean ± SEM. ns = no significant, One-way ANOVA with Dunnett's multiple comparisons. N = 4 independent experiments. Data from C22 and C22.7 is taken from Fig. 2A. Source data are provided as a Source Data file. **H** Total number of CFUs from 10⁴ iPSC-HPC cells grown for 14 days in semi-solid medium. Statistical results are presented as mean ± SEM. *p < 0.05; **p < 0.01, One-way ANOVA with Dunnett's multiple comparisons. N = 3 independent experiments. Data from C22 and C22.7 is taken from Fig. 2B. Exact p-values can be found in the Supplementary Data Table 5. Source data are provided as a Source Data file. **I** Number of each type of CFUs after 14 days in semisolid medium. Mean and SEM of different lines are shown. N = 3 independent experiments. Data from C22 and C22.7 is taken from Fig. 2C. Source data are provided as a Source Data file. **J** Number of CFUs obtained from MDS27-C22.7 S/R reverted (C5 and C8) after first and second re-plating, each maintained for 14 days. Mean and SEM are shown. ****p < 0.0001, ***p < 0.001, **p < 0.01, Two-way ANOVA with Dunnett's multiple comparisons. N = 3 independent experiments. Data from C22 and C22.7 is taken from Fig. 2H. Exact p-values can be found in the Supplementary Data Table 5. Source data are provided as a Source Data file. **K** Number of the single cells obtained after each replating. Statistical results are presented as mean ± SEM. ns= no significant ****p < 0.0001, ***p < 0.001, ns, no significant. One-way ANOVA with Dunnett's multiple comparisons. N = 3 independent experiments. Data from C22 and C22.7 is taken from Supplementary Fig. 4C. Source data are provided as a Source Data file. **L** Fraction of erythroblasts (CD71+ CD235a+), immature myeloid (CD33+ CD11b+) and mature myeloid cells (CD11b+). Mean and SEM are shown. *p < 0.05; **p < 0.01; ****p < 0.0001; ns, no significant. One-way ANOVA with Dunnett's multiple comparisons. N = 3 independent experiments. Exact p-values can be found in the Supplementary Data Table 5. Source data are provided as a Source Data file. **M** Diff-quick stained cytospins from colony assay showing erythroid cells (Pink arrows), and monocytes (Black arrows). The pictures were taken with a Leica DM6000 at ×63, 40 μm scale bar. N = 3 independent experiments.

CD11b+CD14+). By day 7 all hiPSC lines without the C/EBPα mutation were capable of differentiation to granulocytes (around 20%). In contrast, the C/EBPα^mut hiPSC lines (high risk) had a significantly limited capacity to produce granulocytes and their differentiation seemed to be skewed to a higher proportion of monocytes in the culture (Fig. 4B, C).

Morphological examination of the myeloid cells by light microscopy was performed to evaluate distinct maturation features such as granule content, nuclear morphology, and cytoplasm/nucleus ratio. Interestingly, the committed eosinophils and neutrophils were identified within the myeloid population of hiPSC control, MDS27-C22 (low risk,) and MDS27-C22 CRISPR control hiPSCs, accompanied by appropriate segmented nuclear morphology and granules (Fig. 4D, green arrows). However, granulocytes were not detected in the myeloid cultures derived from high-risk hiPSCs, in agreement with the flow cytometry data (lack of the CD11b+CD14- population). In addition, morphological assessment of myeloid cells derived from high-risk hiPSCs showed that mutant clones differentiated into a heterogenous population that contained mainly erythrocytes (Fig. 4D, red arrows). Interestingly, the morphological results of myeloid cells of CEBPA WT lines (MDS27-C22 and MDS27-C22 CRISPR control) revealed dysplastic morphology in granulocytes such as hypo-segmentations, ringed nuclei, and hypo-granularity (Fig. 4E). These aberrant morphologies are consistent with those reported previously in MDS patients, including patient MDS27 (Fig. 4F)[34].

As expected by the change in the open reading frame, the C/EBPα bZIP mutation had significant effects on the functionality of the protein and consequently downregulation of the C/EBPα target genes CEBPA, SPI1 (PU.1) was observed in myeloid cells derived from the CEBPA^bZIP-fs iPSC lines (Fig. 4G). Moreover, LMO2 was shown to be upregulated in all MDS27 iPSC lines (low and high-risk) by day 7 of myeloid differentiation compared to the iPSC control line. As LMO2 has been shown to induce erythroid differentiation[35], the increase in LMO2 expression in our cultures could explain the increase of erythroid cells during the myeloid differentiation (Fig. 4G). The qRT-PCR data indicated that the CEBPA^bZIP-fs mutation leads to deregulation of the transcription factor expression signature resulting in the inhibition of myeloid differentiation and induction of transition to the erythroid lineage.

Overall, the CEBPA^bZIP-fs mutation in the context of RUNX1 (Gly217fs) and SRSF2 (Pro95His) mutations leads to an inhibition of granulocytic differentiation and a significant increase in the proportion of erythroid lineage cells. However, the original mutations RUNX1 and SRSF2 are responsible for the specific dysplastic morphology observed in myeloid cells.

## Dyserythropoiesis of high-risk MDS-iPSC containing the CEBPA^bZIP-fs mutation

Next, we wanted to determine if the formation of mature erythrocytes could be affected by the CEBPA^bZIP-fs mutation. We profiled erythroid differentiation kinetics by measuring the expression of the erythroid markers CD71 and CD235a: from erythroid progenitor cells (CD71+) to erythroblasts (CD71+ CD235a+) and mature erythrocytes (CD71- CD235a+). The kinetics of erythroid differentiation in hiPSC control and MDS27-C22 and MDS27-C22-CRISPR control iPSC lines, showing similar percentages of erythrocyte progenitors, erythroblasts, and mature erythrocytes at any given time point (Fig. 5A, and Supplementary Fig. 6A). In contrast, the CEBPA^bZIP-fs high-risk hiPSC lines displayed a greater percentage of erythroblasts (CD71+ CD235+) at day 4 and 7 of differentiation compared to any of the control cells (Fig. 5A and Supplementary Fig. 6A) and displayed a higher CD71 expression at the end of the erythroid differentiation (Fig. 5B). Interestingly, this result indicates that disruption of the bZIP domain in cells harbouring RUNX1 and SRSF2 mutations impaired proper terminal erythroid differentiation as well.

Morphological analysis showed a higher percentage of mature erythroid cells at day 11 and 14 compared to isogenic controls (Supplementary Fig. 6B, C). Aberrant morphologies such as giant bi-

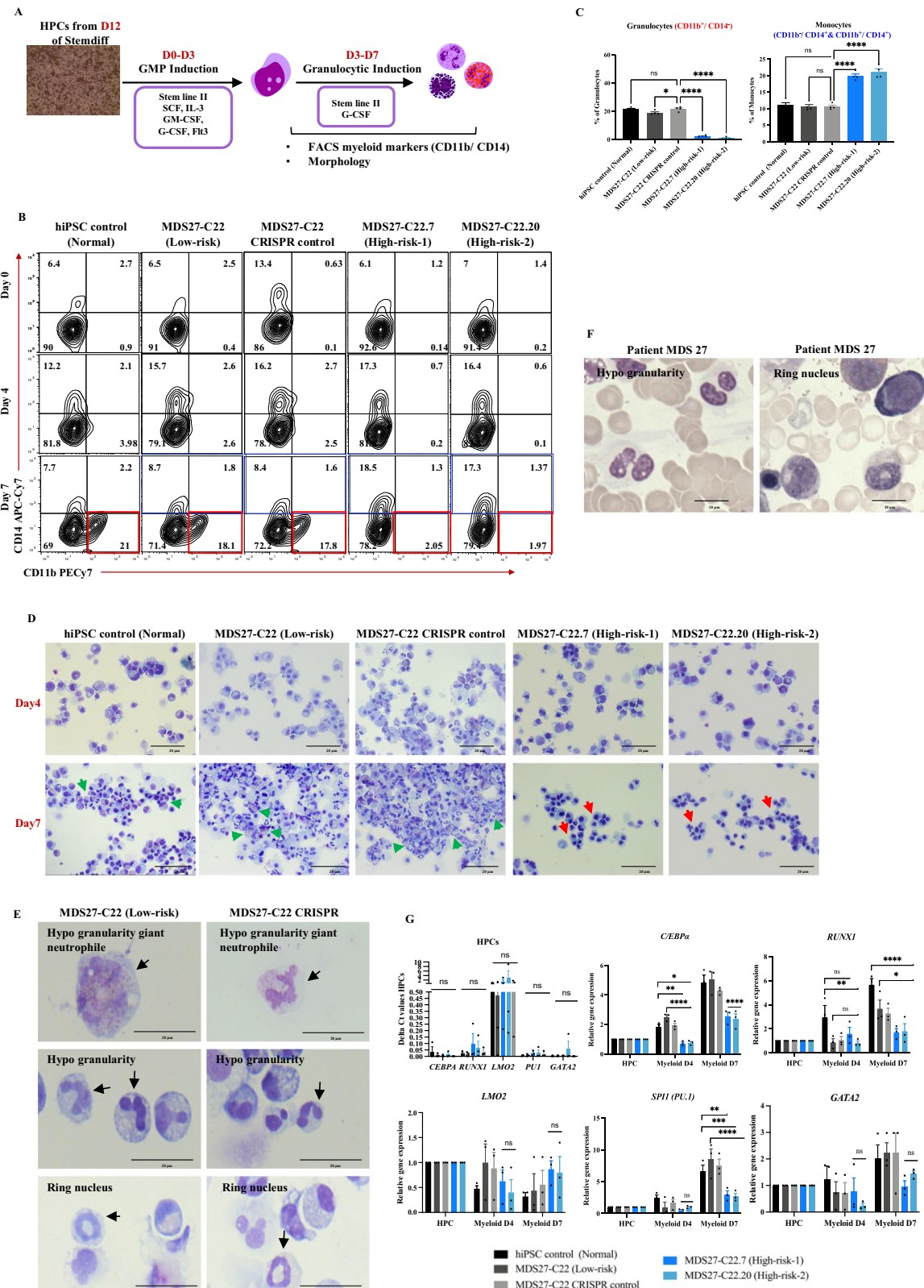

nucleated and mitotic erythroblasts were also observed in *CEBPA*^bZIP-fs MDS27-iPSC lines (Fig. 5C). Moreover, the percentage of erythroid cells with aberrant morphology of high-risk iPSC compared to control hiPSC was significantly higher during differentiation (Fig. 5D). However, the percentage of aberrant cells in the high-risk hiPSC lines

compared with MDS27-C22 low risk and MDS27-C22-CRISPR control hiPSCs was slightly increased but not statistically significant (Fig. 5D).

Taken together, cells harbouring mutations in the three genes (*SRSF2, RUNX1*, and *CEBPA*) display altered kinetics of terminal erythroid maturation with evident signs of dysplastic erythroid differentiation.

**Fig. 4 | High-risk MDS-iPSC containing a *C/EBPα* bZIP frameshift mutation exhibits a block in the granulocytic differentiation. A** Schematic representation of the myeloid differentiation protocol. Artwork generated with powerpoint Bundle-Biology. **B** Representative images of flow cytometric analysis of CD11b+ CD14− (granulocytes) and CD11b+ CD14+ (Monocytes) for the indicated cell types on day 0, day 4, and day 7 of differentiation. $N = 4$ independent experiments. Source data are provided as a Source Data file. **C** Percentages of CD11b+ CD14− (granulocytes) and CD11b+ CD14+ (Monocytes). Mean and SEM are shown ****$p < 0.0001$, *$p < 0.05$, and (ns, no significant). One-way ANOVA with Dunnett's multiple comparisons. $N = 4$ independent experiments. Exact p-values can be found in the Supplementary Data Table 5. Source data are provided as a Source Data file. **D** Diff-quick stained cytospins showing the presence of granulocytic cells (green arrows) and erythroblasts (red arrows). The pictures were taken with a Leica DM6000 microscope at ×40 magnification, 20 µm scale bar. $N = 4$ independent experiments. **E** Diff-quick stained cytospin showing aberrant morphology of myeloid cells (black arrows) in MDS-C22 (Low-risk) and MDS27 C22 CRISPR control. The pictures were taken with a Leica DM6000 microscope at ×100 magnification, 20 µm scale bar. $N = 4$ independent experiments. **F** May–Grunwald Giemsa stained bone marrow smears from MDS27 patient at disease progression showing aberrant myeloid cells (from Supplementary Fig. 1A). The pictures were taken with a Leica DM6000 at ×63 magnification. **G** RNA expression of *C/EBPα, RUNX1, SPI1 (PU.1), GATA2* and *LMO2* measured by qRT-PCR, normalized against the *GAPDH* housekeeping gene and expressed relative to HPCs expression. $N = 3$ independent experiments. Mean and SEM are shown ****$p < 0.0001$, ***$p < 0.001$, **$p < 0.01$, *$p < 0.05$, and (ns, no significant), Two-way ANOVA with Tukey's correction. $N = 3$ independent experiments. Exact *p*-values can be found in the Supplementary Data Table 5. Source data are provided as a Source Data file.

## High-risk MDS-iPSC containing CEBPA^bZIP-fs are resistant to 5-Azacitidine treatment

Our work so far showed that the isogenic iPSC lines generated could replicate the disease phenotype with dysplasia observed in both erythroid and myeloid lineages upon HPC differentiation. The MDS27 patient relapsed after DA treatment and did not respond to 5-Azacytidine (5-AzaC) showing an increase in the percentage of blasts and immature erythroid cells and the presence of bi-nucleated dysplastic erythroid cells (Supplementary Fig. 1A–D). We were interested to determine whether our in vitro system could recapitulate the therapy resistance observed in the patient. To determine the effects of cytarabine (Ara-C) in our cells we performed similar treatments as previously described[10]. Our data showed that both low-risk and high-risk iPSC-derived HPCs were partially responsive to Ara-C treatment, with a 50% reduction in proliferation rates (Supplementary Fig. 7A). Additionally, as MDS27 patient did not respond to the conventional demethylating agents 5-Azacytidine (5-AzaC)[30], we cultured iPSC-derived HPCs in methylcellulose in the presence or absence of two different concentrations of 5-AzaC. Treatment of 5-AzaC did not have any effect on colony growth in healthy iPSC or low-risk iPSC clones (Fig. 5E). In contrast, 5-AzaC treatment had a profound effect in cells derived from the high-risk MDS, with a massive expansion on colony number, specifically in BFU-E (Fig. 5E and Supplementary Fig. 7B, C) which was also dose-dependent. To determine whether the expansion of the BFU-E colonies was an indication of healthy recovery, we performed a morphological analysis of the cells grown in colony assays. This analysis revealed the presence of dyserythropoietic in the high-risk iPCS after 5-AzaC treatment, characterized by binucleated (pink and green arrows, Fig. 5F), multinucleate (blue arrow, Fig. 5F) and budding nuclei (red arrows, Fig. 5F) erythroid cells. Similar erythroid dysplasia features were observed in the MDS27 patient after 5-AzaC treatment (Fig. 5G), thus confirming that cells did not respond to 5-AzaC treatment. We then sought to determine the clonogenic capacity of the cells from MDS27 patients in vitro. Primary CD34+ cells from MDS27 patients were cultured in methylcellulose under normoxia which revealed a lack of clonogenic capacity in the high-risk sample. Although not performed with low-risk samples because of the lack of primary material, under hypoxia (5% $O_2$), CD34+ cells from MDS27 high-risk were capable of forming colonies and the number of colonies was slightly increased after 5-AzaC treatment, contrary to the low-risk cells (Fig. 5H). Flow cytometry analysis of the cells revealed a high proportion of early progenitor cells (CD71+ CD235a−) in the high-risk sample with and without 5-AzaC treatment (Supplementary Fig. 7D). Additionally, cells from high-risk were able to form colonies after the second replating even after treatment with 5-AzaC (Fig. 5H). After the second replating, a difference in the colony size was detected in those receiving 5-AzaC, which showed increased colony size compared to untreated cells, indicative of higher proliferation after treatment (Supplementary Fig. 7E).

Taken together, our work shows that the isogenic iPSC lines generated recapitulate the disease phenotype and the therapy resistance observed in the MDS27 patient.

## The acquisition of the CEBPA mutation leads to similar changes in the chromatin landscape in iPSC-derived cells and in patients

Our data show that the acquisition of a CEBPA mutation leads to a drastic change in the phenotype of cells concomitant with the acquisition of an increased risk of developing into AML. The chromatin landscape is altered in leukaemic cells and each leukaemic sub-type, including CEBPA mutations, displays its own specific chromatin pattern with specific transcription factor binding motif signatures[36] demonstrating that each mutation deregulates the gene regulatory network of hematopoietic progenitor cells in a similar way. To test whether the phenotypic alteration seen in iPSC-derived cells and in patients after the acquisition of a CEBPA mutation were based on similar chromatin changes, we performed the assay for transposase-accessible chromatin with high-throughput sequencing (ATAC-seq). Figure 6A shows the analysis of ATAC-seq data from sorted CD34+CD45+ HPCs derived from low-risk (MDS27-C22) and high-risk (MDS27-C22.7) iPSCs. To do this, we calculated the fold difference in peak height between low and high-risk samples and peaks which had a difference of more than 1.5 fold were deemed to be differentially accessible. We then conducted a TF binding motif enrichment analysis and identified a set of motifs that were enriched in these differentially accessible sites (Fig. 6B, C). The presence of these motifs was plotted alongside the ATAC-Seq data in Fig. 6A. The analysis shows that 1691 sites were gained, and 1576 sites were losing accessibility after CEBPA was mutated. The effect of the *CEBPA* mutation was more pronounced in distal elements (observed 79.3% and 75.9% differential peaks, in MDS-High and Low risk, respectively, compared to 59.6% in all peaks; Chi-square *p*-value 5.4E−14) than in promoter regions (Fig. 6D). The analysis of the motif content showed a loss of a myeloid signature dominated by PU.1 and C/EBP motifs consistent with the *CEBPA* mutation disrupting the DNA-binding domain and thus preventing the interaction of *CEBPA* with its genomic targets[37,38] (Fig. 6B–E). We also observed an increase in AP-1, RUNX, and GATA motif enrichment after CEBPA was mutated, indicating a shift into a more immature state[39,40] (Fig. 6B, C). The chromatin landscape in purified CD34+ cells from patients shifted in a similar way after the acquisition of *C/EBPα* mutation in high-risk patients during disease progression. 8290 regions gained accessibility, and 7795 regions lost accessibility (>1.5-fold change) in high-risk cells (Fig. 6F). High-risk specific open chromatin regions again showed enrichment of GATA and AP1 motifs and changes in the distribution of PU.1/ETS and RUNX motifs (Fig. 6F–H). In summary, our data shows shared alterations in the chromatin pattern and motif composition during disease progression in iPSC-derived and patient cells.

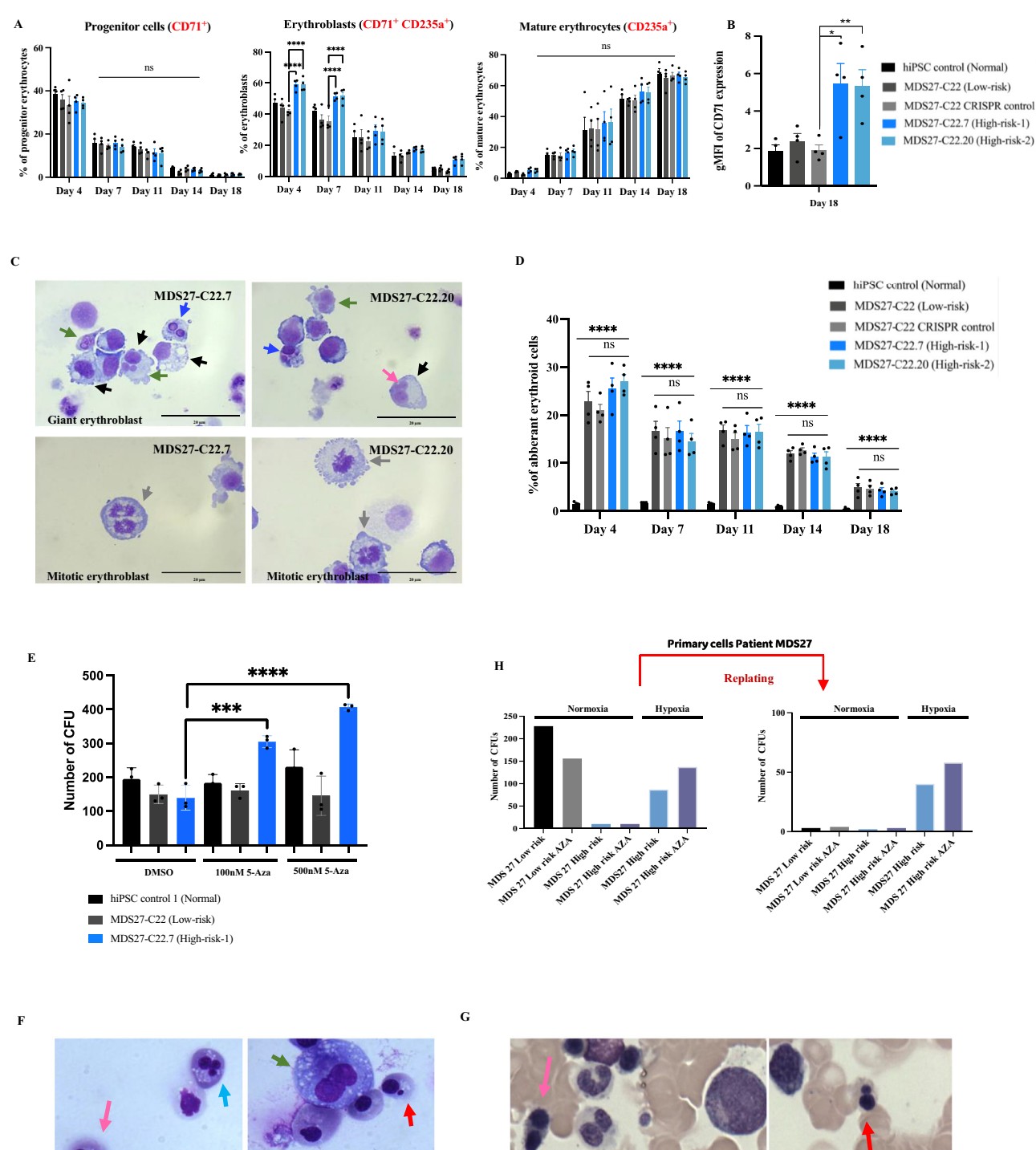

## Changes in cellular composition during disease progression

To determine whether changes in the chromatin landscape associated with the acquisition of the *CEBPA* mutation affected the transcriptional programme, we performed single-cell transcriptome analysis in Day 12 HPCs−iPSCs derived from the BU3.10 iPSC control line, MDS27-C22 (low risk) and MDS27-C22.7 (high risk). We also performed single-cell transcriptomics in HPCs 4 days after differentiation towards the erythroid lineage from BU3.10 iPSC control line and low risk to define transcriptional changes during differentiation (Supplementary Fig. 8A).

Unsupervised clustering revealed 13 clusters (Fig. 7A and B). Clear changes in cluster representation were observed between wild-type, low-risk, and high-risk HPCs (Fig. 7A). Also, changes in the cell cycle

status were evident between the samples, with a lower proliferation rate in the high-risk sample compared to low-risk, indicative of a more quiescent stage (Fig. 7B and Supplementary Fig. 8B) as observed before[41–43]. Based on gene expression markers, wild-type HPCs could be found mainly in three clusters: 2, 9, and 4 which were identified as early stem/progenitors (HPCI) (*HMG high, GYPB*), early erythroid/ megakaryocyte progenitors (MEP) (GP9, GP1BB, Rap1b, PF4) and myeloid progenitor cells (GMP/monocytic) (*MPO, AZU1*), respectively (Fig. 7C, Supplementary Fig. 8C, D)[44,45]. A prominent early progenitor cluster was also present in the MDS low-risk sample (HPCs II, cluster 1), expressing similar markers as the wild-type HPC cluster (*MYB, NPM1, GATA2, HMGA* and *GYPB*), but this cluster was really underrepresented

**Fig. 5 | Erythroid differentiation of high-risk MDS-iPSC containing a *C/EBPα* bZIP frameshift mutation reveals erythrodysplasia even after 5-Aza treatment.** **A** Percentage of early erythroid progenitor cells (CD71+), erythroblasts (CD71$^+$ CD235a$^+$) and mature erythrocytes (CD235a$^+$) during several time points of erythroid differentiation. Mean and SEM are shown. **** <0.0001, **$p$ < 0.001, *$p$ < 0.05 and (ns, no significant), Two-way ANOVA with Dunnett's correction. $N$ = 4 independent experiments. Exact $p$-values can be found in the Supplementary Data Table 5. Source data are provided as a Source Data file. **B** Bar graph showing CD71 geometric mean (gMFI) of each cell line at day 18 of erythroid differentiation. Mean and SEM are shown. * $p$ < 0.05, **$p$ < 0.001. One-way ANOVA with Dunnett's multiple comparisons. $N$ = 4 independent experiments. Exact $p$-values can be found in the Supplementary Data Table 5. Source data are provided as a Source Data file. **C** Diff-quick stained cytospins showing erythroid dysplasia; Giant erythroblast (black arrow), Mitotic erythroblast (Grey arrows) in MDS27-C22.7 (High-risk-1) and MDS27-C22.20 (High-risk-2), Multinucleated erythroblast (green arrows), bi-nucleated erythroblast (blue arrows) and nuclear bridge (pink arrows). The pictures were taken with a Leica DM6000 microscope at ×100 magnification, 20 μm scale bar. $N$ = 4 independent experiments. **D** Bar graph represents the percentage of the aberrant cells of MDS27 clones relative to the aberrant morphology of hiPSC control. Results are presented as mean ± SEM and ****$p$ > 0.0001, Two-way ANOVA

with Dunnett's multiple comparisons. $N$ = 4 independent experiments. Source data are provided as a Source Data file. **E** Bar graph represents the number of CFUs obtained from control, MDS27-C22 (Low-risk), and MDS27-C22.7 (high-risk) in the presence of different concentrations of 5-Aza or vehicle control (DMSO). Mean and SEM are shown. ****$p$ < 0.0001, **$p$ < 0.001, *$p$ < 0.05 and (ns, no significant), Two-way ANOVA with Dunnett's multiple comparisons. $N$ = 3 independent experiments. Exact $p$-values can be found in the Supplementary Data Table 5. Source data are provided as a Source Data file. **F** Diff-quick stained cytospins from colony assay showing aberrant morphology in MDS27- C22.7 (high risk) after 5-Aza treatment; nuclear budding early orthochromatic erythroblasts (red arrow), bi-nucleated pronormoblast (green arrow), multinucleated basophilic erythroblast (blue arrows) and binucleated orthochromatic (nuclear bridge) (pink arrows). The pictures were taken with a Leica DM6000 microscope at ×63 magnification, 40 μm scale bar. $N$ = 3 independent experiments. Source data are provided as a Source Data file. **G** May−Grunwald Giemsa stained bone marrow smears from MDS27 patient after receiving 3 rounds of Aza treatment showing aberrant erythroid cells. The pictures were taken with a Leica DM6000 at ×63 magnification. **H** Number of CFUs obtained from CD34+ sorted cells from MDS27 patient samples in normoxia (before and after disease progression) and hypoxia (after disease progression). Cell were maintained for each replating. Source data are provided as a Source Data file.

in the high-risk MDS comprising of 6% of the total cell population as compared to 60% in low-risk MDS population (Fig. 7C, D and Supplementary Fig. 8D). MDS clusters 3, 6, 7, 8, 11 and 12 were distinct from wild type. A gradual loss of cell identity from low risk to high risk could be observed representing unique MDS/AML cell populations, with 30% of the cells in low risk forming part of the unique MDS/AML signature compared to 90% of the cells in high-risk. Additionally, changes in the size of these clusters were noticeable, indicating clonal dynamics during disease progression. Hence, some of the clusters were greatly expanded from low to high-risk MDS (red squares, Fig. 7D), whereas others were reduced from low to high-risk MDS or did not show a clear difference in composition (blue squares and green squares, respectively, Fig. 7D). By performing GSEA with available gene signatures from HSC[46,47], LSC[46,48], and AML harbouring a *CEBPA* mutation[49], we observed an HSC, and possibly LSC-like signature in MDS clusters 6 and 7, with sets of genes that have been found to be downregulated in LSC and HSC also being depleted in the clusters, while several HSC/LSC signatures were enriched (Fig. 7E).

Among the most highly expressed genes in the MDS clusters were genes associated with AML as for example, *CD99*[50,51], *EpCAM*, related to enhanced cell survival, diminished mitochondrial metabolism, ribosome biogenesis, and differentiation capacity and an activated transcriptomic signature associated with AML[52] (Fig. 7F and Supplementary Fig. 8D), *ID1*, which is expressed by AML cells and has been shown to increase leukaemic proliferation and promote tumour progression by impairing myeloid differentiation[53,54], *ID3*, which promotes erythroid differentiation[55,56] (Supplementary Fig. 8D). Other leukaemia associated genes were expressed by cells within clusters that had expanded during disease progression, for example *HOXB-AS3* (Fig. 7F), a gene encoding a long non-coding RNA shown to be an adverse prognostic marker for AML and MDS[57], *CD93* (Fig. 7F), a marker associated with proliferative leukaemic stem cells[58] and genes associated with angiogenesis and therapy resistance (*RAMP2*, *CALCRL*)[59,60], angiogenesis and poor prognosis in MDS (*FLT1, KDR*)[61,62] (Supplementary Fig. 8D).

To gain a better insight into the changes occurring during disease progression, we performed a differential gene expression analysis comparing cells from low and high-risk clusters. KEGG pathways analysis of the approximately 500 differentially expressed genes (Supplementary Data Table 2) indicated that low-risk-specific genes were related to a myeloid-differentiation type signature whilst genes expressed in high-risk MDS cells encoded for cell signalling components, such as Wnt signalling and Hippo pathway, both associated with AML (Supplementary Fig. 8E, F). Additionally, a clear difference in the

expression of important transcription factors regulating hematopoiesis was noticeable between low-risk and high-risk HSPCs samples, with high-risk cells showing reduced expression of *RUNX1* and *SPI1* (*PU.1*) and gaining expression of the AP1 factors *FOS* and *JUN* (Fig. 7G, Supplementary Fig. 8D). We additionally performed a differential gene expression together with a GO-term analysis comparing up and down-regulated genes between clusters 1, 2, 3, 7 and 8 of HPCs iPSC derived cells (Supplementary Fig. 8E, Supplementary Data 3. The volcano plots below show the number of genes upregulated and downregulated in the high-risk cells. This analysis revealed the upregulation of a number of MDS/AML-relevant genes such as that of Ly6E expression which has been associated with poor survival outcomes in multiple malignancies[63], in high-risk samples across all clusters. In addition, the data revealed the upregulation of multiple AP-1 family members in high-risk compared to low-risk in clusters 1, 3, and 8, which supports the increased enrichment of AP1 motifs in our ATACseq data from this population (Fig. 6 A-C; F−H). Note that we have shown AP-1 to be an essential factor for the growth of *CEBPA* mutant AML cells[64]. In addition, we see a downregulation of cell cycle regulators and S-phase-specifically expressed genes (such as Ki-67 and histone genes) which tally with the fact that high-risk HPSCs have a much-diminished proportion of S-phase cells (Fig. 7B). Note also the up-regulation of SOX4 in cluster 3 which has been shown to be a C/EBPα-repressed gene and is important for self-renewal of AML cells with a *CEBPA* mutation[65]. In summary, this analysis is entirely consistent with our iPSC-derived cells displaying the molecular features of *CEBPA*-mutant cells (Supplementary Fig. 8G and Supplementary Data 3).

Once HPC were subject to erythroid differentiation, cells from control and MDS low risk were both found in clusters that represented different stages of erythroid (clusters 0 and 5) and myeloid (clusters 4 and 10) differentiation[45]. A different representation of myeloid and erythroid cells between control and MDS low-risk samples was evident, with less myeloid and more erythroid cells in MDS low-risk compared to control (Fig. 7A and D). By inspecting the most highly expressed genes within each cluster, we found clear differences in gene expression between the control and MDS low-risk samples. For example, *ID1, APOE*, and *CD14* were expressed by MDS low-risk cells from the myeloid cluster whereas *MPO, PRTN3*, and *AZU1* genes were expressed by wild-type BU3.10 cells, revealing changes in myeloid composition between control and MDS low-risk cells (Fig. 7H, Supplementary Fig. 8D and Supplementary Data Table 4).

To gain more insight into the position of the aberrant cell population within the differentiation trajectory, we performed a pseudotime analysis (Fig. 7I). The control cells showed a clear trajectory, with

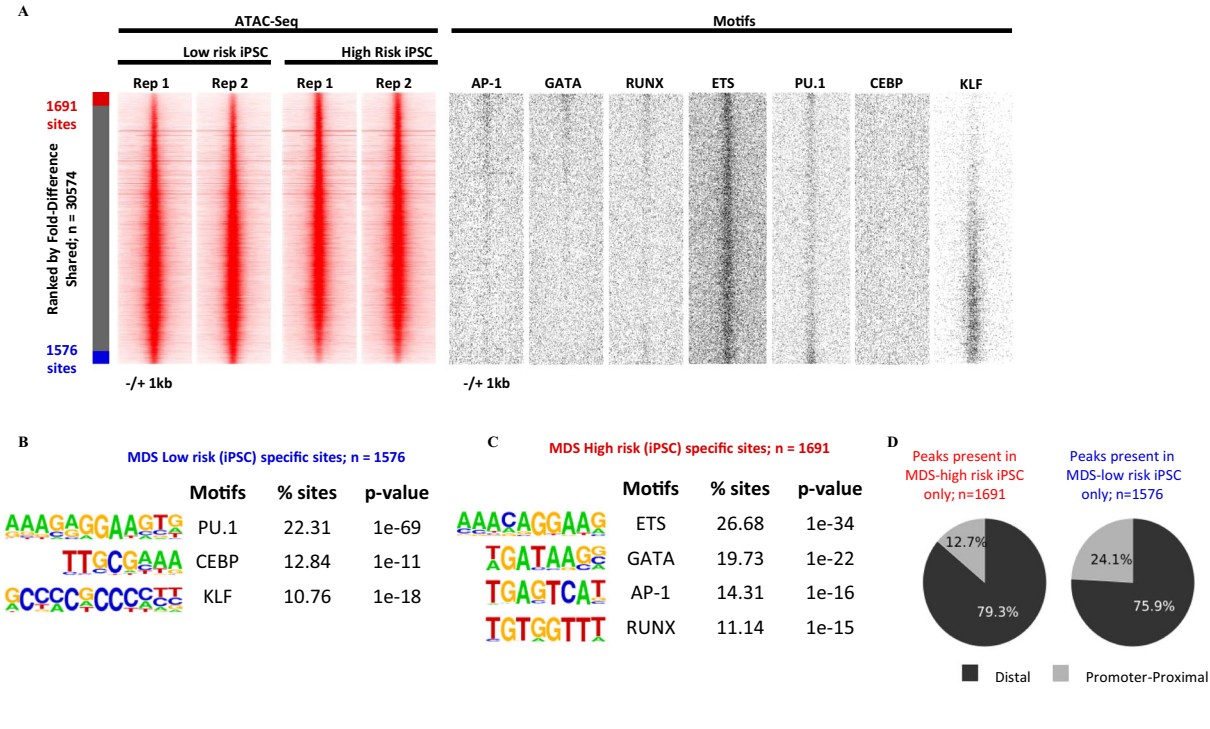

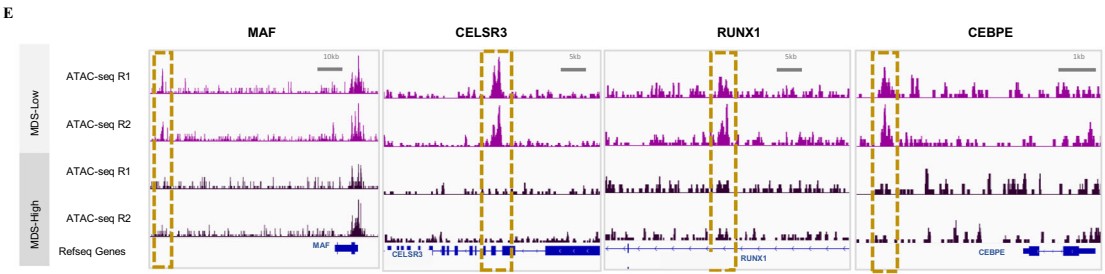

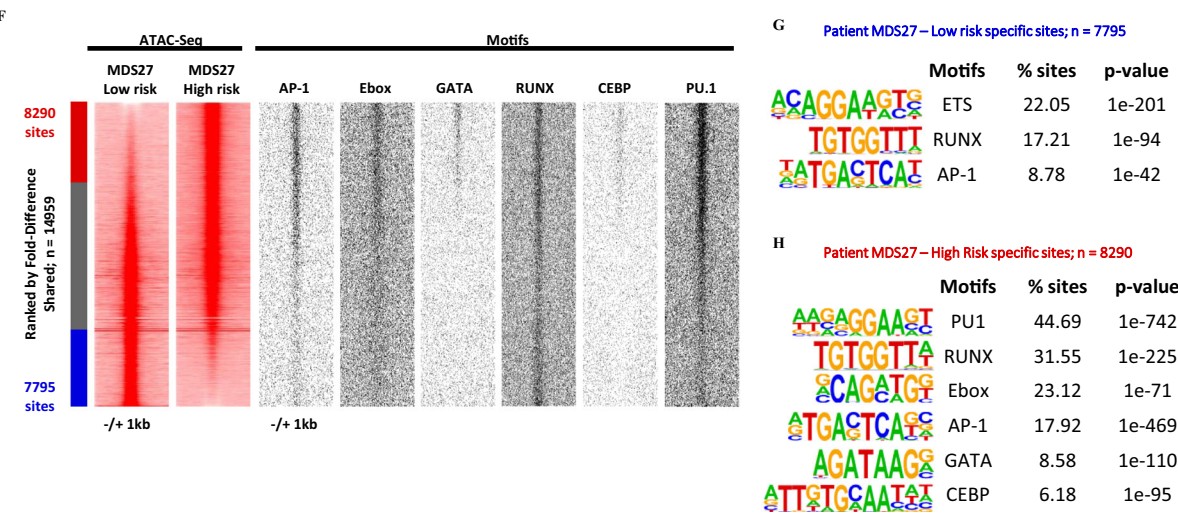

myeloid progenitors (left) and erythroid progenitors (right) branching off from the more immature HPCs (purple/blue). In clear contrast, MDS cells showed an additional trajectory branching off from the immature HPCs, with branch points corresponding to the myeloid lineage and a disorganized erythroid lineage (Fig. 7I). These results are consistent with our differentiation results, and an increase in erythroid differentiation at the expense of the myeloid differentiation.

In summary, the single cell transcriptome analysis is consistent with the notion of MDS disease arising from an early stem/progenitor cell, leading to dysregulation of genes involved in stem/progenitor cell development which is translated into a different clonal composition, loss of cell identity and expansion of specific clones during disease progression. Our isogenic model system of disease progression shows changes in cellular composition and differentiation hierarchies when

**Fig. 6 | Changes in distal cis-regulatory elements in HPCs with *C/EBPα* mutation.** **A** Profiles of the ATAC-seq signals within each 2000-bp window centered on each peak for Day14 CD34⁺CD45⁺ HPC-iPSC derived from Low-risk MDS (C22) and from high-risk MDS (C22.7) (*N* = 2 independent experiments). Peaks are shown in order of decreasing log2 fold-difference between Low and High-risk samples (see the "Methods" section). Positions of transcription factor binding motifs are plotted alongside. **B** Motif enrichment analysis (Homer de novo motifs) for peaks only present in low-risk HPCs compared to unchanged peaks. **C** Motif enrichment analysis (Homer de novo motifs) for peaks only present in high-risk HPCs compared to unchanged peaks. **D** Pie chart showing the percentage of peaks at promoters and distal regulatory elements from peaks present in high-risk MDS HPCs only (left) or in low-risk MDS HPCs (right). **E** ATAC-seq UCSC genome browser screenshot depicting accessible chromatin sites being differentially regulated between the low- and high-risk HPCs. Red squares show differential ATAC-seq peaks between both conditions. *Y*-axis is set at 70 RPKM. **F** Profiles of the ATAC-seq signals within each 2000-bp window centered on each peak for CD34+ sorted cells from MDS27 patient samples before and after disease progression. Peaks are shown in order of decreasing log2-fold-difference between Low and High-risk samples (see the "Methods" section). Positions of transcription factor binding motifs are plotted alongside. **G** Motif enrichment analysis (Homer de novo motifs) for peaks only present in low-risk CD34⁺ cells compared to unchanged peaks. **H** Motif enrichment analysis (Homer de novo motifs) for peaks only present in high-risk CD34⁺ cells compared to unchanged peaks.

comparing low-risk and high-risk isogenic iPSC models, which reflect the changes in clonal composition and disease progression observed in patients.

## Discussion

The extensive heterogeneity of MDS is caused by a combination of different mutations acquired by blood progenitor cells during ageing, leading to clonal evolution and disease progression. In patients, mutations in components of the spliceosome, and epigenetic modifiers can act as driver mutations as well and when occurring together generate a combination that is causal for MDS development[5]. Through PCR and sequencing-based studies, mutations in *CEBPA* have also been shown to associate with disease progression and AML[66–68]. However, direct proof for this idea has so far been lacking as it requires to systematically test the contribution of different mutations to MDS pathogenesis in an isogenic background. The generation of iPSC from an AML patient containing a *CEBPA* mutation (H24fs*84) in combination with *TET2, IDH2, ASXL1,* and *SRSF2* mutations has been reported[10]. However, iPSC generation from an AML patient containing *CEBPA* mutation F31fs*130 as well as mutations in *TET2, STAG2, SRSF2,* and *CSF3*R was not successful[69]. In our hands, we were also unable to generate iPSC from the patient after disease progression harbouring a mutation disrupting the CEBPA bZIP domain (Gly257fs), highlighting that the complex genetic background of high-risk MDS and AML patients dictates the ability of generating iPSC.

In the work described here, we applied somatic reprogramming to iPSC and CRISPR technology to generate human isogenic MDS cell lines from a representative patient before and after disease progression. Using CRISPR/Cas9 to disrupt the CEBPA bZIP domain in wt and mutant background, and to revert the *SRSF2* P95H and *RUNX1* G217P fs mutations, we were able to dissect the contribution of different mutations to the disease phenotype. These experiments identify the disruption of the CEBPA bZIP domain as causative for disease progression in this mutational background, affecting cell fate decisions in the evolution from low risk to high risk. In addition, we show (i) that the heterozygous disruption of the bZIP domain in a wt background leads to an impediment of granulocyte differentiation, but does not confer self-renewal capacity (ii) that the primary mutational pattern (*SRFS2* and *RUNX1*) leads to dysplasia of erythroid and myeloid lineages, and (iii) that the additional introduction of the *CEBPA* mutation causing the disruption of the bZIP domain leads to a combination of both, and promotes self-renewal capacity of HPCs, thus mimicking the increased risk of AML development seen in the patient.

Our findings are broadly consistent with the pathology data reported for most MDS patients, including the presence of a low number of CFU progenitor cells[8,30,70–72]. Following aberrant blood cell development of cells with different mutational backgrounds in liquid culture revealed the development of mature cells with aberrant morphology, such as nuclear bridges and multinuclear content which are common dysplastic features of MDS patients[34,73]. Moreover, we were also able to recapitulate the therapy resistance to 5-AzaC as seen in the patient. Our MDS27-iPSCs model is, therefore, a bonafide in vitro

model for early MDS disease pathology showing erythroid and myeloid dysplasia similar to the anaemia and cytopenia of the patient at the time of diagnosis and after disease progression.

Our work also shines an interesting light on the role of mutant C/EBPα and its interaction with other mutations in MDS pathology when all mutations are present as heterozygotes. Similar percentages of HPCs were produced by the isogenic low- and high-risk clones. In agreement with the essential role of C/EBPα in the formation of granulocyte progenitors[74,75], neither CFU-G nor mature granulocytes were detected in colony assays or liquid cultures, respectively. Nonetheless, the high-risk iPSC had the potential of forming CFU in methylcellulose, in much higher numbers than those reported previously[8]. It has been reported that *CEBPA* null foetal liver progenitors are hyperproliferative, fail myeloid differentiation, and show increased self-renewal potential in vitro and in vivo[75,76]. In addition, mouse bone marrow mononuclear cells transduced with a C-terminal mutated C/EBPα (*CEBPA^Cmut*) increased the self-renewal of the CFU forming colonies after six rounds of re-plating[66]. Moreover, 5-FU-treated mice transduced with these *CEBPA^Cmut* bone marrow cells developed AML when transplanted into recipient mice[66]. However, our data suggest that in a wild-type background, a frameshift mutation in the mid-region of *CEBPA* did not confer self-renewal capacity. Thus, the increase in self-renewal capacity after the acquisition of *CEBPA* mutation had to be due to a synergistic effect with other mutations present within the clone. It has been previously reported that RUNX1 mutations enhance self-renewal and impede granulocytic differentiation[77]. Indeed, reverting the SRSF2 and RUNX1 mutations in the high-risk iPSCs (harbouring *CEBPA* bZIP mutation), abolished enhanced self-renewal capacity, demonstrating that (i) the combination of mutations dictates the phenotype, (ii) the addition of the *CEBPA* mutation affecting the bZIP domain to RUNX1/SRSF2mutations further imposes a granulocytic differentiation impediment, (iii) promotes an increased proportion of erythroid progenitors at the expense of myeloid progenitors and (iv) confers self-renewal capacity to these cells.

In spite of the presence of a wild-type allele, our data show that the presence of a C/EBPα protein with a frameshift mutation in the mid-region of the protein affects the chromatin landscape through dysregulation of accessible chromatin regions in distal cis-regulatory elements, suggesting a dominant deregulatory activity of this protein, probably by interfering with dimerization. Lost chromatin sites were enriched in PU.1, KLF, and CEBPA motifs, suggesting that C/EBPα binding was lost which agrees with the change in the open reading frame of the mutated protein. Interestingly, both iPSC and patient HPCs were still capable of producing CD11b+ cells, supporting previous reports of PU.1 independent macrophages in mouse embryos which go on to generate tissue macrophages[78–80]. High-risk HPCs gained ATAC-seq peaks enriched in ETS and RUNX motifs, which have been reported to regulate the leukaemic signature in AML patients with CEBPA biallelic mutations (*CEBPA^N/O*)[64], whilst enrichment of GATA motifs indicates that the HPCs from high-risk cells are less differentiated than those from low risk, which is consistent with a defect in myeloid differentiation. The same motif families were enriched in CD34+ cells from iPSCs and the

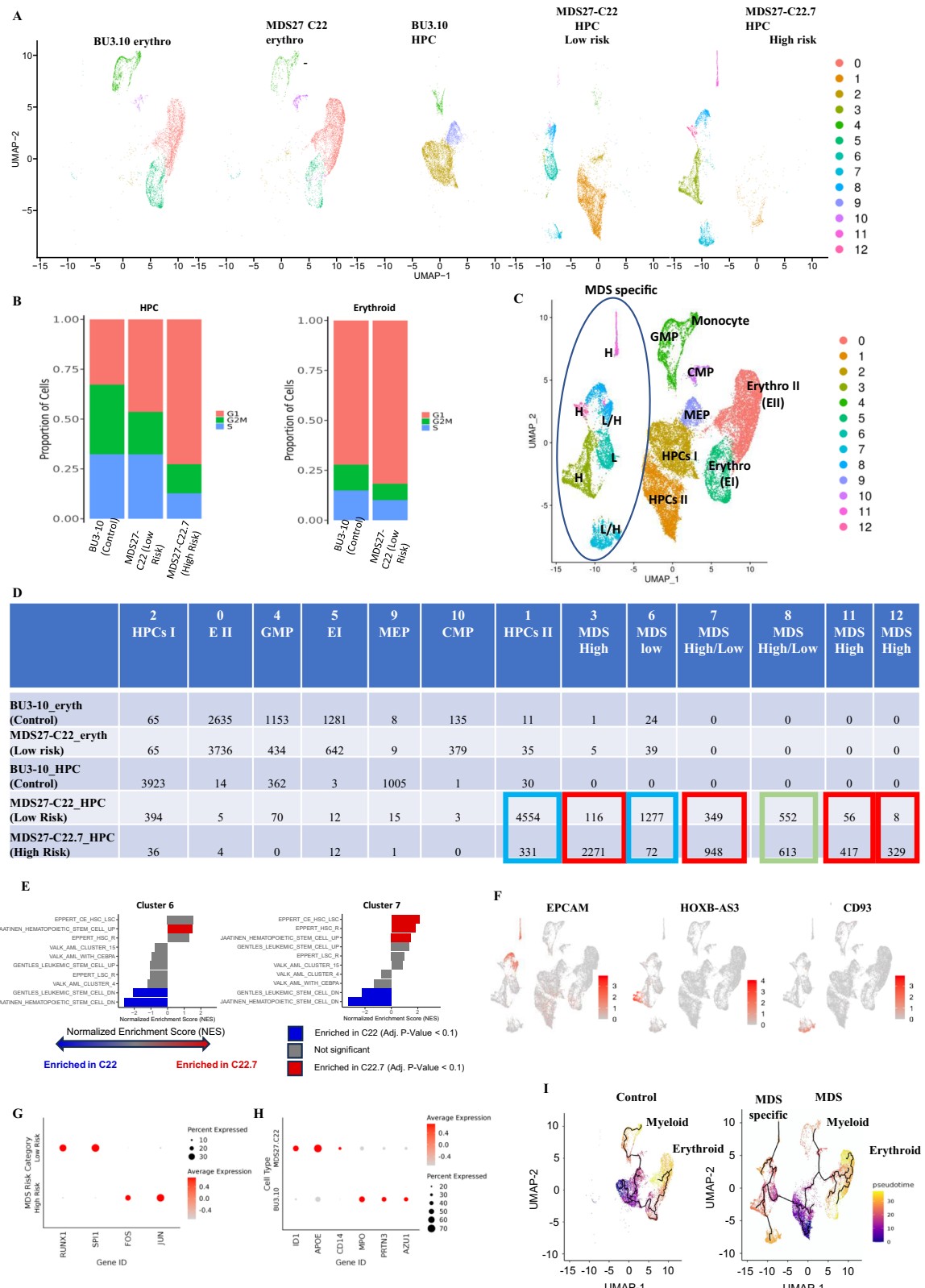

patient following disease progression, suggesting a shared pattern of chromatin reorganization upon acquisition of the CEBPa mutation and thus, reinforcing the validity of our in vitro system.

By single-cell mutational studies, it has been reported that changes in clonal architecture occur during MDS disease progression[81,82] and single-cell transcriptomics for paired samples from two patients that evolve from MDS to AML revealed changes in genes associated to

signalling pathways such as TGF-β and TNFα[83–85]. Our isogenic iPSC lines derived HPCs harbouring the genetic make-up of MDS27 patient allowed us to study the details of clonal evolution and the transcriptional changes imposed by heterozygous disruption of the C/EBPα bZIP domain during the disease progression of this patient as measured by single-cell RNAseq. Changes in the size of MDS-specific clusters were observed between low and high-risk samples and genes

**Fig. 7 | Distinct MDS transcriptome signature and changes in cellular composition during disease progression measured by single-cell transcriptome analysis. A** Analysis of scRNA-Seq data. Uniform Manifold Approximation and Projection for Dimensional Reduction (UMAP) map for individual samples. Each dot in the map represents a cell and is coloured according to cluster assignment. **B** Histogram showing the proportion of cells in each cell cycle phase within each cluster as identified by the expression of cell cycle-regulated genes. **C** 13 clusters based on the expression of specific genes. Cluster annotation was based on manual curation of marker genes. Each dot in the map represents a cell and is coloured according to cluster assignment. L = Low risk, H = high risk. **D** Table representing the number of cells in each cluster. Blue squares, clusters reduced during disease progression; red squares, clusters increased during disease progression; green squares, no change during disease progression. **E** Normalized enrichment scores (NES) calculated using gene set enrichment analysis (GSEA) with gene sets related

to HSC and LSC signatures in single-cell RNA-Seq clusters 6 and 7, showing significant association with HSC and LSC signatures in these clusters (as indicated by a positive NES and Adjusted *p*-value < 0.1). NES and *p*-values were calculated using a permutation-based test (one-sided) using the GSEA software[109]. **F** Expression of indicated genes projected on the UMAP map. Colour intensity represents expression data log2 normalized unique molecular identifier (UMI) counts. **G** Dot plots showing the scaled expression level of the indicted transcription factors in low-risk and high-risk MDS samples. Colours represent the scaled expression and size encodes the proportion of gene-expressing cells. **H** Dot plots showing the scaled expression level of the indicted myeloid genes in control and low-risk samples. Colours represent the scaled expression and size encodes the proportion of gene-expressing cells. **I** Monocle pseudo-time trajectory for MDS specific, myeloid, and erythroid cells projected on the UMAP map of scRNA clusters. Cells are coloured according to their pseudo-time value.

previously associated with AML such as *ID1, CD93* and *CALCRL* were activated by cells within these clusters together with genes coding for several insulin-growth factor binding proteins (IGFBP). For example, *IGFBP3*, associated with the early stage of MDS[86], was expressed by cells in low-risk but not by cells in high-risk clusters. Additionally, our data showed an upregulation of genes encoding for important signalling pathways associated with MDS progression and AML such as Wnt and TGFβ signaling[84,87–90]. Our single-cell RNAseq analysis also revealed that disruption of C/EBPα bZIP domain affects the cell cycle of the cells, with a block in G1.

High-risk MDS *CEBPA* mutations occur at disease progression; a disease stage with more than 50% of patients not responding to hypomethylated agents. Our isogenic iPSC lines provide a bona fide in vitro experimental system that could be utilized as a platform for drug screening and for future studies regarding the molecular mechanisms of disease development and drug resistance in both settings.

## Methods

### Human iPSCs reprogramming using the integration-free Sendai virus

Human patients with MDS were recruited from the clinic held at the Centre for Clinical Haematology, University Hospital Birmingham NHS Foundation Trust. Patient MDS27 has read the patient information sheet and signed the consent form. The study was conducted according to Good Clinical Practice guidelines, consistent with the principles that have their origin in the Declaration of Helsinki. The study was approved by the West Midlands−Solihull Research Ethics Committee (10/H1206//58). Human iPSCs from MDS27 were generated from the peripheral blood sample collected in 2013. MDS27 cryopreserved peripheral blood mononuclear cells were thawed and cultured in an expansion medium in a 24-well plate for 3–6 days. On day 6, the cells were counted and $25 \times 10^4$ cells were resuspended in 300 fresh expansion media with 4 μg/ml Polybrene (Sigma, TR-1003) and placed in a 15 ml falcon tube for transduction, MOI 3 of Sendai Virus containing the four Yamanaka factors was added to the cells for three hours at 37 °C. Two days later, the transduced cells were centrifuged at 300×*g* for 8 min and plated in a six-well plate containing 90%-confluent of mitomycin C-treated MEFs and "day 8 medium" for 3 days at 37 °C, and every 2 days the medium was changed. At day 11, the medium was replaced with "day 11 medium" and incubated for 4 days at 37 °C and every 2 days the medium was changed with "day 11 medium". On day 16, the medium was changed to hiPSC medium and every day thereafter, the medium was changed to fresh "hiPSC medium". At day 25, emerging iPSC colonies were picked individually into a six-well plate coated with Matrigel hESC-qualified Matrix (Corning Life Science, 354277). The colonies were cultured in a hiPSC medium with 10 μM of Rock inhibitor (Y-27632, LKT laboratory, Y1000) and incubated at 37 °C. From the first day after picking onwards, the colonies were

cultured continuously using hiPSC medium without Rock inhibitor. Media composition is provided in Supplementary data.

### Human iPSCs generation by integration-free episomal reprogramming

The episomal reprogramming of MDS27 PBMNCs from the sample obtained at diagnosis was performed as previously described[9], using the episomal reprogramming plasmids pCXLE-OCT4+shP53 (27078), pCXLE-hOCT3/4 (27076), pCXLE-hSK (27078), pCXLE-hUL (27080) and pCXWB-EBNA1 (37624) obtained from Addgene. Briefly, PBMNCs were cultured in one well of a 24-well plate with 1 ml of "PBMNCs expansion medium" for 3 days (Supplementary Information Table 1). After 3 days, cells were collected by centrifugation and counted, $3 \times 10^5$ were used for nucleofection. Two nucleofection reactions were performed ± pCXLE-OCT4+shP53. Plasmids were used at 3 μg per reaction (<5% DNA volume). Condition one: 0.83 μg of pCXLE-OCT4 + shP53 + 0.83 μg of pCXLE-hSK + 0.83 μg of pCXLE-hUL + 0.5 μg of pCXWB-EBNA1. Condition two: 0.83 μg of pCXLE-hOCT3/4 + 0.83 μg of pCXLE-hSK + 0.83 μg of pCXLE-hUL + 0.5 μg of pCXWB-EBNA1. The nucleofection was performed using the CD34+ cell nucleofector kit (Lonza, VPA-1003), programme U-008, for the nucleofector 11/2b device (Lonza). Following nucleofection, the cells were cultured in one well of a 24-well plate with 1 ml of "PBMNCs expansion medium" for 3 days. On day 3 after nucleofection, cells were plated onto 90% mitomycin C-treated MEFs in a six-well plate with 2 ml of "PBMNCs expansion medium". On day 5 after nucleofection, an equal volume of hESC medium was added to the cells. On day 7 after nucleofection, the medium was replaced with hESC medium + 10 μM Y27632, and the medium changed daily from day 8 onwards. On day 17 after nucleofection, the appearance of hiPSC colonies was apparent and the plate was maintained until day 32 to allow colony formation to occur. Individual colonies were picked and passaged for one passage into a single well of a 6-well plate coated with mitomycin C-treated MEFs and maintained in hESC medium without Y27632. Then, each hiPSC line was maintained on a matrigel in StemFlex medium to use them for further experiments. Media composition is provided in Supplementary data.

### AP staining
AP staining was performed according to standard protocols[91].

### Immunofluorescence for cytoplasmic marker TRA1-81
Cells were washed twice with StemFlex medium and incubated with TRA1-81 mouse primary Ab (MA1-024, Clone C1.261, Invitrogen) for one hour at 37 °C. After three washes with StemFlex medium, the secondary Ab goat anti-mouse-IgM-Alexa, Fluor 488 (A10684, Invitrogen) was added to the cells and incubated for one hour at 37 °C. Then, cells were washed with PBS and fixed with 2% (v/v) Formaldehyde methanol free (PFA, Thermo Scientific, 28906) for 10 min at room

temperature. Cells were washed twice, mounted and imaged under a fluorescence microscope (Leica DM6000, Leica Microsystems).

## Immunofluorescence for the nuclear markers NANOG and SOX2

Cells were fixed with 4% PFA (v/v) in PBS for 20 min at room temperature. Then, the cells with aldehyde groups were quenched for 10 min with 50 mM NH$_4$Cl (Sigma, 254134) in PBS at room temperature. Afterwards, the cells were permeabilized with 0.5% (v/v) Triton X-100 (Sigma, 11332481001) in PBS for 15 min at room temperature and blocked with PBS containing 1% (w/v) bovine serum albumin (BSA, Sigma, A1933) + 0.3% (v/v) Triton X-100 + 10% (v/v) FBS + 1% (v/v) goat serum (Sigma, G9023) for 1 h at room temperature. Then, the cells were incubated with diluted primary mouse antibodies SOX2 (AF2018, clone 245610, R&D) and NANOG (AF1997, clone ABZ92376, R&D) in a blocking buffer for 1 h at room temperature. After the primary antibody, the cells were washed for 30 min with PBS + 0.1% (v/v) Tween-20 (Sigma, P9416), and goat anti-mouse secondary antibody (A21052, Invitrogen) was applied to the cells for 1 h at room temperature. Finally, the cells were washed with PBS, mounted, observed, and imaged under a fluorescence microscope (Leica DM6000, Leica Microsystems).

## Trilineage differentiation of human iPSC

Differentiation towards the three germ layers was assessed using the STEMdiff Trilineage differentiation kit (Stem Cell Technology, 05230) performed as described by the manufacturer.

## Hematopoietic differentiation using Stemdiff protocol

Stemdiff protocol (Stem cell technology, 05310) was followed to differentiate hiPSCs into hematopoietic progenitor cells as described by the manufacturer. Flow cytometer analysis was used to evaluate HPC formation using CD43-APC (560198, clone 1G10, BDPharmingen), CD45-FITC (11045942, clone HI30, eBioscience), and CD34-PE (550619, clone 8G12, BD Pharmingen). Cells were subjected to 1 h Fcblock (eBioscience, 14916173, Clone AB468581) on ice prior to staining with antibodies Isotype controls mouse IgG1k-PE (clone MOPC-31C, BD Pharmingen) and mouse IgG1k-FITC/APC (clone P.3.6.2.8.1, Thermofisher) were used.

## Clonogenic progenitor assay

10,000 Hematopoietic cells from day 12 of stemdiff differentiation were plated in 35-mm plastic dishes using 1.2 ml per dish of complete MethoCult H4435 medium in duplicate. The plates were incubated at 37 °C, 5% CO$_2$ for 14 days. Colonies were scored after 14 days of incubation based on the morphological criteria as erythroid colonies (CFU-E), granulocyte, erythrocyte, macrophage, megakaryocyte colonies (CFU-GEMM), granulocyte/macrophage colonies (CFU-GM), granulocyte colonies (CFU-G), and macrophage colonies (CFU-M).

## Erythroid differentiation

1–2 × 10$^5$/ml Day 10 hematopoietic Stemdiff cultures were plated in a 6-well plate and cultured for 18 or 20 days using the media with different cytokines according to published protocol[92]. Briefly: cells were cultured in IMDM containing 3% (w/v) Albumin serum (sigma, H4522), 3 mg/ml human plasma (Finished Product, QE Pharmacy) 10 µg/ml insulin (Sigma, 19278), 3 U/ml heparin (Sigma, H3149) and 500 µg/ml human transferrin (R&D, 2914-HT-001G). Hematopoietic cells were stimulated with the following cytokines. Day 0–Day 8: 100 ng/ml SCF, 5 ng/ml IL-3, and 3 U/ml EPO; Day 8 to Day 11: 100 ng/ml SCF and 3 U/ml EPO; Day 11 to Day 20: 3 U/ml EPO. The erythroid differentiation was monitored every 4 days by flow cytometry analysis of erythroid markers: CD71-APC (17071941, clone OKT-9, eBioscience), glycophorin A (CD235a-PE)(12-9987-80, clone GA-R2, eBioscience). Cells were subjected to 45 min Fcblock (eBioscience, 14916173, Clone AB468581) on ice prior to staining with antibodies. Isotype controls mouse IgG1k-

APC (clone P.3.6.2.8.1, Thermofisher) and mouse IgG2bk-PE (clone Ebmg2b, Thermofisher) were used for gating. Morphological analysis of erythrocytes was assessed after Kwik-Diff staining (Thermofisher, 9990700).

## Myeloid differentiation

1 × 10$^5$/ml Day 12 hematopoietic Stemdiff cultures were plated in a 12-well plate and cultured for 7 days in myeloid differentiation media[93], containing Stem line II supplemented with 1% Pen/strep and cytokines. Hematopoietic cells were stimulated with the following cytokines: Day0-day3: 10 ng/ml IL-3, 10 ng/ml GM-GSF, 30 ng/ml G-CSF, 50 ng/ml FLT-3 (300-19, Peprotech), 50 ng/ml hSCF; Day3-Day7: 30 ng/ml G-CSF. The myeloid differentiation was monitored on day 4 and day 7 by flow cytometry analysis of myeloid markers: CD14-APCCy7 (47014942, clone 61D3, eBioscience) and CD11b- PECy7 (15518356, clone ICRF44, eBioscience). Cells were subjected to 45 minutes Fcblock (eBioscience, 14916173, Clone AB468581) on ice prior to staining with antibodies. Morphological analysis of myeloid cells was assessed after Kwik-Diff staining. Isotype controls mouse IgG1k-APCCy7/PECy7 (clone P.3.6.2.8.1, Thermofisher) were used for gating.

## Xenotransplantation

NOD/SCID/IL2rγ$^{-/-}$/Tyr+/KitW41J (NBSGW) mice were purchased from the Jackson Laboratory (Bar Harbor, Maine, USA) and bred at the Francis Crick Institute Biological Resource Facility. Both male and female mice, aged 8 to 12 weeks, were used in this study. All animal experiments were performed at the Francis Crick Institute in accordance with UK Home Office and institutional guidelines under the Home Office project license PLL 70/8904.

Human hematopoietic cell enrichment was performed using EasySep release human CD45 positive selection kit (StemCell Technologies) according to the manufacturer's instructions. Following on, CD45$^+$ human haematopoietic cells were injected into the mice via two routes: either intravenously through the tail vein (up to 1 × 10$^6$ cells) or intra-bone (0.5 × 10$^6$ cells) into the femur. Mice were sacrificed 12 weeks post-xenotransplantation, and bones (femurs, tibias, pelvis) and spleens were harvested. For those receiving human hematopoietic cells via intra-bone route, the injected and non-injected bones were processed separately. Recovered cells were then processed for flow cytometry analysis.

## Humanized ectopic scaffolds in immunodeficient mice

Humanized scaffolds were prepared as described previously[32]. Briefly, gelatin-based sponges (Spongostan standard, Ethicon, MS0002) were sectioned into 48 slices (8.75 mm × 6.25 mm × 10 mm). Adult bone marrow MSCs (1 × 10$^5$, Passage P2–P3) in 100 µl of MSC culture media (MEM-α Medium, Gibco; 1% Penicillin–Streptomycin, Sigma-Aldrich and 10% human MSC-FBS, Gibco) were injected into each scaffold using a sterile insulin syringe. Scaffolds were maintained for 48 h. Following on, enriched CD45$^+$ human haematopoietic cells (0.5 × 10$^6$ cells, as described above) were injected into each scaffold. Scaffolds were maintained at 37 °C and 5% CO$_2$ for 24 h in MyeloCult H5100 (with 1% P/S, Stemcell technologies) supplemented with cytokines (20 ng/ml G-CSF, 20 ng/ml IL-3, and 20 ng/ml TPO from PeproTech).

NOD/SCID/IL2rγ$^{-/-}$/IL-3/GM/SF (NSG-SGM3) mice were originally obtained from Leonard Shultz (The Jackson Laboratory, Bar Harbor, Maine, USA) and bred at the Francis Crick Institute Biological Resource Facility. Both male and female mice, aged 8–12 weeks, were used in this study. Surgical implantation of the humanized scaffolds was performed by following local named veterinary surgeon (NVS) guidelines for aseptic techniques as described previously[32]. Briefly, mice were anaesthetised in a chamber filled with 0.5% isoflurane and 2 L/min O2. Analgesia (Buprenorphin, 0.1 mg/kg and Meloxicam, 10 mg/kg) was administered via the subcutaneous route. 10% chlorhexidine solution was used to sterilize the skin around the surgical area. A 0.5 cm

anterior-to-posterior incision of the skin was created and then a pocket was made under the skin. Up to 3 pre-seeded scaffolds were inserted into the incision, making sure it was placed deep within the pocket and the incision site was closed with surgical staples. Surgical staples were removed on day 7 after the surgical procedure. Mice were maintained for 12 weeks and then sacrificed to recover the humanized scaffolds.

## Flow cytometry analysis for xenografted samples

Xenografted cells recovered following culling of the mice were recovered as described previously[32]. Cells were then stained with antibodies specific for human or murine antigens [mCD45-PerCPCy5.5 (Clone 30-F11, eBioscience), hCD45-APCef780 (Clone HI30, eBioscience), hCD33-APC (Clone WM53, BD Pharmingen), hCD3-PE (Clone UCHT1, BD Pharmingen), and hCD19-PE (Clone HIB19, BD Pharmingen)]. Live/dead cells were discriminated using DAPI (4,6, diamidino-2-phenylindole) staining. Following on, cells were immunophenotyped by using a Fortessa flow cytometer (BD Biosciences, Oxford, UK).

## CRISPR-Cas9 to generate C/EBPα bZIP frameshift mutation and revert RUNX1Gly217Profs and SRSF2 Pro95His mutations

A sgRNA targeting C/EBPα gene on chromosome 19 at position 13.11, sgRNA targeting RUNX1 exon 7 and sgRNA targeting SRSF2 in H95 were designed using an online tool (Trust Sanger Institute Editing database). sgRNAs were first phosphorylated, annealed and cloned into pSpCas9 (BB)−2A-GFP (PX458) (Addgene, 48138) following the protocol provided by Ran et al. [94]. For reverting the RUNX1Gly217Pro mutation, and SRSF2 P95H, oligos with wild type RUNX1 and SRSF2 sequences were designed using IDT webpage (ssODN). P3 amaxa kit (Lonza, V4XP-3024) was used to nucleofect iPSC (programme DS-150). In the case of RUNX1 and SRSF2, 1 μg of plasmid containing sgRNA was co-transfected with 5 μl of 100 μM ssODN and cells were cultured in the presence of 0.5 μM Alt-R HDR Enhancer V2 (IDT). GFP+ cells were sorted 24 h after using BD FACSAriaTM Fusion (BD bioscience). Seven days post sorting, 24 clones for CEBPA and 16 clones for SRSF2/RUNX1 were picked and expanded to extract genomic DNA for assessing the gene editing efficiency. To evaluate nucleofection efficiency, the target region was amplified using Q5 High-Fidelity DNA polymerase (NEB, M0491) and two designed primers used for C/EBPα PCR: C/EBPα forward 5'GGCCTCTTCCCTTACCAGCC3' and C/EBPα reverse 5'CTGGTCAGCTCCAGCACCTT3'; for RUNX1 PCR: RUNX1 forward 5'TGGGCTCCATCTGGTACTTA3' and RUNX1 reverse 5'TGGCATATCTCTAGCGAGTCTAT3'; for SRSF2 PCR: SRSF2 Forward 5'GTGGACAACCTGACCTACCG3' and SRSF2 Reverse 5'ATTATCTCGCCGCCAGACG 3'. For CEBPA, PCR products were processed using a T7 endonuclease I (T7EI) assay using the Alt-R genome editing detection Kit (IDT, 1075932) according to the manufacturer's protocol. For RUNX1, PCR products were subjected to SmaI enzymatic digestion whose restriction site is present in the wild type but not in the mutant allele.

## qRT-PCR

RNA extraction was performed with Trizol reagent (15596026, Invitrogen) from HPCs from day 12 of stemdiff differentiation and myeloid cells from day 4 and day 7 of myeloid differentiation. qPCR was carried out in a Stratagene Mx3005P (Agilent Technologies) using Taqman primers CEBPA Hs00269972_s; RUNX1 Hs01021970_m1; GATA2 Hs00231119_m1; SPI1 Hs02786711_m1; LMO2 Hs99999906_m). Ct values were calculated and generated by MxPro 3000 Stratagene software. Different gene relative expression values were calculated against GAPDH (Hs02758991_g1) using the ΔΔCt mathematical model[95].

## ATAC-seq

Fragment transposition ATAC libraries were generated following the Omni-ATAC protocol from[96]. $50 \times 10^3$ viable sorted cells (CD34+ cells

from MDS27 patient and CD34+/CD45+ sorted cells from HPC-iPSC derived) were pelleted in a fixed angle centrifuge at $500 \times g$ at 4 °C for 5 min, resuspended in 50 μl of cold ATAC-Resuspension Buffer (RSB) (1 M Tris−HCl pH 7.4, 5 M NaCl, 1 M $MgCl_2$ in sterile H2O) containing 0.1% NP40 (Sigma-Roche), 0.1% Tween-20 (Sigma-Roche), 0.01% Digitonin (Promega) and incubated in ice for 3 min. Lysis was washed out by adding 1 ml of cold ATAC-RSB containing 0.1% Tween-20 (Sigma/ Roche) only. Nuclei were pelleted at $500 \times g$ at 4 °C for 10 min and resuspended in 50 μl of transposition mix (25 μl of 2x TD buffer (Illumina), 2.5 μl of Tn5 Transposase enzyme (Illumina), 16.5 μl of PBS, 0.5 μl of 1% Digitonin (Promega), 0.5 μl of 10% Tween-20 (Sigma-Roche), 5 μl of $H_2O$). The mixture was then incubated at 37 °C for 30 min in a heated block shaking at 1000 RPM. This reaction allowed the Tn5 transposase to simultaneously fragment and tag the chromatin with sequencing adapters. Transposed fragments were purified by using the MinElute Reaction Cleanup Kit (QIAGEN) following the manufacturer's instructions and eluted in 21 μl of $H_2O$. The whole eluted product was amplified for 5 cycles. Pre-amplified library was stored in ice. Reagent volume (μl) 29 Primer Ad1 25 μM 2.5 Primer Ad2 25 μM 2.5 NEBNext Master Mix 2x (NEB) 25 Transposed Sample 20 PCR reagents for the pre-amplification of transposed fragments Temperature (°C) Duration Cycles 72 5 min 98 30 s 1 98 10 s 63 30 s 5 72 1 min 4 ∞ PCR conditions for the pre-amplification of transposed fragments. To determine the number of additional cycles needed for an optimal library amplification, 5 μl of pre-amplified library was used to set up an RT-qPCR reaction. Linear relative fluoresce values were plotted against cycles, and the number of additional cycles needed was defined as the cycle number corresponding to 1/3 of the maximum fluorescence intensity (7). Reagent volume (μl) Sterile $H_2O$ 3.76 Primer Ad1 25 μM 0.5 Primer Ad2 25 μM 0.5 SYBR Green 25x (in DMSO) 0.24 NEBNext Master Mix 2x (NEB) 25 Transposed Sample 20 30 Setup of the RT-qPCR to determine the number of additional cycles of ATAC library amplification Temperature (°C) Duration Cycles 98 30 s 1 98 10 s 63 30 s 20 72 1 min 4 ∞ RT-qPCR conditions to determine the number of additional cycles of ATAC library amplification The remainder of pre-amplified library was amplified for the additional cycles required. The reaction was purified using the QIAquick PCR Purification Kit (QIAGEN) following the manufacturer's instructions. Briefly, the sample was mixed with 5 volumes of Buffer PB and loaded onto a MinElute column (QUIAGEN). The column was centrifuged and washed with 750 μl of Buffer PE. Two sequential centrifugations were performed to remove any residual ethanol. All centrifugation steps were performed at $17,900 \times g$ for 1 min. Libraries were eluted in 20 μl of $H_2O$ and, to avoid adapter contamination, further purified by adding 1.2x volumes of AMPure XP beads (Beckman Coulter) following the manufacturer's instructions. The libraries were then eluted in 20 μl of $H_2O$. Sequencing was performed using a NextSeq2000 to obtain a minimum of 50 million reads per sample.

Raw sequencing reads were processed to remove low-quality bases and Nextera ATAC adapter sequences using Trimmomatic v0.39[97]. Reads were then aligned to the human genome (version hg38) using Bowtie2 v2.2.5[98] with the --very-sensitive-local parameter. Potential PCR duplicates were removed from the alignments using Picard MarkDuplicates v2.16.10 (http://broadinstitute.github.io/ picard). Peaks were called using MACS2 v2.2.7.1[99] with the options --nomodel --call-summits -q 0.05 -B --trackline. The resulting peaks were then filtered to remove peaks found in the hg38 blacklist[100] and to retain only peaks that had a summit height >5 in both replicates of either low- or high-risk MDS clones. Peaks were then combined to produce a single peak union using the merge command in BedTools v2.30.0[101]. Read counts were obtained using featureCounts v2.0.1[102] and normalized using edgeR v3.36.0[103] in R v4.1.2. Differentially accessible peaks were identified using the voom method in the Limma package v3.50.3[104]. A peak was considered to be differentially accessible if it had a fold-difference of at least 1.5 between conditions. Motif

enrichment analysis was carried out in the sets of differentially accessible peaks using the findMotifsGenome.pl function in Homer v4.9.1[105] with the options -size 200 -noknown. To create read density plots, peaks were first ranked according to their fold-difference between low- and high-risk MDS cells. Read densities were then calculated in a 2 kb window centered on the peak summit using the annotatePeaks.pl function in Homer with the options -size 2000 -hist 10 -ghist -bedGraph and the bedGraph files produced by MACS2. These were then plotted as a heatmap using Java TreeView v1.1.6r4[106].

## Single-cell RNA sequencing
Hematopoietic cells from the BU3-10 hiPSC control, MDS27-C22 (low risk) and MDS27-C22.7 lines were collected on day 12 of hematopoietic differentiation culture. Erythroid cells were collected on day 5 of the erythroid culture from BU3-10 hiPSC control and MDS27-C22. Freshly collected cells were passed through columns for cell removal to achieve a viability of >90%. Cells were resuspended in PBS/0/04% BSA (PBS without EDTA and Mg). 100 µl of single-cell suspension at a concentration of 700–1200 cells/µl ($1.2 \times 10^5$/100 µl) was submitted to the sequencing service. Library was performed according to the manufacturer's instruction (3' scRNAseq v3.1 10x Chromium) and samples were sequencing using Illumina NextSeq 500. Samples were sequenced at an average of 50,000 reads per cell.

Reads from single-cell RNA-Seq experiments were aligned to the human genome (version hg38) and quantified using the count function in CellRanger v4.0.0 from 10x Genomics and using gene models from Ensembl as the reference transcriptome. Single-cell analysis was carried out using the Seurat package v4.1.0[107] in R v4.1.2. The data was first filtered to remove cells that had <500 or >5000 expressed genes detected, as well as those that had >20% of reads aligned to mitochondrial transcripts. The count data from the individual samples were then combined into a single dataset, which was then normalized and scaled using the NormaliseData and ScaleData functions in Seurat, using the top 2000 most variable genes. Clustering was then performed by first conducting a principal components analysis (PCA) and selecting the top 26 principal components for further analysis. The clusters were then identified using the FindClusters function in Seurat using a resolution value of 0.3.

Single-cell trajectory analysis was carried out using Monocle3 v0.2.3[108]. Processed data from Seurat was imported into Monocle using the as.cell_data_set command in SeuratWrappers (https://github.com/satijalab/seurat-wrappers). Trajectories were then inferred using the learn_graph command in Monocle. Pseudotime was calculated using the order_cells command, using the earliest inferred HSC population as the root node. These were then plotted using the UMAP coordinates calculated by Seurat.

## Statistical analysis
The statistical analysis was performed using GraphPad Prism 8 (GraphPad Prism version 8.0 for Mac, GraphPad Software, San Diego, CA, USA). All data are expressed as mean ± standard error of the mean (SEM). $P$ values < 0.05 were considered statistically significant and a star (*) was labelled in the figures. The following statistical analyses were used: Two-way ANOVA to analyse data describing two factors across multiple parametric groups. One-way ANOVA to analyse data describing one factor across multiple parametric. For complete statistical comparisons see Supplementary Data Table 5.

## Reporting summary
Further information on research design is available in the Nature Portfolio Reporting Summary linked to this article.

## Data availability
The ATAC-seq and single-cell RNAseq data generated in this study have been deposited in the Gene Expression Omnibus database under accession code GEO:GSE236710. The processed data generated in this study are provided in the Supplementary Information/Source Data file. The human gene sets used in this study are available at the Molecular signatures database (gsea-msigdb.org), under the following links: EPPERT_CE_HSC_LSC (gsea-msigdb.org); EPPERT_HSC_R(gsea-msigdb.org); EPPERT_LSC_R(gsea-msigdb.org); GENTLES_LEUKEMIC_STEM_CELL_DN (gsea-msigdb.org); GENTLES_LEUKEMIC_STEM_CELL_UP (gsea-msigdb.org); VALK_AML_WITH_CEBPA (gsea-msigdb.org); VALK_AML_CLUSTER_4 (gsea-msigdb.org); VALK_AML_CLUSTER_15 (gsea-msigdb.org); JAATINEN_HEMATOPOIETIC_STEM_CELL_DN(gsea-msigdb.org); JAATINEN_HEMATOPOIETIC_STEM_CELL_UP (gsea-msigdb.org). Artwork generated with powerpoint Bundle-Biology: scienceppt.com/b/science-ppt-bundle-biology. Source data are provided with this paper.

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

## Acknowledgements

The authors wish to thank Dr. Giacomo Volpe for the critical reading of the manuscript. We also thank Genomics Birmingham for NGS sequencing and Mary Clarke for cell sorting. R.A. and Y.A. PhD studentships were funded by the Ministry of Education and the Royal Embassy of Saudi Arabia Cultural Bureau. This work was also funded by a FIS grant ISCIII (PI19/00730) to EB and (PI21/01641) to R.T.-R. Work in the D. Bonnet lab was supported by the Francis Crick Institute, which receives its core funding from Cancer Research UK (CC2027), the UK Medical Research Council (CC2027)), and the Wellcome Trust (CC2027.) Work in C. Bonifer's lab was funded by grants from Blood Cancer UK (15001) and the Medical Research Council (MRC) (MR/S021469/1) and is currently funded by the Novo Nordisk Foundation Center for Stem Cell Medicine, reNEW, supported by Novo Nordisk Foundation grant number NNF21CC0073729.

## Author contributions

Methodology, R.A., A.Y., R.B., S.M., K.H., A.A., P.G.; Formal analysis, P.K., C.W., I.A., T.S., S.M., R.-T.-R.; Investigation, R.A., A.Y., C.S., E.B., T.S., A.F., S.K., M.B., P.G.; Resources, G.J.M., P.M., M.R., E.P., D.B.; Writing—original draft, P.G.; Writing—review and editing, P.G., C.B., S.K., E.B., G.J.M., P.M., S.M., and D.B.; Funding acquisition, P.G., C.B., D.B., and E.B.; Conceptualization, P.G.; Supervision, P.G. and C.B.; Project administration, P.G.

## Competing interests

Pablo Menendez is co-founder of OneChain Immunotherapeutics, a spin-off company from the Josep Carreras Leukaemia Institute. The remaining authors declare no competing interests.

## Additional information

[1]Department of Cancer and Genomic Sciences, College of Medicine and Health, University of Birmingham, Birmingham B15 2TT, UK. [2]Department of Laboratory Medicine (Haematology), Faculty of Applied Medical Sciences. Albaha University, Kingdom of Saudi Arabia, Al Bahah, Saudi Arabia. [3]Department of Medical Laboratory Technology, College of Nursing and Health Sciences, Jazan University, Jazan, Saudi Arabia. [4]School of Bioscience, College of Life and Environmental Sciences, University of Birmingham, Birmingham B15 2TT, UK. [5]Haematopoietic Stem Cell Laboratory, The Francis Crick Institute, London NW1 1AT, UK. [6]Molecular Biology Unit, Clinical Analysis Service, Hospital Universitario y Politécnico La Fe, Valencia 46026, Spain. [7]Josep Carreras Leukemia Research Institute, Barcelona, Spain. [8]Centro de Investigación Biomédica en Red de Cancer (CIBERONC), Instituto de Salud Carlos III, Madrid 2029, Spain. [9]Red Española de Terapias Avanzadas (TERAV), Instituto de Salud Carlos III, Madrid, Spain. [10]Institució Catalana de Recerca i Estudis Avançats (ICREA), Barcelona, Spain. [11]Department of Biomedicine, School of Medicine, University of Barcelona, Barcelona, Spain. [12]Institut de Recerca Hospital Sant Joan de Déu–Pediatric Cancer Center Barcelona (SJD-PCCB), Barcelona, Spain. [13]Section of Hematology and Oncology, Department of Medicine, Boston University School of Medicine, Boston, MA 02118, USA. [14]Center for Regenerative Medicine (CReM), Boston University and Boston Medical Center, Boston, MA 02118, USA. [15]Research Center, King Faisal Specialist Hospital and Research Center-Jeddah, Jeddah, Saudi Arabia. [16]Pathology Department, Faculty of Medicine, Umm Al-Qura University, Makkah, Kingdom of Saudi Arabia. [17]Department of clinical Laboratory sciences, Faculty of Applied medical sciences, Umm Al-Qura University, Makkah, Kingdom of Saudi Arabia. [18]Molecular Cytogenetics and Genome Editing Unit, Human Cancer Genetics Program, Centro Nacional de Investigaciones Oncologicas (CNIO), Madrid, Spain. [19]Division of Hematopoietic Innovative Therapies, Biomedical Innovation Unit, Centro de Investigaciones Energeticas, Medioambientales y Tecnologicas (CIEMAT), Madrid, Spain. [20]Advanced Therapies Unit, Instituto de Investigacion Sanitaria Fundacion Jimenez Diaz, Madrid, Spain. [21]Centro de Investigacion Biomedica en Red de Enfermedades Raras (CIBERER), Madris, Spain. [22]Department of Oncological Sciences, Tisch Cancer Institute, Icahn School of Medicine at Mount Sinai, New York, NY, USA. [23]Center for Advancement of Blood Cancer Therapies, Icahn School of Medicine at Mount Sinai, New York, NY, USA. [24]Department of Metabolism and Systems Science, College of Medicine and Health, University of Birmingham, Birmingham, UK. [25]Novo Nordisk Foundation Center for Stem Cell Medicine, Murdoch Children's Research Institute The Royal Children's Hospital, 50 Flemington Road, Parkville, VIC 3052, Australia. [26]These authors contributed equally: Ruba Almaghrabi, Yara Alyahyawi. ✉e-mail: p.garcia@bham.ac.uk

