## [Transparent Peer Review file · Nature Communications]

A heterozygous *CEBPA* mutation disrupting the bZIP domain in a *RUNX1* and *SRSF2* mutational background causes MDS disease progression

Corresponding Author: Dr Paloma Garcia

Version 0:

Reviewer comments:

Reviewer #1

(Remarks to the Author)

Almaghrabi et al. established iPSCs from one single MDS patient who over a 2-year period progressed from low-risk MDS to high-risk MDS and aimed to apply the generated iPSCs to cellularly and molecularly model MDS disease progression associated with acquisition of a mutation predicted to disrupt the CEBPA bZIP domain. Using several in vitro culture assays including replating capacity and lineage-differentiation capacity, epigenetic profiling and single cell RNA sequencing, they conclude that their system model clonal evolution of MDS, where acquisition of a CEBPA mutation is the main driver for disease progression, in part through restructure of the chromatin landscape. Furthermore, the authors claim to demonstrate changes in the clonal composition as a consequence of CEBPA mutations through their single cell gene expression analysis. While it remains important and novel to understand some of the fundamental questions raised related to MDS disease progression and therapy resistance, there are several limitations with the approaches used by the authors to address these and therefore these studies fail to provide convincing new evidence towards the conclusions listed above. As further detailed below, the modelling of MDS disease progression using iPSCs has already been performed by other studies, there already is convincing data in the literature with regard to the transformation potential of mutant CEBPA, the study is applied only to one iPSC line and without comparison to the patient's primary cells, and the authors do not model the specific mutation found in the patient.

Major comments:

1. A major limitation with this study is that it is restricted to the mutations found in one single MDS patient. For comparison, a recent study reported the generation of iPSCs from 15 AML cases, several who had progressed from MDS to AML and including one line which contained a late-occurring CEBPA mutation (Kotini et al., Blood Discovery, 2023). In contrast to Almaghrabi et al., Kotini et al. were able to generate iPSCs directly from the patient resulting in iPSC clones without CEBPA (but including all mutations acquired prior to CEBPA) and iPSC clones with CEBPA. In those studies no differences in hematopoietic function, including ability to engraft NSG mice, was observed between the two isogenic clones. This at minimum raises questions towards the study by Almaghrabi et al. as they applied CRISPR to molecularly engineer in a CEBPA mutation, which in fact generated a different mutated sequence compared to the mutation observed in the MDS patient at the high risk stage. In light of the concerns raised about the side-effects associated with CRISPR, including potential introduction of new genetic lesions as a consequence of low p53 expression (Haapaniemi et al., Nature Medicine, 2018), also raises the possibility that the effect observed by the authors could be mediated by other genes than CEBPA.
2. A second concern throughout the whole study is that iPSCs from one single MDS patient are not compared to their isogenic wildtype cells lacking the oncogenic mutations observed in the MDS patients, but instead compared to one single iPSC line generated from another, presumably healthy individual. This limits the strength of the conclusions in this study, both towards functional in vitro assays and molecular profiling, where the differences observed could in part potentially be explained by inherited genetic heterogeneity rather than somatically acquired mutations associated with the MDS. In addition to this, the authors focus most of their assays on only one single iPSC clone (clone 22).
3. The role of CEBPA towards leukemic transformation is well established from genetic modelling in mice (Heath et al., Blood, 2004; Zhang et al., Immunity, 2004; Kirstetter et al., Cancer Cell, 2008; Bereschenko et al., Cancer Cell, 2009). Combined with the extensive genetic characterization of human hematologic malignancies, including MDS and AML, demonstrating the recurrent nature of CEBPA mutations, predominately associated as a late event and associated with disease progression, already provides strong evidence towards the role of CEBPA in leukemogenesis. However, CEBPA

mutations come in different flavors, heterozygous and homozygous mutated states and variation in location and type of mutation, where some are associated with different clinical outcome. In light of this, novelty would be enhanced if one could model distinct CEBPA mutants and investigate their cellular and molecular impact against each other.

4. In addition to the acquisition of a CEBPA mutation during progression, progression was also associated with a dramatic clonal expansion of a sub-clonal RUNX1 frameshift mutation (15% to 47%), which combined with the already clonal L56 hotspot RUNX1 mutation (50% at both low and high risk stage), generates a bi-allelic RUNX1 mutated state. Considering the important role of RUNX1 in normal and leukemic hematopoietic stem and progenitor cell biology, this represents an interesting collaborator towards potentially enhancing the phenotype observed together with the CEBPA mutant as RUNX1 deletion has been previously shown to reduce Cebpa transcript and result in reduced granulopoiesis (Guo et al., Blood, 2012). This raises potentially interesting mechanistic questions including whether the phenotype observed of the heterozygous CEBPA mutant in the assays used by the authors is dependent on RUNX1 mediated suppression of the CEBPA wildtype allele. Something which could be further investigated by Crispr restoration of one or both RUNX1 alleles to wildtype configuration.

5. An important limitation for the conclusions that the authors make towards the relevance of their iPSCs at modelling the disease progression and therapy-resistance observed in the patient is the obvious lack of direct comparison to the primary cells from the actual patient at low-risk and high-risk stage of disease. Without this, the authors' claims to reproducing the clinical phenotype and relevance of iPSC clones are weak.

6. The authors focus their molecular analysis on the cells that they isolate from the hematopoietic differentiation protocol. As shown by the single cell gene expression analysis this is a highly heterogeneous population, where only 2% contain ability to form colonies in methylcellulose, thereby representing functional hematopoietic stem/progenitor cells. In light of the differences in CFUs, differences in preferential differentiation, and level of expression of critical markers such as CD34 (Suppl Fig. 1B) raises concerns about the cause and effect conclusions by the authors towards the role of CEBPA. The ATAC seq analysis therefore does not distinguish between whether the differences in peaks are due to CEBPA or represent differences in cellular composition of the cells analyzed. To resolve this, such analysis would require much more purified cells and/or single cell analysis.

7. With exception of the CFU assays, the authors make conclusions from in vitro differentiation cultures primarily based on the composition of cells without taking into account the total number of cells within the culture. Instead of presenting the percentage of cells in the culture, the author needs to rather convert these to absolute number of cells in order to allow interpretation of the results and to document the experiment variation.

8. The authors should provide better documentation of the FACS analysis shown in the manuscript, including clone name of antibodies, utilization of viability markers, etc. Furthermore, the profiles shown for several of the differentiation cultures should be further validated to ensure the gates used for distinguishing distinct cell stages and cell types are as intended. For the erythroid culture, it is surprising to see what appears to be down-regulation of CD235a (Figure 2E) in the late stages of the culture, this is particularly evident in the transition from day 7 to day 11 when the distinct CD71/CD235a double positive population is exhausted from the culture. It would be important to see cytopins with hematology stains from the CD71-CD235a^{low} cells and the residual CD71^{low}CD235a^{high} cells to demonstrate their morphology. Similar issues towards the placement of the gates to identify distinct cell types apply to Figure 3E and 4B for the markers applied there. Furthermore, the authors have in their methods indicated that DRAQ5 was included but does not show the gating strategy for this.

9. As the authors were unable to establish iPSCs from the high-risk stage containing the CEBPA mutation, they genetically engineered iPSCs to incorporate a CEBPA mutant. Following genetic engineering of genes, without functionally validating the impact on the protein and specific function, one can only make predictions about the impact the gene has on protein function. As the genetic change introduced by the authors is a frameshift mutation, with intact sequence prior to the mutation location, this leaves the potential for retained or altered function of the protein that is unique to the specific mutation. In light of these, the authors should throughout text adjust their description of the CEBPA mutation to predicted changes. Furthermore, the authors also compare the impact of the CEBPA mutant in iPSCs to CRISPR modified wildtype iPSCs. However, the specific mutation in the wildtype iPSCs is not documented, and if different from the MDS iPSCs limits the conclusions one can draw from this control as the effect of the individual mutations are not validated. Finally, the documentation of the CEBPA mutations in Supplemental Figure 3 represent interpreted sequences and the authors should complement these with the raw sequencing traces. Finally, the iPSCs should be validated by next-generation sequencing to confirm that no other oncogenic driver mutations are present.

10. The authors utilize the single cell RNA sequencing analysis (Figure 7) to make claims towards clonal evolution and loss of cellular identity. It is not clear what the authors mean with regard to clonal evolution as the data so far only suggest differences in cellular composition? In addition to the limitations described above with regard to being derived from one single patient, crispr modified and one single iPSC clone, there are also limitations with the analysis of this single cell RNA sequencing experiment. To understand stage-specific impact of the mutation, it is important to compare as closely related compartments as possible, and the strategy applied by the authors runs the likely risk of representing an apples and oranges comparison. In addition to the method description being poorly documented in order to understand the parameters behind the analysis, such as number of variable genes used for clustering and annotation of cell type, the analysis would benefit from more targeted analysis of corresponding compartments. As demonstrated by Figure 4D, the most distinct clusters occupied by wildtype cells also are represented by MDS cells from both low and high risk stages. It would be more relevant to perform comparisons within distinct clusters in order to understand the impact of CEBPA in a stage-specific manner, but to do so the authors are required to run more cells to reach statistical power for the comparisons. Furthermore, the authors included wildtype and low-risk MDS iPSCs subjected to erythroid differentiation in the analysis but did not do this for the high risk MDS iPSCs. Finally, without complementing the analysis on primary cells from the patient, as well as inclusion of more wildtype and MDS iPSCs from other patients, the conclusions from the authors are weak and speculative.

11. In Figure 5E the authors culture wildtype and MDS iPSCs in presence of Aza. Surprisingly, Aza leads to increase in CFUs from the high-risk iPSCs and does not demonstrate any impact on the low-risk or wildtype iPSCs. Although potentially interesting, the interpretation of this finding is limited as the experiments lack a positive control in the form of iPSCs from a Aza-responsive MDS patient.

12. Perhaps the most interesting data in the manuscript is the indication that high-risk MDS iPSCs confer potential to retain CFU potential following serial replating. Although this could support enhanced self-renewal, it would be more relevant to demonstrate this in a more rigid in vivo assay following transplantation into NSG mice, as reported for other iPSC lines modelling MDS and AML. Furthermore, did the authors investigate the impact of Aza on serial-replating capacity?

13. When comparing MDS low risk to high risk, are the authors using the Crispr control line or ancestral unperturbed line? This is important for the interpretation of the functional and molecular data from these comparisons.

Minor comments:

- In Figure 4E the authors show gene expression relative to HPC for each individual group. However, this could mask potential key differences initiated already at the HPC stage and the authors should present this data relative to the housekeeping gene only.
- It is not easy to follow the statistical comparisons performed in the figures (all groups compared to all groups, or selected groups) and what the significance refer to or the statistical analysis performed. The authors have used ANOVA but do not indicate whether or what method for multiple testing correction is being used. Can the authors also confirm that any bar without indication of significance was compared but not found significant? For example, it would be interesting to know whether normal versus low risk in figure 3B was significant as there appear to be big differences, which contrasts findings in Figure 5E for DMSO conditions.
- The authors claim that the profiles during the HPC differentiation are very similar for normal and low risk MDS lines but Supplemental Figure 1B show clear differences in the profile with regard to the level of intensity for CD34 expression which questions this conclusion.
- With regard to the comments above, lines 107-120 of the introduction contain several overstatements towards the findings of the manuscript and should be revised.
- As indicated above, the methodology should be better documented to provide better clarity of the design of the experiments performed. This pertains to cytokine conditions during iPSCs culture, antibodies and controls for gating strategies in flow cytometry experiments, statistical methods, and details for RNA sequencing analysis.
- There is much text (particularly in result section) and several figures that are purely technical and could be moved to methods and supplemental figures to enhance the focus of the paper.

Reviewer #2

(Remarks to the Author)

In this manuscript, Almaghrabi et al describe work using iPSCs generated from a patient with MDS who progressed from low to high-risk disease. Samples both time points were available for analysis. The authors utilize an induced pluripotent stem cell model to generate cell lines and examine how mutations in CEBPA (seen during progression from low to high risk in this patient) alter the transcriptional and epigenetic landscape of the cells to promote abnormal dysplastic differentiation. The authors should be commended on this extensive and detailed characterization. I do, however, have a number of concerns.

I think that the findings are overly broad given, especially given that samples are taken from only one patient and that the control cells were generated from an unrelated patient with an inherently different genetic background. Thus, there are a number of potential factors that could confound the results here and limit the strength of the conclusions. I think that in general, the analyses comparing the isogenic cells with or without CEBPA mutations is the stronger part of the paper and the comparisons to the wild type control cells less so.

There is also no effort to compare transcriptional changes seen in the iPSC models to that seen in the primary samples. It is always unclear to me how much of the transcriptional changes may be due to the process of in vitro manipulation and reprogramming.

The paper also suffers from some methodological issues, the most significant being the mis-classification of an ASXL1 mutation.

I do think that overall, this represents a significant amount of work and provides new data describing a very fascinating area of science involving the evolution of myeloid malignancies and with appropriate revisions would be of interest to researchers working on cancer evolution and on myeloid malignancies.

My detailed critiques are listed below:

-Line 104-106: "How these mutations affect the bZIP domain in the context of MDS have not been explored and it is unknown whether they are causative for a high risk of MDS disease progression." I think this is the fundamental question. There are a large number of reports sequencing patients with secondary AML, and CEBPA is included on most targeted NGS panels. If CEBPA mutations are common progression events then I would expect this to have been reported previously. It's one thing to show that a genetic event can produce disease-like effects in a model system, but if it doesn't occur regularly in human disease, then the it is unclear what the actual implications are. Is this an interesting case report, or does it have wider implications?

-Line 129: The authors report that the patient has an ASXL1 Glu 1102 Asp mutation. However, pathogenic mutations in ASXL1 are almost exclusively frameshift or nonsense mutations that are truncating, and are rarely, if ever SNVs. This

mutation has a VAF close to 50% in all the samples sequenced (supp table 1), and thus I think it is most likely a germline SNP. Unless the authors have other data demonstrating that this is a somatic mutation (such as sequencing from fibroblasts or T-cells) and pathogenically affects ASXL1 function, I would not say that this patient has an ASXL1 mutation. In contrast, the RUNX1 fs (but not the RUNX1 SNV, which is also likely germline) and the SRSF2 mutations are found at VAF's consistent with somatic events and these mutations expand over time. These SNV mutations, because they are likely germline, can also not be used to infer normal vs MDS derived iPSC colonies (line 138).

-For the clonogenic and differentiation assays, it would be nice to know how this compared to the primary cells from the MDS that did not undergo reprogramming. How representative of the original MDS are these iPSC derived HPCs?

-Line 224: Can the authors explain why they think they were unable to generate iPSC lines from the high-risk MDS sample? Other groups have reported generating iPSC lines from similar CEBPA mutated AMLs (Wang et al. Cell Stem Cell. 2021).

-Line 288-291: "Our results also show that introducing the CEBPA mutation on the background of additional mutations (ASXL1, RUNX1, SRSF2) but not by itself increased the self-renewal capacity of the committed progenitor cells and blocked myeloid differentiation, which is the main characteristic of high-risk MDS and AML." To really make this firm a conclusion, I think the authors would need to engineer a RUNX1 and/or SRSF2 mutation into the wild type control iPSCs and then show that the CEBPA mutation could transform them. As it is, this is just a single control sample generated from an unrelated patient. There are numerous other factors that could contribute to this finding.

-It would be helpful to include flow plots for Figure 5A in a supplemental figure, similar to what was shown in Figure 2E.

-Line 349: Again, these cells do not contain a mutation in ASXL1 (which is also misspelled).

-Section starting with Line 352: Other reports (Wang et al. Cell Stem Cell. 2021) have demonstrated that addition of a CEBPA mutation actually sensitized the cells to AraC. This would fit with a model in which cells that are more AML-like with a higher replicative rate are more sensitive to standard chemotherapy. This may be different with hypomethylating agents, which seem to have less of a direct anti-proliferative effect. It might be interesting to examine the effects of a chemotherapeutic such as AraC in this system and determine whether there are differential effects of the two agents.

-Line 399-401: It seems odd that the higher risk cells would have a lower proliferative rate given that, in general, higher risk MDS and AML are thought to have more rapid proliferation.

-Discussion line 536-539: I'm not sure that the data described in this manuscript inform the biology of de novo AML with CEBPA mutations. That is a disease setting typically without pre-existing mutations, where the CEBPA mutations are likely founding events, and is distinct due to its high response rates to classical chemotherapy agents. This is very different from sAML or high-risk MDS with a CEBPA mutation occurring at progression.

Minor points:

-Introduction line 81: "that the majority of mutations are randomly acquired and not related to the pathogenesis of MDS..." I'm not sure what "not related to the pathogenesis of MDS" means here. These are initiating mutations for MDS and AML.

-Line 125: I would avoid using dates (even month and year), and instead report dates as years (or days) from diagnosis of low-risk MDS.

Reviewer #3

(Remarks to the Author)

In this report, Alghrabi et al, engineer a CEBPA bZip mutation in iPSC clones reprogrammed from a low-risk MDS patient to mimic mutation acquisition in progression to high-risk MDS. The Authors show that the CEBPA mutation reconfigures the distal regulatory landscape of reprogrammed cells and impairs granulocytic differentiation, promoting self-renewal of erythroblasts. The Authors also perform scRNA-seq and identify MDS and MDS-CEBPA-bZip-mutant clusters, which they interpret as recapitulating clonal progression. Finally, the Authors demonstrate paradoxical effects of AZA treatment, which may explain treatment unresponsiveness / resistance.

While the model is elegant and has the potential to provide information on the mechanistic effects of the CEBPA bZip mutation in disease progression, it is unclear how it effectively reflects cellular and molecular characteristics of the patient.

A number of the claims are potentially overstated. For example, while the Authors suggest that the model demonstrates the participation of the CEBPA bZIP mutation in MDS progression, this would be more convincingly demonstrated through in vivo testing in transplantation experiments. It is also not clear to what extent the transient increase in BFU-E self-renewal reflects the behaviour of the mutation in patients. Equally, while the Authors claim that BFU-E replatability in engineered clones indicates that the self-renewal effects of the CEBPA bZip mutation require a background of other mutations, this is never formally demonstrated. One possibility would be to establish an inducible system coupling CEBPA KD with expression of the mutant form in the original reprogrammed MDS-low iPSC cells.

MAJOR POINTS

1. The Authors explain the need to engineer a CEBPA bZip mutation in patient-derived low risk iPS clones with the inability to reprogram the progressed MDS samples. They confirm the presence of the heterozygous mutation and exclude aneuploidy. However, it would be reassuring to exclude the presence of additional off-target heterozygous or homozygous mutations that may confound the conclusions. Did the Authors perform WGS or exome-sequencing of the clones obtained? And is this information available for the progressed MDS samples from the patient?
2. The low-risk MDS clones from the patient capture characteristic aspects of erythroid dysplasia without affecting progression of erythroid differentiation. They also capture a small but significant reduction of CFC frequency. In contrast, MDS-high risk cells carrying the heterozygous CEBPA bZip mutation have low clonogenicity with impaired myeloid differentiation, but display enhanced erythroblast self-renewal. How does this reflect the clonogenicity and lineage affiliation of patient MDS cells, before and after putative CEBPA bZip-mutant -driven progression? And how do the Authors link these changes with the risk of progression to AML, which is not characteristically erythroblastic? Comparisons with patient material are important to establish the relevance of the iPS model and should be included. Testing the differentiation and transformation potential of the reprogrammed cells in vivo through transplantation would also add confidence to the model.
3. Treatment of progressed high-risk iPS clones with AZA results in progression of the phenotype and potentially explain clinical resistance to the treatment. How does this compare with the response of the patient cells to AZA treatment? Is there evidence of similar paradoxical responses from patient material?
4. Similar to the previous comments, it is not clear how the transcriptional programmes specifically-associated with MDS-low risk and MDS-high risk iPS-derived clones compare with the progression of transcriptional programmes observed in this, or other patients carrying the same progression of mutations. Does the single-cell RNA-seq analysis suggest any combination of surface markers that would allow prospective isolation of MDS cells for analysis of disease progression in vitro or in vivo?

MINOR POINTS

1. Erythroid differentiation of iPS cells does not typically produce adult erythropoiesis. How does this affect the interpretation of erythroid clonogenic and differentiation results? Would the effects of the CEBPA bZip mutation be different in adult haematopoietic cells?
2. The differential high-risk vs low-risk reprogrammed MDS programmes include signatures which are not obviously haematopoietic and might translate ectopic expression or incompletely repression of alternative lineages during iPS differentiation. How does that reflect non-haematopoietic effects of CEBPA disruption? Are any of these programmes observed in patients or are they potential confounders?
3. The Authors represent CFC frequency analysis as relative numbers to control rather than absolute comparisons? How does this affect the results? Are similar trends observed? CFC frequencies in the clones have large error bars; is the same true of control and does it affect statistical significance if absolute rather than relative frequencies are represented? While I recognise the variability of these assays, it is important to show absolute values for full interpretation of data trends. I would also recommend showing individual data points in addition to the error bars in all analyses.
4. The Authors should be careful in referring to the MDS-specific clusters in single-cell RNA-seq analysis as reflecting clonal progression. The reprogrammed / engineered MDS iPS cells are clonal in nature, but the programmes associated with low-risk and high-risk MDS-derived clones do not necessarily reflect clonal progression, and may highlight differential lineage affiliation and differentiation progression in the iPS model. In the absence of clonal tracing studies / comparison with patient material / isolation of cells corresponding to individual clusters and transformation analysis representation of clonal progression is speculative.

Version 1:

Reviewer comments:

Reviewer #1

(Remarks to the Author)

The authors have in the revised manuscript included new data as well as provided detailed point-by-point responses that have clarified some concerns and unclarity within the originally submitted manuscript. However, in some instances the new data have raised new questions and still many important concerns raised in the first review remain unanswered as further detailed below.

Comments:

1. As also brought up by reviewer #2, the study is still exclusively performed on iPSCs derived from one MDS patient which limits the extent of potential conclusions that can be drawn from these studies. As the authors now themselves bring up in their point-by-point responses, genetic heterogeneity including the order and type of acquired somatic driver mutations as well as the inherited genetic background of the patient, can influence the effect of sequentially acquired driver mutations have on iPSCs generation and differentiation. In light of this context dependent impact, it is unclear whether or not the

inferences observed upon acquisition of the CEPBa bZip mutation are specific for this patient in the context of this particular genetic background or applicable to other MDS patients with a different genetic background and MDS composition. This limits the breath of the conclusions of this study as the authors only have investigated this in the context of cells containing RUNX1 G127fs and SRSF2 P95H mutations.

2. The authors have included new experiments where they compare data observed in iPSCs to data generated from patient MDS27 BM cells collected at different stages of the disease. However, this data raises questions to what extent conclusions observed in iPSCs can be linked to the impact on primary cells. This is in part highlighted by the ATAC seq data where 5-times more enriched or depleted peaks were detected when comparing low and high risk MDS cells from the patient versus the iPSCs. Furthermore, the peaks that are low-risk specific or high-risk specific in the primary patient cells are different when compared to those observed in iPSC. For example, PU.1 motifs which were found to be low risk specific in the iPSCs are high risk specific in the patient. Discrepant distribution is similarly observed for ETS, AP-1, CEBP and RUNX motifs. Functionally, the patient cells behave differently when compared to the iPSCs as well. This is striking when comparing the data in Figure 5E to 5H where no difference in total CFUs was observed when comparing low-risk to high-risk iPSCs. In contrast, the 150 CFU's observed for the patient's low-risk cells was reduced to almost 0 for the patient's high-risk cells. Furthermore, replating in normoxia was not reproduced with the patient's bone marrow cells, this was only observed in hypoxia which also only was tested for the high risk cells and lacks data from low risk for comparison. Finally, the authors provide representative images of similarities in aberrant morphology. Although this is quantified in the iPSC generated cells, this is not the case for patient cells.

3. The authors have provided an informative description on the challenges with performing in vivo studies of human iPSC-derived HSPCs. Considering this, they have instead identified an interesting approach through the use of scaffolds and now can successfully engraft iPSC-derived cells in immune-compromised mice. However, the interpretation and analysis of this data is unclear. It appears in Figure 2B that the high-risk iPSCs have lower engraftment potential than control and low-risk iPSCs but this is based on total human CD45 engraftment in the scaffolds. As the most striking impact of high-risk iPSCs was observed in their BFU-E serial replating potential, it is surprising that the authors have chosen to focus their analysis exclusively on human CD45 expressing cells, and not cells of the erythroid lineage that have down-regulated or fully lack CD45 expression. This has previously been critical for the identification of in vivo engrafting MDS cells when the phenotype is propagated through the erythroid lineage.

4. The authors use the observation of aberrant morphology during erythroid differentiation (Fig 1J and 5D) to support the iPSC relevance for MDS studies. However, the aberrant morphology is observed at the earliest stages of culture and declines throughout erythroid maturation. The authors do not show data for day 0 for this morphology which could be important towards the conclusion of the authors as this could indicate that the mutations also impact the generation of the hematopoietic progenitor stages, which so far the authors have claimed are normal. This is also relevant for the interpretation of all functional assays, as it could mean that the patient iPSCs are at an earlier or later progenitor stage already prior to seeding in myeloid or erythroid differentiation assays. Inclusion of day 0 timepoint would be important for supplemental figure 6A as well. Regarding aberrant morphology, data in Figure 5D shows no differences between low-risk and high-risk iPSCs, and low-risk iPSCs are also significantly different from normal iPSCs.

5. The authors have provided reasonable comments to the RNA sequencing data. However, this could benefit from more focused analysis. For example, by performing more cluster dependent analysis for comparing low to high-risk MDS HPCs the authors could reveal shared gene expression changes when comparing these two stages in distinct clusters (e.g. cluster 1, 3, 7, 8 where there are sufficient cells in both). In addition, the relevance of the pseudotime analysis in figure 7I is questionable, particularly since the authors have pooled cells from low risk HPC, low risk erythroid cells, and high risk HPCs.

6. Page 5, second paragraph, sentence starting with "These findings suggest that cells harbouring ...". The authors refers to three mutations when there are only two reported in Table 1. The authors also state that RUNX1 and SRSF2 were the predominant clone in the patient which is not supported by a VAF of 15% in the patient (minor clone).

7. In Table 1, sample from 2015 is shown twice. There are also several rows with no text in Table.

8. The title of the second result section states that the low-risk iPSCs have normal hematopoietic progenitor potential but this is not in line with the results showing lower CFC potential. In the text describing this data, the authors only refer to lower myeloid CFCs, when in fact also erythroid CFCs are significantly reduced (Fig 1C). The authors also claim an "evident bias" toward erythroid CFC but this is observed for normal iPSCs as well.

9. In reference to Figure 1G the authors state that the expression of CD235a was "very low" at early stages of differentiation. However, when comparing early timepoints (day 0-7) to late timepoints (day 11-18) the maximum CD235a expression level is observed in the early timepoints (in particular day 7).

10. The quantification in Figure 1H does not seem to be correct according to labelling of the graphs. For example CD71+ cells represent close to 50% of the cells at day 4 (not 25% as shown in the graph).

11. The authors show the same data several times in their figures which seems unnecessary as the authors themselves in their point-by-point responses claim that the manuscript have many figures. As an example, the authors show the same CFC data in three different figures. For the same reason, Figure 2 should only focus on the data from the CRISPR control line and high-risk lines, without inclusion of the lines shown in Figure 1.

12. In reference to Figure 2, the authors claim high-risk iPSCs mainly affected the myeloid lineage, when in fact it only affected the myeloid lineage and erythroid CFCs were preserved similar to low-risk (Fig 2C-E).

13. The authors states in their responses that statistical significance is shown only for those reaching a significant p-value but this does not seem to be consistent throughout the manuscript where there are figures showing ns and also significant values not included. Just as an example, the significant p-value comparing normal iPSC to low risk iPSCs in Figure 2C is not included. Although the authors have improved some of the of documentation for their figures, they can improve this further, especially in the figure legends to enhance clarity to which comparisons are being made etc. Furthermore, the authors have included representative isotype control but this appears to be provided by only one timepoint. To document their analysis, an isotype for each timepoint should be provided. This is particularly critical for in vitro cultures supporting maturation of myeloid cells which are known to upregulate Fc-receptors that can facilitate unspecific antibody staining.

14. The authors make reference to Figure 2I when stating results from a fourth replating but there is not data in Figure 2I on

this.

15. In Figure 3B and 3D the same BFU-E data is shown with different significance results. This should be clarified.

16. The authors make general statement that high-risk iPSC have disrupted myeloid differentiation. However, based on the data shown in Figure 4C an alternative interpretation is that high-risk iPSCs fail to generate mature granulocytes which rather is replaced by generation of mature monocytes. This does not seem to be considered or discussed by the authors.

How does this enhanced generation of monocytes in high-risk iPSC align with the molecular data and data from the patient?

17. With regard to Figure 4G where the authors have included additional data in the point-by-point responses, this reviewer is still not able to interpret the data shown. As stated in the previous review, this data be better to display as a deltaCT values where expression relative the housekeeping gene is shown. The authors state that the left figure in the point-by-point responses show delta-delta CT values but this does not have any indication for what adjustment has been done considering that delta-delta Ct refers to adjustment to a housekeeping gene and a reference population.

18. In supplemental Figure 8B what does HSC refer to?

19. Figure legend 1G states that this shows a dot plot which is not correct as this is a contour plot.

20. What does HPC represent in Figure 1B?

21. The authors reference Table 2.5, 2.6, 2.7 and 2.8 in the manuscript but these are not provided in the review documents.

Reviewer #2

(Remarks to the Author)

I appreciate that the authors have so carefully addressed my concerns and points. While they were not able to directly compare the transcriptome of primary high-risk MDS cells and their reprogrammed cells, they do show that phenotypically their model recapitulates the primary cells quite well, and I think it was probably outside the scope of this work to do more here. Overall, this is very nice work and I have no further significant concerns. I congratulate the authors on their manuscript.

There is a small typo on page 14 from discussion: "...highlighting that the complex genetic background of high-risk and AML patients dictates...". The word MDS appears to be missing after high-risk in this sentence.

Reviewer #3

(Remarks to the Author)

I am satisfied that the Authors have addressed the points raised during my initial review, and I acknowledge the significant amount of work put into this revision.

I would just ask the Authors to review the final section of the Results and abstain from using terms like clonal composition and emergence of new clones when referring to their iPSC model. Instead, they could state that they observe a change in cellular composition and differentiation hierarchies when comparing low-risk and high-risk isogenic iPSC models, which reflect the changes in clonal composition and disease progression observed in patients.

Version 2:

Reviewer comments:

Reviewer #1

(Remarks to the Author)

In the revised manuscript by Almaghrabi and colleagues the authors have clarified many of the points raised to the previous version. The inclusion of additional data and new adjustments to the text has resulted in an improved manuscript. In particular, the adjustments to indicate that the work was done within the context of the SRSF2/RUNX1 mutational background is well appreciated. Although most points raised to the previous version are addressed either in the manuscript or through clarifications in the point-by-point responses submitted by the authors, there are still some remaining issues that remain unclear or where additional analysis of existing data would be important.

Presentation and description of ATAC-seq data

In the responses the authors state that the patient data from the low-risk stage represent only 5% diseased CD34+ cells, where the rest are healthy. This statement is surprising considering no wildtype iPSC could be derived from the low-risk stage and the variant allele frequencies indicate that 60% of the cells are RUNX1 mutated and 30% of the cells also contain the SRSF2 mutation (based on Table 1). At the high risk stage the VAF has increased to almost clonal levels, where close to 80% of the cells carry the three mutations (RUNX1/SRSF2/CEBPA). Does this mean that the CD34+ cells used for ATAC-seq have much lower VAF compared to the peripheral blood cells used for iPSC generation? If so, the VAF of the CD34+ cells should be shown as a separate figure/table.

Secondly, in both iPSC and CD34+ cells the transition from low-risk to high-risk is defined by the acquisition of the CEBPA mutation. Although there are many reasons for why the many up- or down-regulated ATAC peaks detected in the patients CD34+ were not found in the iPSC, one would expect that the majority of the peaks seen in the iPSCs also would be observed in the patient's CD34+ cells. To efficiently show this, the authors should include two Wenn-diagrams, one for the gained and one for the lost ATAC sites, to indicate shared differentially accessible sites among patient and iPSC during disease progression. If the motif analysis on these shared sites demonstrated enrichment for GATA, AP-1 and RUNX1 sites, this would significantly strengthen the authors conclusion.

Lastly, it is not fully clear what the authors mean with “trajectory during disease progression” when they summarize these results. An alternative way to phrase these observation is: “In summary, our data shows shared alterations in the chromatin pattern during disease progression in iPSC-derived and patient cells” (assuming this is supported by the Wenn-diagram of shared peaks).

Interpretation of patient CFC replating data

The experimental design for this experiment does not exclude that the replating potential seen for high-risk patient cells (Figure 5H) is due to initial culture in hypoxia prior to replating in hypoxia. As initial culture in hypoxia was not performed for low-risk cells, one can therefore not exclude that also low-risk cells could have replating potential. To describe these results, I have the following suggestions:

Line 425, change to: “Although not performed with low risk iPSCs, under hypoxia (5% CO₂), CD34+ cells from MDS27 high risk were capable of forming colonies and the number

Line 428, remove “only” resulting in: “Additionally, high risk cells were able to form colonies after second replating

Analysis of RNA-seq data

The previous comments raised towards the RNA sequencing analysis remains as this could reveal important and critical dysregulated genes within more homogenous compartments shared between the two stages of disease. The authors’ argument for the limitation in cell numbers based on the published analysis by Liu et al. in BMC Genomics 2023 is not in line with what Liu et al state: “If a study anticipates DEGs with modest differences, the study should aim for having 2,000 or more cells in a cluster in order to identify the majority of DEGs that would have been identified by a bulk RNA-seq analysis of thousands of physically purified cells. Such studies should be cautious in interpreting a lack of DEGs from clusters with fewer than 100 cells. On the other hand, clusters with as few as 50–100 cells may be sufficient for identifying the majority of DEGs that would have extremely small p values or transcript abundance greater than a few hundred TPM in a bulk RNA-seq analysis.”

What Liu et al argues is that meaningful analysis can be done with as few as 50-100 cells per cluster, with the caveat that this is not expected to pick up all differentially expressed genes, but it will pick up the most significant DEGs and/or those with the highest fold difference in fold expression. The requirement for 2000 cells or more is indicated for studies where the intention is to identify all DEGs including those with very modest changes.

Minor point:

Line 179: To avoid confusion, I suggest that the title is change to “.....retain myeloid and erythroid progenitor potential” instead of “.....display normal hematopoietic progenitor potential”.

Version 3:

Reviewer comments:

Reviewer #1

(Remarks to the Author)

Almaghrabi et al. have provided further clarification in the response letter and included new data in the manuscript that addresses most issues raised in the previous review. In particular the new RNA sequencing analysis has provided further strength and information to the identification of candidate genes impacted by the acquisition of the CEBPa mutation. However, the authors have chosen not to follow the suggested analysis for the ATAC sequencing analysis to provide more insight towards the degree for how molecular changes observed in iPSCs are recapitulated in the corresponding patient cells. As the authors argues that “culture conditions, cytokines, environment, oxygen levels and niche components have profound effect on chromatin landscape” thereby themselves raising major questions towards the relevance of performing ATAC sequencing analysis if the goal is to use the iPSC platform for identification of molecular changes of relevance to disease biology in the leukemic cells in the patient. The above argument provided by the authors further does not fit with their following responses in the manuscript where they state “we find the 30% similarities in the cis-environment quite remarkable given the differences between in vitro and in vivo conditions.” As I rather agree with the last argument by the authors, I therefore still strongly believe that the proposed analysis would be informative, important and further strengthen the authors data, as well as the readers interpretation of the data. The proposed added figure also does not require single cell analysis as my question is quite simple: how many of the 1691 peaks significantly enriched in high risk and 1576 peaks significantly enriched in low-risk iPSC (figure 6a) were also significantly enriched in the corresponding CD34+ high-risk (among the 8290 sites) and low-risk (among the 7795 sites) cells from the patient (Figure 6F), respectively. Secondly, if the authors perform motif analysis on the shared sites altered upon progression from low- to high-risk, do they still recover the AP-1, RUNX1 and other highlighted motifs?

Minor comments:

- Line 59: based on the concerns the authors themselves raise in the rebuttal letter regarding the impact of culture conditions, cytokine etc, on iPSC, the authors should remove “faithfully”.

- Lines 136-137: remove “affecting their expression” as the expression level of these genes is now show in the figures

- Lines 626-628: Dependent on how the authors respond to the suggested ATAC analysis, and what that analysis shows, the statement starting with “Importantly, a similar”. One potential alternative here is to replace the sentence with: “The

same motif families were enriched in CD34+ cells from iPSCs and the patient following disease progression, suggesting shared pattern of chromatin reorganization upon acquisition of the CEBPa mutation”.

- Line 1273: The indicated fold change cut-off is wrong.

Reviewer #1 (Remarks to the Author):

Almaghrabi et al. established iPSCs from one single MDS patient who over a 2-year period progressed from low-risk MDS to high-risk MDS and aimed to apply the generated iPSCs to cellularly and molecularly model MDS disease progression associated with acquisition of a mutation predicted to disrupt the CEBPA bZIP domain. Using several in vitro culture assays including replating capacity and lineage-differentiation capacity, epigenetic profiling and single cell RNA sequencing, they conclude that their system model clonal evolution of MDS, where acquisition of a CEBPA mutation is the main driver for disease progression, in part through restructure of the chromatin landscape. Furthermore, the authors claim to demonstrate changes in the clonal composition as a consequence of CEBPA mutations through their single cell gene expression analysis. While it remains important and novel to understand some of the fundamental questions raised related to MDS disease progression and therapy resistance, there are several limitations with the approaches used by the authors to address these and therefore these studies fail to provide convincing new evidence towards the conclusions listed above. As further detailed below, the modelling of MDS disease progression using iPSCs has already been performed by other studies, there already is convincing data in the literature with regard to the transformation potential of mutant CEBPA, the study is applied only to one iPSC line and without comparison to the patient's primary cells, and the authors do not model the specific mutation found in the patient.

We thank the reviewer for his/her comments which are answered in detailed below.

Major comments:

1. Reviewer1: A major limitation with this study is that it is restricted to the mutations found in one single MDS patient. For comparison, a recent study reported the generation of iPSCs from 15 AML cases, several who had progressed from MDS to AML and including one line which contained a late-occurring CEBPA mutation (Kotini et al., Blood Discovery, 2023). In contrast to Almaghrabi et al., Kotini et al. were able to generate iPSCs directly from the patient resulting in iPSC clones without CEBPA (but including all mutations acquired prior to CEBPA) and iPSC clones with CEBPA.

We would like to clarify that the published work the reviewer refers to is different to ours as the iPSC generated in Kotini et al., Blood Discovery, 2023 are not paired isogenic samples. In this paper the authors described the successful generation of iPSC from 15AML patients out of 32 samples they have tried over the years. In previous work from the same group (Wang et al Cell Stem Cell, 2021) the authors generated iPSC from patient sample AML32 containing a CEBPA mutation (H24fs*84). In the paper from 2023, they described a lack of success on generating iPSC from another patient sample harbouring CEBPA mutation F31fs*130 (AML34). Thus, the same group has tried to get iPSC from two different AML patient samples harbouring different CEBPA mutations and only one of them was successful.

The mutational background between those two samples was different: AML32 patient had TET2, IDH2, ASXL1 and SRSF2 mutations and AML34 had TET2, STAG2, SRSF2 and CSF3R, pointing to the possibility that the genetic background of the patient dictates the ability of generating iPSC. We decided to re-create iPSCs with a much more restricted mutational spectrum to be able to understand what each mutation does.

We have included this point in the discussion:

*It has previously been reported the successful generation of iPSC from an AML patient containing a CEBPA mutation (H24fs*84) in combination with TET2, IDH2, ASXL1 and SRSF2 mutations¹⁰ but not in an AML patient containing CEBPA mutation F31fs*130 as well as mutations in TET2, STAG2, SRSF2 and*

CSF3R⁵⁶. Thus, our lack of success in generating iPSC from the patient after disease progression harbouring a mutation disrupting the CEBPA bZIP domain (Gly257fs), likely points that a particular genetic background is incompatible with the ability to generate iPSC.

1 (cont.) Reviewer1: In those studies no differences in hematopoietic function, including ability to engraft NSG mice, was observed between the two isogenic clones. This at minimum raises questions towards the study by Almaghrabi et al. as they applied CRISPR to molecularly engineer in a CEBPA mutation, which in fact generated a different mutated sequence compared to the mutation observed in the MDS patient at the high- risk stage.

In the studies that the reviewer refers to, iPSCs were generated from patients already diagnosed with an established AML. We would like to clarify that our work aimed to model the evolution of low to high-risk disease. We started with low risk MDS patient samples harbouring mutations in two genes (SRSF2 and RUNX1) which were not sufficient for transformation at that stage. We then modelled disease progression by the acquisition of the CEBPA mutation seen in the patient. Our CRISPR initiated change resulted in the same functional outcome due to a frameshift mutation that disrupts the bZIP domain. Equally, not all patient samples harbour the same mutation, but there are hotspots on the gene where mutations will lead to the same changes in functionality.

In vivo engraftment of our HPCs (healthy and MDS) was not possible by intravenous injection, as reported by others (groups of Eirini Papapetrou, and George Dayley (Kotini et al CSC, 2017; Vo and Dayley, Blood 2015). Nonetheless, using humanised scaffolds (Mian et al Blood Cancer Discovery 2021), we could show positive engraftment and no difference in the ability to engraft.

1 (cont.) Reviewer 1: In light of the concerns raised about the side-effects associated with CRISPR, including potential introduction of new genetic lesions as a consequence of low p53 expression (Haapaniemi et al., Nature Medicine, 2018), also raises the possibility that the effect observed by the authors could be mediated by other genes than CEBPA.

We understand the concerns of the reviewer. We therefore generated several clones and the chances of the two of them harbouring the same mutation due to possible side effects during CRISPR editing is quite unlikely. Two additional sets of experiments exclude this possibility: (i) the same CRISPR CEBPA construct in a wild type background did not have the same effect than in the low risk clones (Figure 3 A-F), and (ii) the effect that the CEBPA CRISPR mutation has in the low risk clones could be reverted after repairing the RUNX1 and SRSF2 mutations (Figure 3G-M).

2.Reviewer 1: A second concern throughout the whole study is that iPSCs from one single MDS patient are not compared to their isogeneic wildtype cells lacking the oncogenic mutations observed in the MDS patients, but instead compared to one single iPSC line generated from another, presumably healthy individual. This limits the strength of the conclusions in this study, both towards functional in vitro assays and molecular profiling, where the differences observed could in part potentially be explained by inherited genetic heterogeneity rather than somatically acquired mutations associated with the MDS.

Our work was aimed at generating mechanistic insights in the same way than studying a knock-out mouse, which has a phenotype that may or may not be modified by the genetic background. As the reviewer is aware it is not always possible to get isogenic wild type lines from a patient. This is also observed by Kotini et al, in the Blood Discovery paper from 2023, in which from the 15 AML lines that they were able to generate iPSC from, isogenic wild type iPSC lines were not generated in 8 of them.

We agree with the reviewer that the fact of not having isogenic wild type lines makes it difficult to make good comparisons for changes from a wild type situation to a low risk. However, that was not the aim of our study. Our aim was to study disease progression. We used the wild type line as a positive control for pluripotency staining, alkaline phosphatase and an approximation of the percentage of HPCs and colonies that we could observe following the differentiation protocol. Similar percentages have been also published elsewhere.

We can safely presume that the wild type iPSC line used came from a healthy individual with no known chromosomal abnormalities or issues with structural integrity: this wild type cell line, BU3.10, is commercially available at Wicell (<https://www.wicell.org/home/stem-cells/about-our-cell-lines/our-cell-lines.cmsx>). Comparisons of the molecular profiling studies were done between the two isogenic lines generated (low risk vs high risk) and thus conclusions are valid.

2 (cont.) Reviewer 1: In addition to this, the authors focus most of their assays on only one single iPSC clone (clone 22).

For the majority of the studies, we used three different clones (clone 8, clone 11 and clone 22), including iPSC characterization, timecourse of hematopoietic differentiation and colony assays. Once we determine that they behave in the same way, we chose one of them (Clone 22) to perform CRISPR (control and CEBPA), and followed two clones for CEBPA for all functional assays, which also show the similarity between both clones. scRNAseq was done with one representative clone of each due to the cost of the flowcell.

3. reviewer 1: The role of CEBPA towards leukemic transformation is well established from genetic modelling in mice (Heath et al., Blood, 2004; Zhang et al., Immunity, 2004; Kirstetter et al., Cancer Cell, 2008; Bereschenko et al., Cancer Cell, 2009). Combined with the extensive genetic characterization of human hematologic malignancies, including MDS and AML, demonstrating the recurrent nature of CEBPA mutations, predominately associated as a late event and associated with disease progression, already provides strong evidence towards the role of CEBPA in leukemogenesis. However, CEBPA mutations come in different flavors, heterozygous and homozygous mutated states and variation in location and type of mutation, where some are associated with different clinical outcome. In light of this, novelty would be enhanced if one could model distinct CEBPA mutants and investigate their cellular and molecular impact against each other.

We agree with the reviewer that this would be a fascinating area of research. Given the potential that iPSC brings, is an area that we would like to pursue in the future. We think that it is out of the scope of this manuscript.

4. Reviewer 1: In addition to the acquisition of a CEBPA mutation during progression, progression was also associated with a dramatic clonal expansion of a sub-clonal RUNX1 frameshift mutation (15% to 47%), which combined with the already clonal L56 hotspot RUNX1 mutation (50% at both low and high risk stage), generates a bi-allelic RUNX1 mutated state. Considering the important role of RUNX1 in normal and leukemic hematopoietic stem and progenitor cell biology, this represents an interesting collaborator towards potentially enhancing the phenotype observed together with the CEBPA mutant as RUNX1 deletion has been previously shown to reduce Cebpa transcript and result in reduced granulopoiesis (Guo et al., Blood, 2012). This raises potentially interesting mechanistic questions including whether the phenotype observed of the heterozygous CEBPA mutant in the assays used by the authors is dependent on RUNX1 mediated suppression of the CEBPA wildtype allele. Something which could be further investigated by Crispr restoration of one or both RUNX1 alleles to wildtype configuration.

We would like to clarify that the L56 RUNX1 mutation is a benign missense mutation, and it does not have an impact on the disease; it is generally not reported (see also commented by reviewer 2). Thus, it is not considered a bi-allelic RUNX1 mutated state.

As mentioned above, we have performed a CRISPR-Cas9 to restore the original mutations (SRSF2 and RUNX1) in the high-risk iPSC line to help to clarify whether the frameshift mutations found in two transcription factors important for myeloid differentiation RUNX1 and CEBPA are responsible for the disease progression. Our data shows that reverting RUNX1 and SRSF2 mutations did not have an effect on the number and percentages of HPCs. In clonogenic assays, the erythroid differentiation bias that occurred in the presence of both RUNX1 and CEBPA mutations was ameliorated. Moreover, high risk cells, with the C/EBPA mutation and reverted RUNX1/SRSF2 mutation, lose their repopulation capacity. These data in combination with our previous data showing that the sole mutation of CEBPA in a wild type background did not grant a repopulation capacity provides further evidences that the self-renewal capacity is conferred in the context of RUNX1 and SRSF2 mutations. All these data have been combined in New Figure 3.

We also aimed to generate a RUNX1Gly127fs mutation in our wild type iPSC. After two attempts we were only able to generate one clone harbouring homozygous mutation for RUNX1 Gly217fs. Hematopoietic differentiation of this clone was severely impaired with 20-30x less HPCs numbers at Day 14. Also, a reduction in the number of BFU-E colonies was observed after colony assays and the percentage of erythroid cells CD71+/CD235a+ was practically null. No serial replating capacity was observed (see data on page 16, answer to Reviewer 2). As this mutation was homozygous and not heterozygous as found on the patient, we have not included the data in the manuscript.

5. Reviewer 1: An important limitation for the conclusions that the authors make towards the relevance of their iPSCs at modelling the disease progression and therapy-resistance observed in the patient is the obvious lack of direct comparison to the primary cells from the actual patient at low-risk and high-risk stage of disease. Without this, the authors claims to reproducing the clinical phenotype and relevance of iPSC clones are weak.

We understand the concerns of the reviewer and agree that this is a vital point. Fortunately, we had the clinical data of the patient and primary cells left. We have now included in Supplementary Figure 1A-D the patient clinical data including: Giemsa staining of blood smears, CD34 staining on bone marrow trephine, bar-graphs displaying the percentage of CD34+ and early erythroid cells during the course of the disease (see below). Also, we have included several panels in the main figures: In new Figure 1K we have highlighted that the erythro-dysplasia observed in our iPSC is also present in the patient sample.

In new Figure 4F we show that the hypogranularity and ring-nuclei present in our iPSC is also a characteristic found in the MDS patient.

In New Figure 5H and Supplementary Figure 7D and 7E we show the clonogenic and differentiation potential of both low risk and high risk sorted CD34+ cells from the patient, showing similarities to our iPSC cells (lower clonogenic potential in high risk and increase erythroid potential), as well as repopulation capacity on the high risk sample.

In new Figure 6G-H, we show the ATAC-seq analysis we have performed in CD34+ cells from the patient before and after disease progression to highlight the similarities at the molecular level between the primary cells and the iPSC generated.

6. Reviewer 1. The authors focus their molecular analysis on the cells that they isolate from the hematopoietic differentiation protocol. As shown by the single cell gene expression analysis this is a highly heterogeneous population, where only 2% contain ability to form colonies in methylcellulose, thereby representing functional hematopoietic stem/progenitor cells. In light of the differences in CFUs, differences in preferential differentiation, and level of expression of critical markers such as CD34 (Suppl Fig. 1B) raises concerns about the cause and effect conclusions by the authors towards the role of CEBPA. The ATAC seq analysis therefore does not distinguish between whether the differences in peaks are due to CEBPA or represent differences in cellular composition of the cells analyzed. To resolve this, such analysis would require much more purified cells and/or single cell analysis.

We would like to clarify that the ATAC-seq data was performed in sorted CD34⁺/CD45⁺ cells and not from the bulk cultures coming from the stemdiff differentiation protocol. Additional profiling of the cells was performed using CD11b and CD71 surface markers, which indicated that the CD34/CD45 cells isolated were not biased towards one of the lineages in low vs high risk samples.

7. Reviewer 1: With exception of the CFU assays, the authors make conclusions from in vitro differentiation cultures primarily based on the composition of cells without taking into account the total number of cells within the culture. Instead of presenting the percentage of cells in the culture, the author need to rather convert these to absolute number of cells in order to allow interpretation of the results and to document the experiment variation.

We understand the concerns of the reviewer. Total number of HPCs cells was similar in all cases. This data has been included in New Figure 1B and New Figure 2A.

8. Reviewer 1: The authors should provide better documentation of the FACS analysis shown in the manuscript, including clone name of antibodies, utilization of viability markers, etc. This data has been provided in the materials and methods section.

8 (Cont.) Reviewer 1: Furthermore, the profiles shown for several of the differentiation cultures should be further validated to ensure the gates used for distinguishing distinct cell stages and cell types are as intended. For the erythroid culture, it is surprising to see what appears to be down-regulation of CD235a (Figure 2E) in the late stages of the culture, this is particularly evident in the transition from day 7 to day 11 when the distinct CD71/CD235a double positive population is exhausted from the culture. It would be important to see cytopins with hematology stains from the CD71-CD235alow cells and the residual CD71lowCD235ahigh cells to demonstrate their morphology

We would like to clarify that our data does not point to an exhaustion of the CD71/CD235a double positive population in culture but a maturation of the cells as it can be observed on the cytopins for each of the time points included in Supplementary Figure 2G (mature cells are indicated with yellow arrows).

As requested by the reviewer, a cytopsin has been performed for the aforementioned regions after sorting (New Supplementary Figure 2F).

8 (cont.) Reviewer 1: Similar issues towards the placement of the gates to identify distinct cell types apply to Figure 3E and 4B for the markers applied there.

Figure 3E and 4B gating is performed according to isotype controls. Graphs are provided in New Supplementary Figure 2E.

8 (cont.) Reviewer 1: Furthermore, the authors have in their methods indicated that DRAQ5 was included but does not show the gating strategy for this.

We apologise for the inclusion of DRAQ5 in the text. We indeed performed a DRAQ5 staining and FACS analysis to determine the enucleated rate during erythroid differentiation, but we felt that due to the fragility of the culture at this stage, many of these events could be fragmented cells and not truly mature erythroid cells. As the results obtained could have been misleading, we decided not include these data in the manuscript. We have removed DRAQ5 from the methods section.

9. Reviewer 1: As the authors were unable to establish iPSCs from the high-risk stage containing the CEBPA mutation, they genetically engineered iPSCs to incorporate a CEBPA mutant. Following genetic engineering of genes, without functionally validating the impact on the protein and specific function, one can only make predictions about the impact the gene has on protein function. As the genetic change introduced by the authors is a frameshift mutation, with intact sequence prior to the mutation location, this leaves the potential for retained or altered function of the protein that is unique to the specific mutation. In light of these, the authors should throughout text adjust their description of the CEBPA mutation to predicted changes.

In both instances, low risk MDS C22 and wild type iPSC line, the CEBPA mutation generated created a frameshift mutation leading to the disruption of the bZIP domain. In both cases, the functional consequences expected for a disruption of CEBPA bZIP domain were the same, including (i) downregulation of CEBPA, GATA2, PU.1 gene expression (Real time PCR Figure 4G), reducing myeloid differentiation, as observed for the high risk MDS27 clones (Figure 2C) and wild type CEBPA (Figure 3B); (ii) reduction of granulocytic formation not only in colony assays but also in liquid cultures, as observed for the high risk MDS27 clones (Figure 2E-G and Figure 4B-E) and wild type CEBPA (Figure 3C-3E); and (iii) formation of dysplastic myeloid cells, as observed for the high risk MDS27 clones (Figure 4D-E) and wild type CEBPA (Figure 3E).

Moreover, modelling of the effects of the patient and CRISPR-derived mutations using AlphaFold 3.0 shows that in each case the helical motif associated with dimerization and DNA-binding is lost and the C-terminal regions instead consist largely of random-coil structure.

We have changed the nomenclature from CEBPA mutant to CEBPA bZIP fs to denote the frameshift mutation of the bZIP domain and be more accurate as requested by the reviewer.

9 (cont.) Reviewer 1: Furthermore, the authors also compare the impact of the CEBPA mutant in iPSCs to CRISPR modified wildtype iPSCs. However, the specific mutation in the wildtype iPSCs is not documented, and if different from the MDS iPSCs limits the conclusions one can draw from this control as the effect of the individual mutations are not validated.

We understand the concerns of the reviewer. The specific target differs between the low risk and wild type lines and between both wild type lines. In the case of wild type C12 line, the deleted region changes the open reading frame producing the same C-terminal aminoacid sequence than the patient (New Supplementary Figure 5A), whilst the wild type C5 line the bZIP mutation differs to the rest. Still in all cases the disruption of the bZIP domain is generated and the functional consequences also are the same. Contrary to the reviewer, we believe that this strengthens the conclusions regarding the disruption of the bZIP domain. Please, also see above the predicted structures using AlphaFold 3.0.

9 (cont.) Reviewer 1: Finally, the documentation of the CEPBA mutations in Supplemental Figure 3 represent interpreted sequences and the authors should complement these with the raw sequencing traces.

The mutation generated was not a point mutation but deletion of 43 bp in one of the alleles, and thus it was clearer to present the data highlighting the region where deletion had occurred. As the reviewer can see by the dendrograms below, it is quite difficult to follow the sequence and thus not very informative. Nonetheless, as requested we have included the dendrograms as part of the raw data files (New Supplementary figure 3D).

We have also included the work in this other format (below) but presumably it falls under the same category of “interpreted sequence”, so we leave the decision on how best to represent it (Supplementary Figure 3D) to the discretion of the reviewer/editor.

10. Reviewer 1: The authors utilize the single cell RNA sequencing analysis (Figure 7) to make claims towards clonal evolution and loss of cellular identity. It is not clear what the authors mean with regard to clonal evolution as the data so far only suggest differences in cellular composition?

We thank the reviewer for this comment, he/she is correct. It is true that in this instance, we cannot talk about changes in clonal composition but changes in cellular composition. This has been corrected in the text.

10 (cont.) Reviewer 1: In addition to the limitations described above with regard to being derived from one single patient, crispr modified and one single iPSC clone, there are also limitations with the analysis of this single cell RNA sequencing experiment. To understand stage-specific impact of the mutation, it is important to compare as closely related compartments as possible, and the strategy applied by the authors runs the likely risk of representing an apples and oranges comparison. In addition to the method description being poorly documented in order to understand the parameters behind the analysis, such as number of variable genes used for clustering and annotation of cell type, the analysis would benefit from more targeted analysis of corresponding compartments.

We apologise for the omission of this methodology. The default parameter on Seurat (2000 variable genes) was used for clustering. The annotation of different populations in the control and more differentiated samples was performed based on manual curation based on gene expression clustering from publicly available datasets. The number of genes used for clustering annotation differs in each category (please see examples below). The scRNA seq data aimed to determine whether the disruption of the CEBPA bZIP domain leads to changes in cellular composition. As it is appreciated, this is the case, with different cell clusters expanding from low risk to high-risk.

GMP signature

Erythroid signature

10 (cont.) Reviewer 1: As demonstrated by Figure 4D, the most distinct clusters occupied by wildtype cells also are represented by MDS cells from both low and high risk stages. It would be more relevant to perform comparisons within distinct clusters in order to understand the impact of CEBPA in a stage-specific manner, but to do so the authors are required to run more cells to reach statistical power for the comparisons. Furthermore, the authors included wildtype and low-risk MDS iPSCs subjected to erythroid differentiation in the analysis but did not do this for the high risk MDS iPSCs. Finally, without

complementing the analysis on primary cells from the patient, as well as inclusion of more wildtype and MDS iPSCs from other patients, the conclusions from the authors are weak and speculative.

We would like to bring to attention that the most distinct clusters present in wild type cells (HPCs) are clusters 2 (3923 cells), 4 (362 cells) and 9 (1005 cells). The number of cells in these clusters are really low in MDS samples: the low-risk samples (394, 70 and 15 cells, respectively) and high-risk sample (36, 0 and 1 cell, respectively). The reviewer suggested to run more cells to reach statistical power for the comparisons. Even in the best scenario, to compare for example cells within cluster 2, we would require to run 10x more low risk cells and 100x more high risk cells, which is not affordable. Single cell data analysis was meant to determine changes in cellular composition and to do pseudotime analysis. Single cell RNAseq comes with its limitations when comparing specific clusters as in some samples these ones are misrepresented. As we could not perform comparison within distinct clusters, we performed pseudo-bulk comparisons instead. We included the erythroid differentiation analysis as a means of better identification of the clusters, but we can remove this data if required.

Previous work generated by the group of Papapetrou has revealed that RNAseq comparisons between HPCs-iPSC derived cells and the AML patient samples from which they derive are not possible, as iPSC are derived from a single cell. This would be even more difficult in the case of MDS samples as less than 20% of the cells of the patient will be malignant blasts.

We disagree with the reviewer regarding that our conclusions from the scRNAseq data are weak and speculative. Our conclusions from this analysis are (i) that the cells coming from MDS express genes associated with leukaemia. These genes have been previously reported and appropriate references are included, and (ii) that there are changes in cellular composition during disease progression, which is based on the actual numbers of cells present on each cluster.

11. Reviewer 1: In Figure 5E the authors culture wildtype and MDS iPSCs in presence of Aza. Surprisingly, Aza leads to increase in CFUs from the high-risk iPSCs and does not demonstrate any impact on the low-risk or wildtype iPSCs. Although potentially interesting, the interpretation of this finding is limited as the experiments lack a positive control in the form of iPSCs from a Aza-responsive MDS patient.

As the reviewer likely is aware, there are no genetic features associated with Aza response and any non-genetic determinant will be lost upon reprogramming. We have approached other researchers including Eirini Papapetrou, who doesn't have any lines with known information regarding Aza response of the patient. However, we know the patient's clinical history that they were resistant to Aza, with Aza treatment needed to be stopped after a sixth round as number of blasts reached 30% (New Supplementary Figure 1D).

Moreover, we performed colony assays of CD34+ cells from the patient before and after disease progression, and observed that under hypoxia (5%O₂), there is an increase in the number of CFUs after Aza treatment (New Figure 5H), strengthening our data.

12. Reviewer 1: Perhaps the most interesting data in the manuscript is the indication that high-risk MDS iPSCs confer potential to retain CFU potential following serial replating. Although this could support enhanced self-renewal, it would be more relevant to demonstrate this in a more rigid in vivo assay following transplantation into NSG mice, as reported for other iPSC lines modelling MDS and AML. Furthermore, did the authors investigate the impact of Aza on serial-replating capacity?

To this end, we have sought a collaboration with the group of Dominique Bonnet to perform transplants with our HPCs. Intravenous injection of HPCs did not lead to engraftment, as reported by several groups (with the exception of AML derived iPSC). Intrabone injections did not lead to engraftment either. When HPCs were transplanted together with MSC scaffolds, we obtained engraftment with both low and high risk HPCs, indicating that they mimic primary MDS samples. HPC-iPSC derived cells were able to generate both myeloid and lymphoid lineages. This is the first time that HPCs-iPSC derived from MDS patients are shown to be capable to engraft in vivo. This data has been presented in the New Figure 2B and New Supplementary Figure 4A-C.

Additionally, the CFU potential of primary sorted CD34+ cells from low and high risk revealed that the low risk samples did not retain CFU potential following serial replating, whilst high-risk retained self-renewal potential and can form colonies after secondary replating.

In our experiments, we did not investigate the impact of Aza on serial-replating capacity as our aim was to determine whether the cells showed resistance to Aza treatment as observed in the MDS27 patient. This analysis indeed would be of great interest to determine mechanisms of therapy resistance that we will follow in the near future. We thought it would be more relevant to focus our efforts on demonstrating the similarities with primary material as well as on the generation of new iPSC lines restoring RUNX1 and SRSF2 mutations to clarify the contribution of CEBPA mutation to the disease progression.

13. Reviewer 1: When comparing MDS low risk to high risk, are the authors using the Crispr control line or ancestral unperturbed line? This is important for the interpretation of the functional and molecular data from these comparisons.

When we compare MDS low risk with high risk, we use the MDS low CRISPR control. We have 5 different lines: wild type (non-isogenic), MDS27-C22 Low-risk, MDS27-C22 CRISPR control, MDS27 C22.7 High risk-1 and MDS27 C22.20 High-risk-2.

Reviewer 1: Minor comments:

- In Figure 4E the authors show gene expression relative to HPC for each individual group. However, this could mask potential key differences initiated already at the HPC stage and the authors should present this data relative to the housekeeping gene only.

The data in Figure 4F (Now Figure 4G) is corrected by the house keeping gene and relative to their respective time 0 (HPCs). The experiment read out is to determine the expression of those genes during myeloid differentiation and hence we think that the correct way of analysing is with respect to the time 0 for each of the individual groups, as otherwise we will be comparing non-isogenic cells.

To satisfy the reviewer, we have nonetheless re-analysed the data in HPCs to determine the gene expression variability between them. The left graph shows the delta-delta Ct values. This analysis reveals low expression for all the markers at the exception of LMO2 in HPCs.

The right graph shows the relative Ct values, as an indication of mRNA abundance in each sample with respect to GAPDH. We can include any of these graphs in supplementary data if required.

- It is not easy to follow the statistical comparisons performed in the figures (all groups compared to all groups, or selected groups) and what the significance refer to or the statistical analysis performed. The authors have used ANOVA but do not indicate whether or what method for multiple testing correction is being used. Can the authors also confirm that any bar without indication of significance was compared but not found significant?

We have decided to represent only the significant comparisons between the CRISPR control and high-risk isogenic lines, as otherwise there are too many comparisons, and it is illegible. The rest of comparisons can be found in the new excel file (Table 4). Dunnett's correction was applied in most of the cases for multiple comparisons. The statistical method used has been included in the figure legends.

Comparison between normal versus low risk should be taken with caution as they are not isogenic lines.

- (cont.) For example, it would be interesting to know whether normal versus low risk in figure 3B was significant as there appear to be big differences, which contrasts findings in Figure 5E for DMSO conditions.

Indeed, although the trend is the same on figure 3B and 5E (Now Figures 2C and 5E, respectively), in the sense that there are less colonies being formed in low risk compared to normal, in 3B (Now Figure 2C) the difference is significant and in 5E is not. These could be due to the different conditions: in Figure 5E cells are culture in the presence of DMSO. The other possibility is that the data in 3B comes from four experiments instead of three, making the data more robust.

We have compared the data side by side, and there are no significant differences between the number of colonies formed by low risk or control cells in these two settings:

- The authors claim that the profiles during the HPC differentiation are very similar for normal and low risk MDS lines but Supplemental Figure 1B show clear differences in the profile with regard to the level of intensity for CD34 expression which questions this conclusion.

Our claim refers to the percentage of definitive HPCs based on CD45/CD34 expression at day 14 of differentiation (red square, Supplementary Figure 2B).

There might be slight differences on the profiles with respect to CD34 expression, which is normal and varies between different experiments. To satisfy the reviewer, we have analysed the geometric mean of CD34 expression in four independent experiments. This analysis did not reveal significant variation between normal and low risk MDS lines.

- With regard to the comments above, lines 107-120 of the introduction contain several overstatements towards the findings of the manuscript and should be revised.

We have removed clonal evolution and written: *Moreover, disruption of the CEBPA bZIP domain led to changes in cellular composition.*

We believe that the new data generated supports the statements towards the findings of our work.

- As indicated above, the methodology should be better documented to provide better clarity of the design of the experiments performed. This pertains to cytokine conditions during iPSCs culture, antibodies and controls for gating strategies in flow cytometry experiments, statistical methods, and details for RNA sequencing analysis. *This data has been provided.*

- There is much text (particularly in result section) and several figures that are purely technical and could be moved to methods and supplemental figures to enhance the focus of the paper.

We have now moved Figure 1 to supplementary figures and brought the CEBPA CRISPR data to main figures. Regarding the text, we will seek advice by the editor regarding this point.

Reviewer #2 (Remarks to the Author):

In this manuscript, Almaghrabi et al describe work using iPSCs generated from a patient with MDS who progressed from low to high-risk disease. Samples both time points were available for analysis. The authors utilize an induced pluripotent stem cell model to generate cell lines and examine how mutations in CEBPA (seen during progression from low to high risk in this patient) alter the transcriptional and epigenetic landscape of the cells to promote abnormal dysplastic differentiation. The authors should be commended on this extensive and detailed characterization. I do, however, have a number of concerns.

I think that the findings are overly broad given, especially given that samples are taken from only one patient and that the control cells were generated from an unrelated patient with an inherently

different genetic background. Thus, there are a number of potential factors that could confound the results here and limit the strength of the conclusions. I think that in general, the analyses comparing the isogenic cells with or without CEBPA mutations is the stronger part of the paper and the comparisons to the wild type control cells less so.

There is also no effort to compare transcriptional changes seen in the iPSC models to that seen in the primary samples. It is always unclear to me how much of the transcriptional changes may be due to the process of in vitro manipulation and reprogramming.

The paper also suffers from some methodological issues, the most significant being the misclassification of an ASXL1 mutation.

I do think that overall, this represents a significant amount of work and provides new data describing a very fascinating area of science involving the evolution of myeloid malignancies and with appropriate revisions would be of interest to researchers working on cancer evolution and on myeloid malignancies.

We thank the reviewer for his/her overall comments. Please see the detailed answers below.

Reviewer 2: -Line 104-106: "How these mutations affect the bZIP domain in the context of MDS have not been explored and it is unknown whether they are causative for a high risk of MDS disease progression." I think this is the fundamental question. There are a large number of reports sequencing patients with secondary AML, and CEBPA is included on most targeted NGS panels. If CEBPA mutations are common progression events then I would expect this to have been reported previously. It's one thing to show that a genetic event can produce disease-like effects in a model system, but if it doesn't occur regularly in human disease, then the it is unclear what the actual implications are. Is this an interesting case report, or does it have wider implications?

The reviewer is of course correct. Indeed, it is known that mutation in transcription factors regulating myeloid differentiation, in particular RUNX1 and CEBPA are common in the progression to secondary AML (Coleman, L et al Blood 2015; Xu, F et al, J Cell Mol Med 2022 and also reviewed in Capelli et al Leukaemia 2022), but to what extent they are causative of disease progression is not known. We have included this sentence in our introduction and added the respective references.

Reviewer 2: -Line 129: The authors report that the patient has an ASXL1 Glu 1102 Asp mutation. However, pathogenic mutations in ASXL1 are almost exclusively frameshift or nonsense mutations that are truncating, and are rarely, if ever SNVs. This mutation has a VAF close to 50% in all the samples sequenced (supp table 1), and thus I think it is most likely a germline SNP. Unless the authors have other data demonstrating that this is a somatic mutation (such as sequencing from fibroblasts or T-cells) and pathogenically affects ASXL1 function, I would not say that this patient has an ASXL1 mutation. In contrast, the RUNX1 fs (but not the RUNX1 SNV, which is also likely germline) and the SRSF2 mutations are found at VAF's consistent with somatic events and these mutations expand over time. These SNV mutations, because they are likely germline, can also not be used to infer normal vs MDS derived iPSC colonies (line 138).

I would like to thank the reviewer for this comment. We have removed in the text ASXL1 mutation. We were not aware that since 2017, the ASXL1 Glu 1102 Asp was not reported. Also, we have removed from the table the RUNX1 SNV.

Reviewer 2: -For the clonogenic and differentiation assays, it would be nice to know how this compared to the primary cells from the MDS that did not under reprogramming. How representative of the original MDS are these iPSC derived HPCs?

The clinical hematology unit provided us with bone marrow smears and trephine from the patient. Pictures from BM are showing deserythropoiesis and myeloid dysplasia similar to the dysplasia observed on our differentiation assays. Moreover, flow cytometry data from the patient shows that patient responded to DA, relapsed 4 months later and did not respond to Aza, showing at this point an increase in early erythroid cells as observed in our cultures. All these data have been included in Supplementary Figure 1 A-D; New figure 1K, 4F and 5H.

Additionally, we performed colony assays with CD34+ cells from the patient before and after disease progression. We observed that under normal conditions the number of CFUs colonies was minimal for the high-risk sample, as previously reported in the literature. Nonetheless, under hypoxia conditions, we observed an increase in the number of CFUs when cells were treated with 500nM Aza, similar to our iPSC clones, and they are able to form colonies after second replating (New Figure 5H), albeit with a less proliferative capacity than those cells treated with 5-Aza (New Supplementary Figure 7E). Additionally, an increase in early erythroid cells is observed when comparing low risk with high risk cells as well as after Aza treatment (New Supplementary Figure 7D), indicating that our iPSC are truly representative of the original MDS.

Reviewer 2: -Line 224: Can the authors explain why they think they were unable to generate iPSC lines from the high-risk MDS sample? Other groups have reported generating iPSC lines from similar CEBPA mutated AMLs (Wang et al. Cell Stem Cell. 2021).

As explained to reviewer 1, we believe that this might be due to the genetic make-up. The group of Eirini Papapetrou successfully generated iPSC from a patient with CEBPA mutation AML32) Wang et al 2021. Nonetheless, on her last article published in Blood Discovery, the group was unable to successfully generate iPSC from an additional patient harbouring CEBPA (AML 34). The mutational background between those two samples was different: AML32 patient had TET2, IDH2, ASXL1 and SRSF2 mutations and AML34 had TET2, STAG2, SRSF2 and CSF3R. These likely points that the genetic background of the patient could dictate the possibility of generating iPSC. Because similar comment has been brought up from another reviewer, we have included this explanation in the discussion.

Reviewer 2: -Line 288-291: "Our results also show that introducing the CEBPA mutation on the background of additional mutations (ASXL1, RUNX1, SRSF2) but not by itself increased the self-renewal capacity of the committed progenitor cells and blocked myeloid differentiation, which is the main characteristic of high-risk MDS and AML." To really make this firm a conclusion, I think the authors

would need to engineer a RUNX1 and/or SRSF2 mutation into the wild type control iPSCs and then show that the CEBPA mutation could transform them. As it is, this is just a single a control sample generated from an unrelated patient. There are numerous other factors that could contribute to this finding.

We understand the concern of the reviewer. We therefore attempted to generate a RUNX1Gly127fs mutation in our wild type iPSC. After two attempts we were only able to generate one clone harbouring homozygous mutation for RUNX1 Gly217fs. Hematopoietic differentiation of this clone was severely impaired with 20-30x less HPCs numbers at Day 14. Also, a reduction in the number of BFU-E colonies was observed after colony assays and the percentage of erythroid cells CD71+/CD235a+ was practically null. No serial replating capacity was observed (see panels below for the reviewer only). As this mutation was homozygous and not heterozygous as found on the patient, we have not included the data in the manuscript. However, the data show that this mutation does show a strong phenotype.

Thus, to reinforce our statement, we decided to revert the original mutations in our clones harbouring the CEBPA mutation (New Supplementary Figure 5C). Reverting the original mutations did not have an effect on HPC differentiation (New Figure 3G) but increased the number of colonies (CFU-Myeloid and CFU-erythroid) when performing colony assays (New Figure 3H-I). Still dysplastic cells were observed (New Figure 3M). The two major phenotypes observed after reverting the original mutations were (i) restore of the erythroid bias differentiation observed in the mutant clones (New figure 3L), and (ii) the loss of repopulation potential (New Figure 3J).

Reviewer 2: -It would be helpful to include flow plots for Figure 5A in a supplemental figure, similar to what was shown in Figure 2E.

As requested, this has been included in New Supplementary Figure 6A. Also, cytopspins for all data points and their analysis have been included in New Supplementary Figure 6B and 6C.

Reviewer 2 -Line 349: Again, these cells do not contain a mutation in ASXL1 (which is also misspelled). This has been removed.

Reviewer 2: -Section starting with Line 352: Other reports (Wang et al. Cell Stem Cell. 2021) have demonstrated that addition of a CEBPA mutation actually sensitized the cells to AraC. This would fit with a model in which cells that are more AML-like with a higher replicative rate are more sensitive to standard chemotherapy. This may be different with hypomethylating agents, which seem to have less of a direct anti-proliferative effect. It might be interesting to examine the effects of a chemotherapeutic such as AraC in this system and determine whether there are differential effects of the two agents.

To satisfy the reviewer, we performed Ara-C treatment as shown in Wang et al. HPCs from day 14 of differentiation were treated with 200nM AraC and counted the cells 48h later using trypan blue. Our data shows that compared to healthy cells, AraC affects the viability of the mutant cells by 50%. This data has been included in New Supplementary Figure 7A.

Reviewer 2: -Line 399-401: It seems odd that the higher risk cells would have a lower proliferative rate given that, in general, higher risk MDS and AML are thought to have more rapid proliferation.

We agree with the reviewer. It seems counterintuitive to get a signature of lower proliferation as it is always assumed that higher risk MDS are highly proliferative. However, our work tallies with the findings reported by Mestrum et al, 2020 <https://doi.org/10.1002/cyto.b.21946> in which they analysed the KI67 expression of 36 MDS patients and 32 MPN patients and compared to healthy controls (50). In their study they found that contrary to MPN, Ki-67 expression was low in MDS in the early maturation stages compare to healthy individuals; with a slight decrease in the Ki-67 positive fraction of CD34 positive blast cells.

Also, Matarraz et al. (2012) <https://doi.org/10.1371/journal.pone.0044321> described a low proliferative activity in advanced and high-risk MDS patients. The authors explained this decrease in proliferation could be due to the induction of quiescence in dysplastic and malignant cells to protect the cells from undergoing apoptosis.

Moreover, in a recent report published in Blood Cancer Discovery (Petti et al, 2022), [10.1158/2643-3230.BCD-21-0050](https://doi.org/10.1158/2643-3230.BCD-21-0050) the authors described a relapse leukaemic cell state characterized by expression of adhesion and migration genes and downregulation of cell cycle genes. These references have now been included.

The examples listed may show that the progression from high-risk MDS to AML involves a restoration of proliferative capacity of the cells by additional mutations / epigenetic changes. Our model (once published) would in fact allow to study this hypothesis, which of course is currently just speculation.

Reviewer 2: -Discussion line 536-539: I'm not sure that the data described in this manuscript inform the biology of de novo AML with CEBPA mutations. That is a disease setting typically without pre-existing mutations, where the CEBPA mutations are likely founding events, and is distinct due to its high response rates to classical chemotherapy agents. This is very different from sAML or high-risk MDS with a CEBPA mutation occurring at progression.

We thank the reviewer for this comment, we have changed this statement.

Reviewer 2 Minor points:

-Introduction line 81: "that the majority of mutations are randomly acquired and not related to the

pathogenesis of MDS..." I'm not sure what "not related to the pathogenesis of MDS" means here. These are initiating mutations for MDS and AML. We have removed this statement.

-Line 125: I would avoid using dates (even month and year), and instead report dates as years (or days) from diagnosis of low-risk MDS. We have modified it and written at diagnosis and two years later.

Reviewer #3 (Remarks to the Author):

In this report, Alghrabi et al, engineer a CEBPA bZip mutation in iPS clones reprogrammed from a low-risk MDS patient to mimic mutation acquisition in progression to high-risk MDS. The Authors show that the CEBPA mutation reconfigures the distal regulatory landscape of reprogrammed cells and impairs granulocytic differentiation, promoting self-renewal of erythroblasts. The Authors also perform scRNA-seq and identify MDS and MDS-CEBPA-bZip-mutant clusters, which they interpret as recapitulating clonal progression. Finally, the Authors demonstrate paradoxical effects of AZA treatment, which may explain treatment unresponsiveness / resistance.

While the model is elegant and has the potential to provide information on the mechanistic effects of the CEBPA bZip mutation in disease progression, it is unclear how it effectively reflects cellular and molecular characteristics of the patient.

A number of the claims are potentially overstated. For example, while the Authors suggest that the model demonstrates the participation of the CEBPA bZIP mutation in MDS progression, this would be more convincingly demonstrated through in vivo testing in transplantation experiments. It is also not clear to what extent the transient increase in BFU-E self-renewal reflects the behaviour of the mutation in patients. Equally, while the Authors claim that BFU-E replatability in engineered clones indicates that the self-renewal effects of the CEBPA bZip mutation require a background of other mutations, this is never formally demonstrated. One possibility would be to establish an inducible system coupling CEBPA KD with expression of the mutant form in the original reprogrammed MDS-low iPS cells.

We thank the reviewer for his overall comments, answers to the questions can be found below.

MAJOR POINTS

1. Reviewer 3: The Authors explain the need to engineer a CEBPA bZip mutation in patient-derived low risk iPS clones with the inability to reprogram the progressed MDS samples. They confirm the presence of the heterozygous mutation and exclude aneuploidy. However, it would be reassuring to exclude the presence of additional off-target heterozygous or homozygous mutations that may confound the conclusions. Did the Authors perform WGS or exome-sequencing of the clones obtained? And is this information available for the progressed MDS samples from the patient?

WGS/exome-sequencing was not performed for this patient. We performed a mutational screening through next generation sequencing using a panel of the 40 most common genes in myeloid disorders. This array is used in clinical settings for the diagnosis of AML: ABL1, BRAF, CBL, CSF3R, DNMT3A, FLT3, GATA2, HRAS, IDH1, IDH2, JAK2, KIT, KRAS, MPL, MYD88, NPM1, NRAS, PTPN11, SETBP1, SF3B1, SRSF2, U2AF1, WT1, ASXL1, BCOR, CALR, CEBPA, ETV6, EZH2, IKZF1, NF1, PHF6, PRPF8, RB1, RUNX1, SH2B3, STAG2, TET2, TP53, ZRSR2. The same mutational screening was performed in our clones before and after CRISPR and no additional mutations could be detected.

Additionally, the increase in CD71+/CD235a+ erythroid population and self-renewal capacity observed after CEBPA CRISPR/cas9 is restored when the additional mutations (SRSF2 and RUNX1) are reverted and thus not due to off-target mutations.

2. Reviewer 3: The low-risk MDS clones from the patient capture characteristic aspects of erythroid dysplasia without affecting progression of erythroid differentiation. They also capture a small but significant reduction of CFC frequency. In contrast, MDS-high risk cells carrying the heterozygous CEBPA bZip mutation have low clonogenicity with impaired myeloid differentiation, but display enhanced erythroblast self-renewal. How does this reflect the clonogenicity and lineage affiliation of patient MDS cells, before and after putative CEBPA bZip-mutant -driven progression? And how do the Authors link these changes with the risk of progression to AML, which is not characteristically erythroblastic? Comparisons with patient material are important to establish the relevance of the iPS model and should be included. Testing the differentiation and transformation potential of the reprogrammed cells in vivo through transplantation would also add confidence to the model.

We understand the concerns of the reviewer and have included comparisons with patient cells as much as possible. We have incorporated the clinical history of the patient, showing the increase in the percentage of blasts as well as percentage of early erythroid progenitors (CD71⁺) during disease progression and after Aza treatment. Thus, the patient clinical data shows the increase on erythroid cells after treatment, similar to our data from colony assays. (See below, data included in Supplementary Figure 1D).

Additionally, we performed a colony assay with the patient sample before and after disease progression and show a dramatic decrease in the number of colonies by colony assay under normoxia (from 280 in low-risk to 10 colonies in high-risk). The low performance of MDS primary sample on colony assays has been reported before. Nonetheless, under hypoxia (5%O₂), the number of colonies obtained after disease progression were much higher (87 colonies) and an increase was observed after Aza treatment (137 colonies). Cells were recovered after the colony assay and stained with myeloid and erythroid markers which revealed an increase on CD71⁺ early erythroblast cells after Aza treatment and after disease progression (New Figure 5H and New Supplementary Figure 7D).

Moreover, we have tested the engraftment potential and differentiation capacity of the HPC-iPSC in vivo by performing transplantation using a humanised model as engraftment via intravenous injection was unsuccessful (as reported by other groups: Kotini et al CSC, 2017; Vo and Dayley, Blood 2015). Our data shows that both low risk and high risk are capable to engraft and differentiate into myeloid and lymphoid lineage. Unfortunately, cells were very fragile, and we could not recover enough cells for further analysis after sorting. The data has been included in New Figure 2B and New Supplementary Figures 4A-C.

3. Reviewer 3: Treatment of progressed high-risk iPS clones with AZA results in progression of the phenotype and potentially explain clinical resistance to the treatment. How does this compare with the response of the patient cells to AZA treatment? Is there evidence of similar paradoxical responses from patient material?

We have the clinical data showing that the patient was resistant to Aza as shown by the increase in the percentage of blast cells (CD34+and/or CD117+) and percentage of early erythroid cells Supplementary Figure 1D).

Also, as explained above, we performed a colony assay with the patient sample before and after disease progression; an increase in the number of colonies was observed in the high risk samples after Aza treatment (New Figure 5H). Additionally, patient data showed the persistence of erythrodysplasia after Aza treatment as observed in our iPSC cells (see New Supplementary Figure 7D).

4. Reviewer 3: Similar to the previous comments, it is not clear how the transcriptional programmes specifically-associated with MDS-low risk and MDS-high risk iPS-derived clones compare with the progression of transcriptional programmes observed in this, or other patients carrying the same progression of mutations. Does the single-cell RNA-seq analysis suggest any combination of surface markers that would allow prospective isolation of MDS cells for analysis of disease progression in vitro or in vivo?

We have performed ATAC seq from CD34+ cells of the patient before and after disease progression and found changes in chromatin landscape between both samples. Motif enrichment analysis revealed changes in transcription factor accessibility as for example PU.1/ETS and RUNX 1 motifs and

enrichment of GATA and AP1, indicating a shift towards a more immature state. A similar shift towards was also observed in our high-risk iPSC when compared to low-risk.

There are several surface markers that could be of potential interest such as CD93, CD44 or CD200, but this has not been validated and we have not used them in combination with the aim to isolate MDS cells for disease progression. Nonetheless, this is a really interesting venue worth to pursue in the near future.

MINOR POINTS

1. Erythroid differentiation of iPS cells does not typically produce adult erythropoiesis. How does this affect the interpretation of erythroid clonogenic and differentiation results? Would the effects of the CEBPA bZip mutation be different in adult haematopoietic cells?

It is true that one of the limitations of the use of iPSC for hematopoietic studies is that when differentiated they do not typically produce adult erythropoiesis. Nonetheless, the role of CEBPA in promoting myeloid and inhibiting erythroid differentiation is well documented in adult mice (Suh, Blood 2006).

Moreover, our colony assays data using CD34+ primary cells from the MDS patient harbouring the CEBPA mutation (high-risk), indeed shows an increase in the percentage of erythroblasts (New figure 5H and Supplementary Figure 7D), indicating that the effect of CEBPA bZIP mutation is not different in adult hematopoietic cells.

2. The differential high-risk vs low-risk reprogrammed MDS programmes include signatures which are not obviously haematopoietic and might translate ectopic expression or incompletely repression of alternative lineages during iPS differentiation. How does that reflect non-haematopoietic effects of CEBPA disruption? Are any of these programmes observed in patients or are they potential confounders?

The nature of the alterations in gene expression was indeed puzzling to us as well, as the cells are clearly of hematopoietic origin as we checked that they were CD43+ before scRNAseq. On trying to understand the meaning of this signature we came across with a very interesting paper which reported a specific signature from relapse-leukaemic cells enriched in genes involved in cell migration and cell adhesion. Other non-hematopoietic signatures, including signatures associated with EMT in AML have also been reported which includes the activation of TGF β and WNT signaling as observed in our high risk HPCs-iPSC (Supplementary Figure 8B).

Note that these cells are also less proliferative, mainly in G1 (Figure 7B). We can speculate that the cells are trying to overcome the differentiation and proliferation block by “moving sideways” and activating ectopic gene regulatory programs as seen in Assi et al 2019. However, this idea is currently

only a speculation and getting to the bottom of this phenomenon is way outside the scope of the paper which is already overly long.

With regards to non-hematopoietic, i.e. potential extrinsic effects of the CEBPA mutation from non-hematopoietic cells: It is true that CEBPA is expressed in other tissues which could influence the differentiation by changing the stroma as these cells also carry the mutation. However, note that when we created a cell line where we repaired the original driver mutations (RUNX1 and SRSF2) much of the phenotype disappeared, meaning that the CEBPA mutation alone, being it in blood cells or the stroma, does not impose a dominant, cell-extrinsic effect on blood cells.

3. Reviewer 3: The Authors represent CFC frequency analysis as relative numbers to control rather than absolute comparisons? How does this affect the results? Are similar trends observed? CFC frequencies in the clones have large error bars; is the same true of control and does it affect statistical significance if absolute rather than relative frequencies are represented? While I recognise the variability of these assays, it is important to show absolute values for full interpretation of data trends. I would also recommend showing individual data points in addition to the error bars in all analyses.

We represented the data from CFC assays in different ways. Comparisons of absolute number of colonies, absolute number of myeloid colonies and absolute number of erythroid colonies (Figure 1C; Figure 2C and 2I; Figure 3B, 3D, 3H, 3J and Supplementary 7B), absolute numbers of all type of colonies (Figure 1D, Figure 2D, Figure 3C and 3I) and relative number of colonies compared to control (Figure 1E, Figure 2E, and Supplementary 7C). Stats were provided for each individual graphs. We have included the individual data points in addition to the error bars in all the analysis as requested by the reviewer.

4. Reviewer 3; The Authors should be careful in referring to the MDS-specific clusters in single-cell RNA-seq analysis as reflecting clonal progression. The reprogrammed / engineered MDS iPS cells are clonal in nature, but the programmes associated with low-risk and high-risk MDS-derived clones do not necessarily reflect clonal progression, and may highlight differential lineage affiliation and differentiation progression in the iPS model. In the absence of clonal tracing studies / comparison with patient material / isolation of cells corresponding to individual clusters and transformation analysis representation of clonal progression is speculative.

We thank the reviewer for this observation. We agree that without tracing studies, it is speculative to talk about clonal progression. We have replaced changes in “clonal progression” by “changes in cellular composition” as also requested by reviewer 2.

In summary, we thank the reviewers for their constructive comments and hope that the paper is now suitable to be published in Nature Communications.

Reviewer #1 (Remarks to the Author):

The authors have in the revised manuscript included new data as well as provided detailed point-by-point responses that have clarified some concerns and unclarities within the originally submitted manuscript.

We thank the reviewer for this statement.

However, in some instances the new data have raised new questions and still many important concerns raised in the first review remain unanswered as further detailed below.

Comments:

1. As also brought up by reviewer #2, the study is still exclusively performed on iPSCs derived from one MDS patient which limits the extent of potential conclusions that can be drawn from these studies. As the authors now themselves bring up in their point-by-point responses, genetic heterogeneity including the order and type of acquired somatic driver mutations as well as the inherited genetic background of the patient, can influence the effect of sequentially acquired driver mutations have on iPSCs generation and differentiation. In light of this context dependent impact, it is unclear whether or not the inferences observed upon acquisition of the CEPBa bZip mutation are specific for this patient in the context of this particular genetic background or applicable to other MDS patients with a different genetic background and MDS composition. This limits the breath of the conclusions of this study as the authors only have investigated this in the context of cells containing RUNX1 G127fs and SRSF2 P95H mutations.

The reviewer is correct in stating that a different genetic background can modify a disease phenotype, as it is commonly also observed in different strains of knock-out mice – which nevertheless are being published. We believe therefore that we were entirely justified in using cells from a representative patient for the generation of our models. However, to satisfy the reviewer and guided by the editor, we have changed the title of the manuscript to reflect that this work was done in a patient sample within the context of the SRSF2/RUNX1 mutation, we have changed the abstract to reflect this fact and also added this information in the main text.

The new title is:

A heterozygous *CEBPA* mutation disrupting the bZIP domain in a RUNX1/SRSF2 mutational background causes MDS disease progression

2. The authors have included new experiments where they compare data observed in iPSCs to data generated from patient MDS27 BM cells collected at different stages of the disease. However, this data raises questions to what extent conclusions observed in iPSCs can be linked to the impact on primary cells. This is in part highlighted by the ATAC seq data where 5-times more enriched or depleted peaks were detected when comparing low and high risk MDS cells from the patient versus the iPSCs. Furthermore, the peaks that are low-risk specific or high-risk specific in the primary patient cells are different when compared to those observed in iPSC. For example, PU.1 motifs which were found to be low risk specific in the iPSCs are high risk specific in the patient. Discrepant distribution is similarly observed for ETS, AP-1, CEBP and RUNX motifs.

We understand the concern of the reviewer, but it would be absolutely impossible to obtain exactly the same pattern. We are not looking at AML in a patient where most cells are diseased and can be purified. These were very difficult experiments to do. They were meant to show that alterations in the chromatin pattern and motif composition follow the same trajectory during disease progression in iPSCs derived and patient cells (we have included this term in the main text for further clarification). Note that the new analysis of patient samples compares samples from the patient before and after disease progression (two years apart), we are comparing CD34+ cells that are mainly healthy (5% would be diseased) with a CD34+ population containing a mixed population of healthy and many more diseased cells. In the case of the ATAC-seq from our iPSC, we are comparing HSPCs coming from one clone with all of the cells containing the SRSF2 and RUNX1 mutation and comparing them with isogenic cells all containing these mutations together with CEBPA. The isogenic iPSCs are going to be more homogenous compared to the cells coming directly from the patient blood obtained two years apart - which is of course the much cleaner system that truly allows to make statements of causality – which, in turn, was the point of this study establishing a new model system for MDS disease progression. Again, we would like to reiterate that this paper shows that the trajectory of motif usage in the isogenic iPSC system follows the same direction as in the patient. We see a similar (not equal) alteration in motif enrichment, with gain of GATA motifs, indicating a more immature state of high-risk cells and an alteration of RUNX1 and other motif usage.

Functionally, the patient cells behave differently when compared to the iPSCs as well. This is striking when comparing the data in Figure 5E to 5H where no difference in total CFUs was observed when comparing low-risk to high-risk iPSCs. In contrast, the 150 CFU's observed for the patient's low-risk cells was reduced to almost 0 for the patient's high-risk cells.

Furthermore, replating in normoxia was not reproduced with the patient's bone marrow cells, this was only observed in hypoxia which also only was tested for the high risk cells and lacks data from low risk for comparison.

As explained above, we are comparing two different scenarios. Low risk patient samples are mixed populations, classified as low risk because the percentage of cells harbouring the disease associated mutations is small. The high risk MDS samples from the patient in normoxia did not grow. Our aim was to be able to obtain colonies to define whether serial replating could be achieved, similar to our high-risk iPSC. As it is known in the field that high risk MDS are capable of colony formation under hypoxic conditions, we took high-risk MDS primary cells and repeated the experiment in hypoxic conditions aiming to obtain some colonies.

Although low risk primary cells were grown in normoxia producing around 150 colonies, second replating was performed under both conditions, normoxia and hypoxia and no colonies were obtained in any of the two conditions for the low risk.

Finally, the authors provide representative images of similarities in aberrant morphology. Although this is quantified in the iPSC generated cells, this is not the case for patient cells.

For patient cells, we obtained the reports from the clinic. The clinical diagnosis is based on the percentage of blasts and presence of dysplasia. The pathologist report does not specify the percentage of aberrant morphology as this percentage is not a criteria for diagnosis. For example, the body of the report regarding the sample at diagnosis states: "*In the smear and a*

cytopsin of the nucleated cell preparation haemopoietic cells predominate with dysplastic features and left shift”.

3. The authors have provided an informative description on the challenges with performing in vivo studies of human iPSC-derived HSPCs. Considering this, they have instead identified an interesting approach through the use of scaffolds and now can successfully engraft iPSC-derived cells in immune-compromised mice. However, the interpretation and analysis of this data is unclear. It appears in Figure 2B that the high-risk iPSCs have lower engraftment potential than control and low-risk iPSCs but this is based on total human CD45 engraftment in the scaffolds. As the most striking impact of high-risk iPSCs was observed in their BFU-E serial replating potential, it is surprising that the authors have chosen to focus their analysis exclusively on human CD45 expressing cells, and not cells of the erythroid lineage that have down-regulated or fully lack CD45 expression. This has previously been critical for the identification of in vivo engrafting MDS cells when the phenotype is propagated through the erythroid lineage.

We thank the reviewer for this comment. The transplantation experiments requested by the reviewer had been conducted as part of follow up studies and future experiments aimed at further characterising the disease phenotype which are outside the scope of this study. Consistent with reports in the literature, we have also observed that erythroid differentiation of human cells is undetectable in NSG-SGM3 mice. This phenomenon may be attributed to two primary factors: (1) insufficient cross-reactivity between murine cytokines and human cells, which likely hampers the development of these cell lineages, and (2) phagocytosis of human red blood cells and platelets by murine macrophages.

Note that hEPO was not administered, we therefore cannot yet assess whether erythropoiesis can be detected in our humanized scaffold model, which requires further investigation. Therefore, our analysis focused on detecting total human CD45+ cells. We believe that this result is enough to address the previous comments and we are allowed to publish the initial observations. Note, that, although the engraftment was slightly lower, there was no statistical differences in the transplantability of LR and HR MDS.

4. The authors use the observation of aberrant morphology during erythroid differentiation (Fig 1J and 5D) to support the iPSC relevance for MDS studies. However, the aberrant morphology is observed at the earliest stages of culture and declines throughout erythroid maturation. The authors do not show data for day 0 for this morphology which could be important towards the conclusion of the authors as this could indicate that the mutations also impact the generation of the hematopoietic progenitor stages, which so far the authors have claimed are normal. This is also relevant for the interpretation of all functional assays, as it could mean that the patient iPSCs are at an earlier or later progenitor stage already prior to seeding in myeloid or erythroid differentiation assays.

It is true that the aberrant morphology observed at the earliest stages decline throughout erythroid maturation as observed in the Figures 1J and 5D, but this could be misleading, as in late days we get less nucleated cells. It is due to differences in maturation and apoptosis reducing the number of overall nucleated cells in a mixed population.

Figure 1J and 5D quantify aberrant erythroid morphology during erythropoiesis. We made attempts to perform this at time 0, however to obtain meaningful data at day 0 a large proportion of the culture needed to be assayed, severely reducing the number of cells in the

culture to perform a differentiation kinetics study for 18 days (collecting cells at 5 different timepoints).

In an attempt to satisfy the reviewer we have taken some HPC cells (Day 0) from BU3.10 (control) and C22 (low risk) that we had frozen and perform a DiffKick staining (below). As it can be appreciated, morphology is not altered in low risk sample compared to control.

Moreover, note that we have shown by flow cytometry that the HPCs at day 0 are very similar between the different lines (control, low risk and high risk) and that the patient iPSCs are not particularly skewed towards a specific lineage or at earlier/later progenitor stage prior to seeding in myeloid/erythroid differentiation assays; all iPSC lines started at similar stages of differentiation (see below).

Inclusion of day 0 timepoint would be important for supplemental figure 6A as well.

Day 0 timepoint for supplementary Figure 6A has been included as requested (see below).

Regarding aberrant morphology, data in Figure 5D shows no differences between low-risk and high-risk iPSCs, and low-risk iPSCs are also significantly different from normal iPSCs.

The reviewer is correct. As shown in Figure 5D, there is no differences between low-risk and high-risk iPSCs as shown by the p value in the graph ($p = ns$), and low-risk iPSCs are significantly different from normal iPSCs ($p < ****$ in the graph).

5. The authors have provided reasonable comments to the RNA sequencing data. However, this could benefit from more focused analysis. For example, by performing more cluster dependent analysis for comparing low to high-risk MDS HPCs the authors could reveal shared gene expression changes when comparing these two stages in distinct clusters (e.g. cluster 1, 3, 7, 8 where there are sufficient cells in both). In addition, the relevance of the pseudotime analysis in figure 7I is questionable, particularly since the authors have pooled cells from low risk HPC, low risk erythroid cells, and high risk HPCs.

According to published analysis (Liu, BMC genomics, 2023), meaningful, statistically significant differential gene expression analysis requires a minimum of 2000 cells per cluster.

Unfortunately, we don't have a minimum of 2000 cells for low risk and for high risk in each cluster, and thus we don't have the power to perform such an analysis. Given this fact, we did not wish to include such an analysis in the manuscript.

The number of cells per cluster are:

Cluster 1 is represented by 4554 cells in low risk and 331 in high risk.

Cluster 3: 116 cells in low risk and 2271 in high risk.

Cluster 7: 349 cells in low risk and 948 in high risk.

Cluster 8: 552 cells in low and 613 in high risk.

We therefore have performed a more focused analysis of our scRNA seq data, which highlights the difference in developmental trajectories and population composition in the different samples. For example, cluster 1 which corresponds to 80% of the cells, has a signature described in supplementary Figure 8A. Cluster 6 and 7 are mainly enriched in a leukaemic stem

cell signature, with cluster 7 being more defined and enriched in high risk. Moreover, different genes associated to MDS/leukaemia which are common to the clusters or specific to different clusters have been identified. Following the hopeful publication of this paper, we hope to perform more in depth analyses of the different cellular phenotypes and expression profiles in our model.

Through pseudotime analysis (Figure 7I) we can delineate the fate of cluster 6 in low and high risk individuals. This analysis tallies with Figure 7E in which we can see that cells from cluster 6 start to have a leukaemic stem cell signature which is augmented in high risk individuals specially in cluster 7. It is standard and necessary to pool/integrate the cell data to perform this kind of analysis. Indeed, pseudotime analysis is used by many scientists precisely to analyse data from multiple individuals, conditions or time points; there are non-exhaustive examples, including from the group of Eirini Papapetrou: Wesley et al, Cell Reports 2020, Blood Cancer discovery, 2023 that the reviewer referred to in his/her first revision.

6. Page 5, second paragraph, sentence starting with “These findings suggest that cells harbouring ...”. The authors refers to three mutations when there are only two reported in Table 1.

We thank the reviewer for identifying this mistake. Indeed, there is two mutations and not three, as the ASXL1 mutation Glu1102Asp is a SNP as appointed by reviewer 2.

The authors also state that RUNX1 and SRSF2 were the predominant clone in the patient which is not supported by a VAF of 15% in the patient (minor clone).

Definitely, a VAF of 15% indicates a minor clone and this statement has therefore been corrected.

7. In Table 1, sample from 2015 is shown twice. There are also several rows with no text in Table.

We apologise for this error as we have overlooked the duplication of the 2015 sample. The rows with no text are due to eliminating ASXL1 from the Table as requested by reviewer 2. We have edited the table accordingly.

8. The title of the second result section states that the low-risk iPSCs have normal hematopoietic progenitor potential but this is not in line with the results showing lower CFC potential. In the text describing this data, the authors only refer to lower myeloid CFCs, when in fact also erythroid CFCs are significantly reduced (Fig 1C). The authors also claim an “evident bias” toward erythroid CFC but this is observed for normal iPSCs as well.

We refer to normal hematopoietic potential from the point of view that they could give rise to myeloid (mono and granulocytes) and erythroid lineages. The difference in myeloid CFCs was more pronounced than for BFU-E (4 stars and 2 stars significance compared to 1 star). We have therefore removed “although we noticed an evident bias towards the formation of erythroid colonies versus myeloid colonies”.

9. In reference to Figure 1G the authors state that the expression of CD235a was “very low” at early stages of differentiation. However, when comparing early timepoints (day 0-7) to late

timepoints (day 11-18) the maximum CD235a expression level is observed in the early timepoints (in particular day 7).

Thank you for the clarification. It was probably not properly expressed. We referred to the percentage of cells expressing only CD235a, no double positive cells CD71/CD235a. We have modified this sentence to clarify:

"...while the percentage of CD71- cells expressing the mature marker CD235a+..."

10. The quantification in Figure 1H does not seem to be correct according to labelling of the graphs. For example CD71+ cells represent close to 50% of the cells at day 4 (not 25% as shown in the graph).

Figure 1H is correct. The graphs represent CD71+ single positive (negative for CD235a), the middle graph is double positive, and the graph on the right represents the CD235a+ single positive (negative for CD71-).

We have modified the labelling to avoid confusion.

11. The authors show the same data several times in their figures which seems unnecessary as the authors themselves in their point-by-point responses claim that the manuscript have many figures. As an example, the authors show the same CFC data in three different figures. For the same reason, Figure 2 should only focus on the data from the CRISPR control line and high-risk lines, without inclusion of the lines shown in Figure 1.

This will be discussed with the editor, as reviewers 2 and 3 have not commented on this. Indeed, one of the reviewers wanted to see both the proportions and absolute values.

12. In reference to Figure 2, the authors claim high-risk iPSCs mainly affected the myeloid lineage, when in fact it only affected the myeloid lineage and erythroid CFCs were preserved similar to low-risk (Fig 2C-E).

We thank the reviewer for pointing out this error. It was mean to say "only". The typo has now been corrected.

13. The authors states in their responses that statistical significance is shown only for those reaching a significant p-value but this does not seem to be consistent throughout the manuscript where there are figures showing ns and also significant values not included. Just as an example, the significant p-value comparing normal iPSC to low risk iPSCs in Figure 2C is not included. Although the authors have improved some of the of documentation for their figures, they can improve this further, especially in the figure legends to enhance clarity to which comparisons are being made etc.

We have included p values comparing normal iPSC to low risk iPSCs in Figure 2C to appease the reviewer. Still, we would like to reiterate that normal and low risk iPSC cells are not isogenic, and therefore the statistical comparisons should be taken with caution. In our previous revision we provided an additional table (Table 4) containing p values (ns or significant) for all comparisons. Within the statistical section in methods, we included a sentence reiterating that all the statistical comparisons can be found in the Table 4.

Furthermore, the authors have included representative isotype control but this appears to be provided by only one timepoint. To document their analysis, an isotype for each timepoint should be provided. This is particularly critical for in vitro cultures supporting maturation of myeloid cells which are known to upregulate Fc-receptors that can facilitate unspecific antibody staining.

We would like to reassure the reviewer that we perform a 45 minutes Fc block prior to staining to prevent non-specific antibody binding. This information can be found within our methods section: *Cells were subjected to 45 minutes Fcblock (eBioscience, 14916173, Clone AB468581) on ice prior staining with antibodies.*

Below there is an example of a myeloid differentiation and erythroid differentiation timecourse experiment with isotype control performed at different timepoints as requested. This has been included in Supplementary Figure 2E.

14. The authors make reference to Figure 2I when stating results from a fourth replating but there is not data in Figure 2I on this.

This is a misunderstanding; our sentence refers to the 3rd replating. No data from 4th replating is presented because there were no colonies: *Cells from the CEBPA^{bZIP-fs} mutant clones (high risk) were able to proliferate in third-replating whereby the number of BFU-E had decreased to 149±29 and 133±18 colonies, respectively and cells lost their proliferation capacity at the fourth re-plating (Figure 2I).*

We have now moved the bracket with Figure 2I so there is no confusion: *Cells from the CEBPA^{bZIP-fs} mutant clones (high risk) were able to proliferate in third-replating whereby the number of BFU-E had decreased to 149±29 and 133±18 colonies, respectively (Figure 2I) and cells lost their proliferation capacity at the fourth re-plating.*

15. In Figure 3B and 3D the same BFU-E data is shown with different significance results. This should be clarified.

This is due to the different statistical test used, ONE way ANOVA vs 2-WAY ANOVA as stated in figure legend.

16. The authors make general statement that high-risk iPSC have disrupted myeloid differentiation. However, based on the data shown in Figure 4C an alternative interpretation is that high-risk iPSCs fail to generate mature granulocytes which rather is replaced by generation of mature monocytes. This does not seem to be considered or discussed by the authors. How does this enhanced generation of monocytes in high-risk iPSC align with the molecular data and data from the patient?

Thank you for pointing this out as this is a very interesting observation that somehow escaped our attention amongst the multitude of revisions. Interestingly, this phenotype of loss of granulocytes and increase in monocytes aligns with the loss of PU.1 and gain of AP1 motifs in both high risk iPSC and high risk patient samples as observed in the ATAC-seq. We regard this as very important and we will certainly follow this up but we cannot do this here. We know that ES-derived hematopoiesis generates primitive unilineage macrophages independently of the presence of PU.1 – they can only make CD11b+ macrophages, definitive myeloid precursors are bipotent and are dependent on PU.1 (See PMID: 10381505; PMID: 10608497; PMID: 10918591). Primitive macrophages populate the tissues (PMID: 29311541). It appears that the proportion of such cells is increased in the patient as well. To highlight this fact we have added a sentence and additional references in the Discussion.

17. With regard to Figure 4G where the authors have included additional data in the point-by-point responses, this reviewer is still not able to interpret the data shown. As stated in the previous review, this data be better to display as a deltaCT values where expression relative the housekeeping gene is shown. The authors state that the left figure in the point-by-point responses show delta-delta CT values but this does not have any indication for what adjustment has been done considering that delta-delta Ct refers to adjustment to a housekeeping gene and a reference population.

The left figure is the delta CT relative to the housekeeping gene without the use of a reference population. We have changed the label to the graph and included as part of Figure 4G.

18. In supplemental Figure 8B what does HSC refer to?

This is a mistake, it should say HPC. This has been corrected.

19. Figure legend 1G states that this shows a dot plot which is not correct as this is a contour plot.

This has been corrected.

20. What does HPC represent in Figure 1B?

Number of “floating” live cells obtained at day 14 of the HPC differentiation process from iPSC.

21. The authors reference Table 2.5, 2.6, 2.7 and 2.8 in the manuscript but these are not provided in the review documents.

We apologise for this mistake. These are the provided in the original protocol by Corces, et al 2017 (reference 92). We have removed these tables from the text.

Reviewer #2 (Remarks to the Author):

I appreciate that the authors have so carefully addressed my concerns and points. While they were not able to directly compare the transcriptome of primary high-risk MDS cells and their reprogrammed cells, they do show that phenotypically their model recapitulates the primary cells quite well, and I think it was probably outside the scope of this work to do more here. Overall, this is very nice work and I have no further significant concerns. I congratulate the authors on their manuscript.

There is a small typo on page 14 from discussion: “...highlighting that the complex genetic background of high-risk and AML patients dictates...”. The word MDS appears to be missing after high-risk in this sentence.

We are pleased to hear that the reviewer comments have been addressed.

The typo In page 14 has been corrected, indeed the word MDS was missing.

Reviewer #3 (Remarks to the Author):

I am satisfied that the Authors have addressed the points raised during my initial review, and I acknowledge the significant amount of work put into this revision.

I would just ask the Authors to review the final section of the Results and abstain from using terms like clonal composition and emergence of new clones when referring to their iPSC model. Instead, they could state that they observe a change in cellular composition and differentiation hierarchies when comparing low-risk and high-risk isogenic iPSC models, which reflect the changes in clonal composition and disease progression observed in patients.

We thank the reviewer for this comment that we have previously overlooked. This statement has now been corrected.

In summary, we thank the reviewers for their careful and thorough review which has made for a much better paper and hope that it is now suitable for publication.

REVIEWER COMMENTS

Reviewer #1 (Remarks to the Author):

In the revised manuscript by Almaghrabi and colleagues the authors have clarified many of the points raised to the previous version. The inclusion of additional data and new adjustments to the text has resulted in an improved manuscript. In particular, the adjustments to indicate that the work was done within the context of the SRSF2/RUNX1 mutational background is well appreciated. Although most points raised to the previous version are addressed either in the manuscript or through clarifications in the point-by-point responses submitted by the authors, there are still some remaining issues that remain unclear or where additional analysis of existing data would be important.

We are pleased that the reviewer appreciates the changes included in the manuscript.

Presentation and description of ATAC-seq data

In the responses the authors state that the patient data from the low-risk stage represent only 5% diseased CD34+ cells, where the rest are healthy. This statement is surprising considering no wildtype iPSC could be derived from the low-risk stage and the variant allele frequencies indicate that 60% of the cells are RUNX1 mutated and 30% of the cells also contain the SRSF2 mutation (based on Table 1). At the high risk stage the VAF has increased to almost clonal levels, where close to 80% of the cells carry the three mutations (RUNX1/SRSF2/CEBPA). Does this mean that the CD34+ cells used for ATAC-seq have much lower VAF compared to the peripheral blood cells used for iPSC generation? If so, the VAF of the CD34+ cells should be shown as a separate figure/table.

We understand the reasons why the reviewer finds surprising that no healthy iPSCs were obtained. We can only comment based on the facts that we did not get any healthy clones despite performing the reprogramming three times with different non integrative approaches (Sendai virus and episomal reprogramming). As mentioned previously, one possible explanation is that we pre-cultured the cells for 8 days prior to the transduction, which may have been more beneficial for the mutant cells. With regards to the VAF of patient cells: The CD34+ cells used for ATAC-seq were from the same batch as those used for mutational screening. The mutational screening was performed on the whole population, as is done clinically for patients with MDS and AML, rather than on isolated CD34+ cells. Consequently, the VAF data obtained are from a mixed population. Clinical data from the patient's bone marrow indicated that at the time of diagnosis there were 5% CD34/ckit+ cells (flow cytometry data) and 6% of blasts (morphological examination). This is why we previously stated in our previous rebuttal letter that at the low-risk stage there would be around 5% diseased cells, based on the blast percentage, which then increased to 30% blasts (morphological examination) as disease progressed. Assuming the VAF is the same in the CD34+ population, based on the mutational screening, 60% of the population would have the RUNX1 mutation and 30% the SRSF2 mutation. This results in a

maximum of 30% of cells having both RUNX1 and SRSF2 mutations, another 30% harbouring only RUNX1 mutation and 40% with no mutations. Therefore, as previously explained, our analysed cell population is highly heterogeneous as compared to the iPSC generated cells. To illustrate this fact for review only, we generated a heat map which shows peaks ranked by median peak height with respect to HPC-iPSC low risk sample. We prefer this to Venn diagrams as we assign a statistical significance to each peak. This analysis shows the high number of peaks in the patient samples not present in the more homogeneous iPSC derived cells, denoting higher heterogeneity. Extra

peaks start to disappear after disease progression but are still present. Moreover, note that at the different stages of disease progression, the different mutations affect different cell populations differently, with the CEBPA mutations blocking myelopoiesis which is not the case in the RUNX /SRF2 mutant background. We did not include this result in the paper as we show a table with the number of cell types in the patient which should be enough.

Secondly, in both iPSC and CD34+ cells the transition from low-risk to high-risk is defined by the acquisition of the CEBPA mutation. Although there are many reasons for why the many up- or down-regulated ATAC peaks detected in the patients CD34+ were not found in the iPSC, one would expect that the majority of the peaks seen in the iPSCs also would be observed in the patient's CD34+ cells. To efficiently show this, the authors should include two Wenn-diagrams, one for the gained and one for the lost ATAC sites, to indicate shared differentially accessible sites among patient and iPSC during disease progression. If the motif analysis on these shared sites demonstrated enrichment for GATA, AP-1 and RUNX1 sites, this would significantly strengthen the authors conclusion.

We politely disagree with the reviewer on this matter. Such an analysis is only possible by performing scATAC-Seq in MDS and iPSC-derived cells and comparing the relevant clusters, which is something we were not asked to do a year ago and are now not able to do as we do not have any more access to primary material. The patient is dead. As shown above, the patient cells are way more heterogenous than the iPSC derived cells. Moreover, factors such as culture conditions, cytokines, environment, oxygen levels and niche components have a profound effect on chromatin landscape (see for example a publication from one of us which highlights this fact in fine detail in murine ESC differentiation (Edginton-White et al., Nat Comm 2023). We therefore opted to analyse how disease progression changed the motif composition of differential peaks as a readout of the mutation based changing trans-regulatory environment and noticed a similar trend as shown again in the figure below comparing patient (P) with iPSC-derived cells. In both settings, we see an increase in AP-1 and GATA motif enrichment as compared to RUNX motifs in differentially accessible sites.

That is all we could do, and we stand by this result.

However, to add additional information for review only we have generated Venn diagrams to show the overlap of all distal sites between the C22 iPSCs (low-risk) and the MDS-low risk patient sample, followed by a motif enrichment analysis on the overlapping peaks. The same analysis was also performed between the high-risk iPSC and the MDS-high risk patient sample. The result again shows that we are really comparing mixed populations with each other, but again, we see that the cistrome has been reprogrammed to contain a higher proportion of AP-1 and GATA motifs in the high-risk samples. This result can actually be explained by the single cell gene expression comparisons between clusters which the reviewer asked us to do and which is explained in more detail below. As shown on the Venn diagrams, 28% to 34% of the peaks are shared, and those shared peaks are enriched for GATA, AP-1 and RUNX motifs, with GATA motifs specific for the high- risk samples. Given the

heterogeneity of the populations, we find the 30% similarities in the cis-environment quite remarkable given the differences between in vitro and in vivo conditions.

Lastly, it is not fully clear what the authors mean with “trajectory during disease progression” when they summarize these results. An alternative way to phrase these observation is: “In summary, our data shows shared alterations in the chromatin pattern during disease progression in iPSC-derived and patient cells” (assuming this is supported by the Wenn-diagram of shared peaks).

We have re-phrase this sentence as requested by reviewer.

Interpretation of patient CFC replating data

The experimental design for this experiment does not exclude that the replating potential seen for high-risk patient cells (Figure 5H) is due to initial culture in hypoxia prior to replating in hypoxia. As initial culture in hypoxia was not performed for low-risk cells, one can therefore not exclude that also low-risk cells could have replating potential. To describe these results, I have the following suggestions:

Line 425, change to: “Although not performed with low risk iPSCs, under hypoxia (5% CO₂), CD34+ cells from MDS27 high risk were capable of forming colonies and the number

Line 428, remove “only” resulting in: “Additionally, high risk cells were able to form colonies after second replating

These suggestions have been included.

Analysis of RNA-seq data

The previous comments raised towards the RNA sequencing analysis remains as this could reveal important and critical dysregulated genes within more homogenous compartments shared between the two stages of disease. The authors’ argument for the limitation in cell numbers based on the published analysis by Liu et al. in BMC Genomics 2023 is not in line with what Liu et al state: “If a study anticipates DEGs with modest differences, the study should aim for having 2,000 or more cells in a cluster in order to identify the majority of DEGs that would have been identified by a bulk RNA-seq analysis of thousands of physically purified cells. Such studies should be cautious in interpreting a lack of DEGs from clusters with fewer than 100 cells. On the other hand, clusters with as few as 50–100 cells may be sufficient for identifying the majority of DEGs that would have extremely small p values or transcript abundance greater than a few hundred TPM in a bulk RNA-seq analysis.” What Liu et al argues is that meaningful analysis can be done with as few as 50-100 cells per cluster, with the caveat that this is not expected to pick up all differentially expressed genes, but it will pick up the most significant DEGs and/or those with the highest fold difference in fold expression. The

requirement for 2000 cells or more is indicated for studies where the intention is to identify all DEGs including those with very modest changes.

This request actually turned out to be a good suggestion. As requested, the DEG analysis for the clusters has been conducted. The DEG analysis compared cells C22-7 High vs. C22 Low risk for clusters 1,3,7 and 8. A gene was considered differentially expressed if it had a fold-change > 2 and Adj. P-Value < 0.1

The volcano plots below show the number of genes upregulated and downregulated in the high-risk cells. The analysis of DEG revealed upregulation of a number of MDS/AML-relevant genes such as the upregulation of Ly6E expression (which has been associated with poor survival outcome in multiple malignancies) in high-risk samples across all clusters. In addition, the data revealed the upregulation of multiple AP-1 family members in high-risk compared to low-risk in clusters 1,3 and 8, which supports the increased enrichment of AP1 motifs in our ATACseq data from this population. Note that we have shown AP-1 to be an essential factor for the growth of *CEBPA* mutant AML cells (Adamo et al., 2023, Leukemia). In addition, we see a downregulation of cell cycle regulators and S-phase-specifically expressed genes (such as Ki-67 and histone genes) which tally with the fact that high-risk HPSCs have a much-diminished proportion of S-phase cells (Fig 7B) whilst upregulating multiple genes expressing signalling molecules. The latter is again consistent with a cell cycle block, something we also observed in AML cells treated with a growth factor receptor inhibitor (Coleman et al., 2024, iScience). Note also the up-regulation of *SOX4* in cluster 3 which has been shown to be a C/EBP α -repressed gene and is important for self-renewal of leukemic cells with a *CEBPA* mutation (Zhang, Alberich-Jorda et al., 2013, Cancer Cell). In summary, this analysis is entirely consistent with our cells displaying the molecular features of *CEBPA*-mutant cells.

This data has been included in Supplementary Figure 8E and New Supplementary Tables 3 which will give the community ample material for data mining.

Up-Regulated = High in C22-7
Down-Regulated = Low in C22-7

Minor point:

Line 179: To avoid confusion, I suggest that the title is change to “.....retain myeloid and erythroid progenitor potential” instead of “.....display normal hematopoietic progenitor potential”.

This sentence has been changed as requested by reviewer

We believe that we have now addressed the reviewer’s comments as much as we possibly could and hope that our paper can now be published.

Reviewer #1 (Remarks to the Author):

Almaghrabi et al. have provided further clarification in the response letter and included new data in the manuscript that addresses most issues raised in the previous review. In particular the new RNA sequencing analysis has provided further strength and information to the identification of candidate genes impacted by the acquisition of the CEBPa mutation.

We are pleased to see that the reviewer finds that the new RNAseq analysis further strengthens our work by (i) confirming changes in the expression pattern in specific populations of cells consistent with what is known about C/EBPa mutated AML and (ii) provides additional information to the identification of candidate genes impacted by the acquisition of the CEBPa mutation.

However, the authors have chosen not to follow the suggested analysis for the ATAC sequencing analysis to provide more insight towards the degree for how molecular changes observed in iPSCs are recapitulated in the corresponding patient cells. As I rather agree with the last argument by the authors, I therefore still strongly believe that the proposed analysis would be informative, important and further strengthen the authors data, as well as the readers interpretation of the data. The proposed added figure also does not require single cell analysis as my question is quite simple: how many of the 1691 peaks significantly enriched in high risk and 1576 peaks significantly enriched in low-risk iPSC (figure 6a) were also significantly enriched in the corresponding CD34+ high-risk (among the 8290 sites) and low-risk (among the 7795 sites) cells from the patient (Figure 6F), respectively.

We reiterate our previous comment. It is really not simple at all. The heterogeneity of the primary samples and the unknown contribution of the different diseased cells in the patient makes it impossible to obtain a statistically significant motif enrichment result in a bulk analysis, in particular at the low risk stage. We simply do not know the baseline here and which cells these peaks come from and whether we miss peaks that are there but disappear in the background to the small number of cells in the population. This is the main reason why single cell analysis is imperative even if we have a 30% overlap of peaks overall. We tried to explain this in our last rebuttal by outlining the numbers as we got them from the clinic. We hope it is clearer now.

As the authors argues that “culture conditions, cytokines, environment, oxygen levels and niche components have profound effect on chromatin landscape” thereby themselves raising major questions towards the relevance of performing ATAC sequencing analysis if the goal is to use the iPSC platform for identification of molecular changes of relevance to disease biology in the leukemic cells in the patient. The above argument provided by the authors further does not fit with their following responses in the manuscript where they state “we find the 30% similarities in the cis-environment quite remarkable given the differences between in vitro and in vivo conditions.”

Yes, of course, the exact set of enhancers that can be accessible will be influenced by the in vitro versus in vivo conditions. However, the similarity in terms of enriched motifs associated with disease progression in the common cisrome i.e. the sum of all cis-regulatory elements serves as a very strong argument in favour of the physiological relevance of our approach, illustrating the similarities in the alterations of the trans-environment in vitro and in vivo. Additionally, we also need to reiterate that this is now a very big study, and the validity of our model does not only rest on our ATAC-Seq data. As detailed by all other reviewers plus additional experts and new co-authors, the detailed cellular analysis of our iPSC model, showing erythroid and myeloid dysplasia, block in granulocyte differentiation, retained treatment response, appropriate transplantation phenotype and much more is consistent with the

clinical phenotype of the patient and strongly validates our in vitro system, together with the other analyses such as additional genome editing and scRNA-Seq.

Minor comments:

- Line 59: based on the concerns the authors themselves raise in the rebuttal letter regarding the impact of culture conditions, cytokine etc, on iPSC, the authors should remove “faithfully”.

This refers to the patient disease phenotypes upon hematopoietic differentiation, which we still believe that is faithfully recapitulated.

- Lines 136-137: remove “affecting their expression” as the expression level of these genes is now show in the figures. *As requested, it has been removed.*

- Lines 626-628: Dependent on how the authors respond to the suggested ATAC analysis, and what that analysis shows, the statement starting with “Importantly, a similar ...”. One potential alternative here is to replace the sentence with: “The same motif families were enriched in CD34+ cells from iPSCs and the patient following disease progression, suggesting shared pattern of chromatin reorganization upon acquisition of the CEBPa mutation”.

We think that “the same” is a stronger statement than “similar”. Nonetheless, as suggested, we have removed “importantly, a similar...” and included the reviewer’s sentence:

The same motif families were enriched in CD34+ cells from iPSCs and the patient following disease progression, suggesting shared pattern of chromatin reorganization upon acquisition of the CEBPa mutation and thus, reinforcing the validity of our in vitro system.

- Line 1273: The indicated fold change cut-off is wrong.

The cut-off that we described is correct. A gene was considered to be differentially expressed if it had a fold-change greater than 2 ($\log_2 \text{fold-change} > 1$) and an adjusted p-value < 0.01 . We have included $\log_2 \text{fold-change} > 1$ within the figure legend to avoid confusion.